# LEARNING TO ACQUIRE RESOURCES IN COMPETITION

## ABSTRACT

We consider multiple agents competing to acquire stakes in some costly divisible resource (*e.g.* shares of a financial asset, compute resources, or commodities) over time. We propose a novel game-theoretic model for this problem that generalizes settings studied in diverse literatures, and analyze it under different assumptions on agent information. Given complete-information, we establish the existence and uniqueness of a pure Nash equilibrium (NE) in this generalized setting. This is shown to be efficiently computable but has worst-case unbounded price of anarchy. Alternatively, under partial-information with a common prior, we establish the existence and uniqueness of a Bayesian Nash equilibrium (BNE), which is also efficiently computable. Finally, we propose a more realistic learning setting for the game, where agents have partial information but no common prior. Instead, they must learn how to act given online contextual feedback from interactions in stochastically sampled game instances. We provide sufficient conditions on agents doing simultaneous no-regret learning for convergence to Bayesian coarse-correlated equilibrium (BCCE) or last-iterate convergence to the BNE. In each setting, we provide detailed simulations, which empirically validates our theory and provides new insights into strategic behavior of resource acquisition.

## 1 INTRODUCTION

Consider multiple traders attempting to acquire a position in a stock ahead of an earnings release, under the belief that its price will rise afterward. If each trader were acting in isolation, they might follow a classical optimal execution strategy – such as that of Almgren & Chriss (2000) – to minimize their trading costs. However, if multiple traders are pursuing their strategies simultaneously, their aggregate activity influences prices and liquidity. This interaction transforms the problem from one of individual optimization into one of understanding the intra-agent strategic behavior, where each agent's decisions affect the market environment faced by others.

This challenge of acquiring costly resources in competitive, dynamically priced environments extends well beyond financial markets. For instance, a firm training a large machine learning model may need to secure substantial cloud computing resources within a given time frame. Here too, spot prices are shaped by aggregate demand across many users, requiring firms to account not only for their own scheduling and budget constraints but also for how their actions interact with others (Shastri & Irwin, 2018). Further, agents in many such environments may only have partial or incomplete knowledge of other market participants or the market itself.

While recent works have attempted to capture these strategic perspectives, they do so with several limitations. Chriss (2024b;c;a) all consider a complete-information setting, which is unrealistic in all but very limited scenarios. Chriss (2025); Kearns & Shi (2025) more recently consider some extensions to deal with this, but these also have major limitations, requiring shared common priors over uncertain information or repeated play of fixed game instances respectively. Furthermore, all of these works: (1) require agents to acquire a fixed target position, thereby ruling out more general action constraints; (2) do not allow for custom objectives that agents may wish to optimize alongside acquisition costs; and most importantly (3) do not address the computational and learning challenges that arise when agents act under incomplete information without common priors. In addition, these works are all finance-specific. The broad scope and practical relevance of this problem thus necessi-

tates a general game-theoretic framework that can gracefully accommodate diverse applications and practical limitations of real-world markets, which is the foundational premise of our work.

In this work we propose a general competitive resource acquisition framework for dynamically priced markets. At the core of our model is the fact that prices respond to the aggregate actions of all participants. Beyond this, our game-theoretic framework makes minimal modeling assumptions. Information may be imperfect or asymmetric, with arbitrary structure, and agents need not know the information structure a priori. Moreover, our framework allows for rich heterogeneity in agents' objectives: agents may target some fixed position or instead seek to maximize a personalized utility function on their final position, and different agents may have different goals. Finally, our framework accommodates most kinds of constraints on behavior that may be important for practical application, such as no short selling constraints, limitations on how much agents can buy or sell at each time step, and limits on the allowed position the player can have at any time. Taken together, these features give a flexible game theoretic model that can capture most common goals, constraints, and information limitations of market participants in real-world resource acquisition problems. Our results and insights herein are thus of practical importance to a wide array of settings.

## 1.1 OUR CONTRIBUTION

- In Section 2 we introduce a novel model for this problem, which generalizes and improves on past settings, by allowing for convex constraints, concave idiosyncratic utility functions, and unknown or incomplete information available to the agents.
- In Section 3 we characterize the complete-information equilibria properties of this game. Even within our very general model, the game still has a unique, pure NE that is efficiently computable. We also show that in the worst-case, the price of anarchy of this game is unbounded.
- In Section 4 we consider the partial-information Bayesian setting: each agent only observes their own private information (their "type"), but all agents have common knowledge of the prior distribution over agent types and game parameters. We extend the complete-information results here, establishing the uniqueness and efficient computability of the Bayesian NE (BNE).
- In Section 5 we further extend our model to a more realistic learning-based setting, where agents only observe their own type, but do not have any knowledge of the prior distribution over types and game parameters. Instead, they learn from repeated interaction, where they iteratively decide their strategy conditioned on their realized type. This naturally models learning to acquire resources in competition, given contextual information. We establish sufficient conditions under which agents engaging in simultaneous no-regret learning either convergence to a Bayesian coarse-correlated equilibrium (BCCE) on average over rounds, or to the BNE in the final round.
- For each setting, we provide simulations showcasing the respective algorithms.

## 1.2 RELATED WORK

The most relevant related work to our setting is the recent line of work on optimal position building under competition in Chriss (2024b;c;a; 2025); Kearns & Shi (2025) that we discussed above. Our work can be seen as a generalization of these, for both financial and non-financial applications. We provide a detailed discussion of how our setting relates to and subsumes the settings in these works in Appendix A.2. To the best of our knowledge, there is no existing work that have considered strategic aspects of resource application in on-financial settings, although researchers have noted noted the relevance of such considerations, *e.g.* in compute markets Shastri & Irwin (2018).

More broadly, our model captures standard notions of *market impact* in finance, of which there is a large literature (see *e.g.* Webster (2023); Li et al. (2024) and citations therein for a recent detailed overview). These works broadly consider how prices change in response to trading (both theoretically and empirically from real markets). In this literature, market impact is often decomposed into permanent and temporary impact (Almgren & Chriss, 2000; Bacry et al., 2015; Moro et al., 2009), which is the same approach that we take. There are also some more flexible models such as the *propagator model* (Bouchaud et al., 2003; Gatheral & Schied, 2013; Obizhaeva & Wang, 2013) that allow for transient impacts in between these extremes; we do not consider these, but such extensions would be an interesting direction for future work.

Our work also relates to the literature on learning in games more broadly. Of particular note, our setting in Section 5 is very similar in spirit to the setting of Hartline et al. (2015), who provide a general framework for no-regret learning in repeated Bayesian games. Although our model is slightly outside their framework (as it allows continuous market types), and has some specific structure that we can leverage (strongly monotone) our Theorem 5 is very motivated by their theory.

Our strategic setup is conceptually related to resource allocation – or more generally, congestion games – where agents share resources, and the cost of any resource depends on its demand (Rosenthal, 1973). In contrast to this setting, our work studies behavior under time-varying prices, which endogenously adjust based on supply and demand. Lastly, while we study competitive resource acquisition in a market setting, a non-market variant has long been studied in the context of *fair division* (Moulin, 2004) for both divisible and indivisible goods. Recent literature here has extended this problem to an online setting (see Aleksandrov & Walsh (2020) for a survey), and often incorporates learning and predictions (Banerjee et al., 2023), spiritually motivating our model in Section 5.

## 2 MODEL

**Preliminaries:** We consider a market consisting of $n$ strategic agents looking to trade (buy/sell) some costly, divisible resource (stock, bond, compute time, *etc.*) over a period of $T$ rounds. In the simplest setting, each strategic agent $i$'s action is a $T$-dimensional vector $\boldsymbol{h}_i$, where $h_{i,t}$ denotes how much they purchased at time $t \in [T]$. Note that $\boldsymbol{h}_i$ is a signed vector, and we conventionally denote positive values as buying and negative values as selling throughout. We assume that each agent has some set of convex constraints on their allowable actions, which we represent by a feasible set of trajectories $G_i \subseteq \mathbb{R}^T$. For example, if agent $i$ wants to procure at most $V_i$ equity shares without short selling or over-buying, they could represent their constraints via $G_i = \{\boldsymbol{h}_i : 0 \leq \sum_{l=1}^{t} h_{i,l} \leq V_i \ \forall t \in [T]\}$. We also assume that each agent has some idiosyncratic utility function on their strategy, which can capture (1) the utility of their final position and/or any preferences on their acquisition schedule. For an agent $i$, we represent this via a concave function $f_i : \mathbb{R}^T \to \mathbb{R}$ (with the same units as price)[1]. For example, if agent $i$ wishes to impose a concave value function $\phi_i$ on their final position and penalize selling, their idiosyncratic utility could be $f_i(\boldsymbol{h}_i) = \phi_i(\mathbf{1}^\top \boldsymbol{h}_i) - \zeta \sum_{t=1}^{T} |h_{i,t}| \mathbb{I}\{h_{i,t} < 0\}$ with $\zeta \geq 0$, which is clearly concave. If an agent has a private valuation $r_i$ for the asset, they could use $\phi_i(x) = r_i x$. In settings like compute markets or optimal trade execution, where the agents' goal is to acquire a fixed target position as cheaply as possible, one can set $f_i = 0$ and include a hard constraint on $\mathbf{1}^\top \boldsymbol{h}_i$ in $G_i$. Lastly, and inspired by the seminal work of Kyle (1985), we allow the market to contain a non-strategic (possibly random) *exogenous* agent, which captures all non-strategic trade flow. Following our convention, the exogenous agent's action is given by a signed vector $\boldsymbol{s} \in \mathbb{R}^T$ where positive values indicate buying.

**Price Model:** Core to understanding how agents strategically interact in acquiring costly resources is how their demand/supply levels influence resource prices. We assume the following dynamic model for determining resource prices from agents' trading schedules:

**Assumption 1** (Price Dynamics)**.** *All agents pay the same price $p_t$ for each share of the resource at time $t$, where $p_t$ is determined from the total trading schedule of all agents up to and including time $t$ according to the following equations:*

$$p_t = p_t^w + \beta \left( \sum_{i=1}^{n} h_{i,t} + s_t \right); \quad p_t^w = p_{t-1}^w + \alpha \left( \sum_{i=1}^{n} h_{i,t} + s_t \right), \quad (1)$$

*where $p_0 = p_0^w$ is the initial price, and $\alpha, \beta \geq 0$ are some problem parameters.*

The dynamic process for $p_t^w$ can be seen as a discretization of the Walrasian price dynamics from general equilibrium theory, which posits that prices evolve from an imbalance of supply and demand: $dp_t = \alpha(\text{demand}_t - \text{supply}_t)dt$, where $\alpha$ is a sensitivity factor (Walker, 1987). The additional $\beta \left( \sum_{i=1}^{n} h_{i,t} + s_t \right)$ term in $p_t$ accounts for additional costs imposed by market makers who provide

---

[1]This allows general concave utility on final position, which corresponds to diminishing marginal utility and is a natural restriction in economics and game theory. See (Mas-Colell et al., 1995; Debreu, 1959).

liquidity to balance supply and demand, causing temporary deviations from the Walrasian price process ($\beta$ controls the strengths of this impact). This also maps to how prices are modeled within the theory of optimal trade execution, with $\alpha$ and $\beta$ corresponding to permanent and temporary impact coefficients, respectively (Almgren & Chriss, 2000). We discuss in detail in Appendix A.1.

**Game Payoff Structure:** We model the total utility for each agent according to their personal utility $f_i$, minus the total cost they incur buying and selling. Formally:

**Definition 1** (Game Payoffs). *Let the price parameters $p_0$, $\alpha$, and $\beta$, and exogenous action $\boldsymbol{s}$, be given. In addition, let $\boldsymbol{h}_{-i}$ denote the trading schedules of all strategic agents other than $i$, and $\boldsymbol{p}(\boldsymbol{h}_i, \boldsymbol{h}_{-i}, p_0, \boldsymbol{\lambda}) \in \mathbb{R}^T$ denote the sequence of prices under Assumption 1 for $\boldsymbol{h}_i$, $\boldsymbol{h}_{-i}$, and $\boldsymbol{\lambda} = (f_1, \ldots, f_n, p_0, \alpha, \beta, \boldsymbol{s})$. Then, the overall utility for agent $i$ is:*

$$u_i(\boldsymbol{h}_i; \boldsymbol{h}_{-i}, \boldsymbol{\lambda}) = f_i(\boldsymbol{h}_i) - \boldsymbol{p}(\boldsymbol{h}_i, \boldsymbol{h}_{-i}, \boldsymbol{\lambda})^\top \boldsymbol{h}_i . \tag{2}$$

We note that these payoff (utility) functions, along with the constraint sets $G_i$ for each agent $i$, fully define the strategic game for fixed game parameters. For $f_i = 0$, these payoff functions can be shown as equivalent to the continuous time linear/quadratic cost-functions considered in Chriss (2024b;c); we provide details of this in Appendix A.2.

**Bayesian Perspectives:** In addition to considering fixed instances of the strategic resource acquisition game, as defined above (which we analyze in detail in Section 3,) we also consider a Bayesian game extension, where there is uncertainty in the game parameters, and each agent only observes some private information that may be correlated with these. This is needed for the partial-information settings we consider in Section 4 and Section 5.

**Definition 2** (Bayesian Game). *The Bayesian version of our game is formalized by the following:*

1. ***Market Type:*** *Define the market type as $\boldsymbol{\lambda} = (f_1, \ldots, f_n, p_0, \alpha, \beta, \boldsymbol{s})$, as in Definition 1.*

2. ***Agent Type:*** *Let $\theta_i$ denote the* type *of agent $i$, which is the set of all information known to them before acting. $\Theta_i$ denotes the possible type space for agent $i$, where $|\Theta_i| = k_i < \infty$.*

3. ***Agent Constraints:*** *Player types $\theta_i$ fully determines their constraints, which we denote $G_i(\theta_i)$. Let $\mathcal{H}_i = \{\boldsymbol{h} : \boldsymbol{h}(\theta) \in G_i(\theta) \ \forall \theta \in \Theta_i\}$ denote the set of feasible strategies for each agent $i$.*

4. ***Distribution over game instances:*** *Let $\mathcal{I} = (\theta_1, \ldots, \theta_n, \boldsymbol{\lambda})$ denote a full game instance. Then there exists a joint probability distribution $P(\theta_1, \ldots, \theta_n, \boldsymbol{\lambda})$ over the components of $\mathcal{I}$.*

5. ***Agent Strategy:*** *Outside of the complete information setting, the strategy for agent $i$ is a function $\boldsymbol{h}_i : \Theta_i \to \mathbb{R}^T$, which determines how they would behave under each possible type.*

6. ***Agent Utility:*** *Each agent $i$ defines their utility given strategy $\boldsymbol{h}_i$ and opponent strategies $\boldsymbol{h}_{-i}$ as the expected utility over $\mathcal{I} \sim P$, which is given by $\mathbb{E}_{\theta_i, \theta_{-i}, \boldsymbol{\lambda} \sim P}[u_i(\boldsymbol{h}_i(\theta_i); \boldsymbol{h}_{-i}(\theta_{-i}), \boldsymbol{\lambda})]$.*

It is trivial to verify that the goal of all agents maximizing their overall expected utility is equivalent to each agent maximizing their conditional expected utility given their private information/type: $\mathbb{E}_{\theta_{-i}, \boldsymbol{\lambda} \sim P|\theta_i}[u_i(\boldsymbol{h}_i(\theta_i); \boldsymbol{h}_{-i}(\theta_{-i}), \boldsymbol{\lambda})]$. Moreover, the agent types $\theta_i$ can specify/influence the agent's constraints $G_i(\theta_i)$. As for the market parameters $\boldsymbol{\lambda}$, the type may either completely determine, be partially correlated with, or completely uninformative of any given component in $\boldsymbol{\lambda}$. In particular, we do *not* generally assume that $\theta_i$ specifies $f_i$, since we allow for idiosyncratic utilities to depend on uncertain market valuations. While no assumptions are made about the types themselves, in practice, they can be perceived as some feature set the the respective agent uses to understand the market decide their trading schedule. Finally, we note that the types for different agents may have very different strengths of correlation with other agent types and components of $\boldsymbol{\lambda}$. This naturally captured information asymmetry that is common in most market.

## 3 COMPLETE INFORMATION SETTING

In the complete information setting, the market type $\boldsymbol{\lambda}$ and agent constraints $G_1, \ldots, G_n$ are observed by all $n$ strategic agents. Therefore, it is unnecessary to consider agent types or the distri-

bution $P$, so we instead just analyze an arbitrary fixed game instance $\mathcal{I}$ defined by $G_1, \ldots, G_n, \boldsymbol{\lambda}$. Recall from Section 2 that for fixed game instances, we let $\boldsymbol{h}_i$ denote a fixed trading schedule (*i.e.* $\boldsymbol{h}_i \in \mathbb{R}^T$) rather than a function of agent types.

Complete information games are routinely studied in game theoretic models, since: (1) they provide clearer intuitions on the strategic dynamics of the problem; and (2) they are the basis for common solution concepts, namely Nash Equilibria (NE) and Price of Anarchy (PoA) (Roughgarden, 2010):

**Definition 3.** *For a complete information instance $\mathcal{I}$, the strategies $(\boldsymbol{h}_1^{eq}, \ldots, \boldsymbol{h}_n^{eq})$ are a* Pure Nash Equilibrium *if and only if: for all buyers $i$ and any strategy $\boldsymbol{h}_i'$: $u_i(\boldsymbol{h}_i^{eq}; \boldsymbol{h}_{-i}^{eq}, \boldsymbol{\lambda}) \geq u_i(\boldsymbol{h}_i'; \boldsymbol{h}_{-i}^{eq}, \boldsymbol{\lambda})$.*

**Definition 4.** *For a complete information instance $\mathcal{I}$, let $NE(\mathcal{I})$ denote the set of all NE strategies, and let $welf(\boldsymbol{h}_1, \ldots, \boldsymbol{h}_n, \boldsymbol{\lambda}) = \sum_{i=1}^n u_i(\boldsymbol{h}_1; \boldsymbol{h}_{-i}, \boldsymbol{\lambda})$ denote the welfare function. The* Price of Anarchy *ratio is then defined as: $\sup_{\boldsymbol{h}_1 \in G_1, \ldots, \boldsymbol{h}_n \in G_n, \boldsymbol{h}^{eq} \in NE(\mathcal{I})} \frac{welf(\boldsymbol{h}_1, \ldots, \boldsymbol{h}_n, \boldsymbol{\lambda})}{welf(\boldsymbol{h}_1^{eq}, \ldots, \boldsymbol{h}_n^{eq}, \boldsymbol{\lambda})}$.*

Informally, the NE are the set of strategies such that no agent has any incentive to unilaterally deviate, and the PoA characterizes the ratio of the best obtainable welfare if agents were to cooperate to the worst obtainable welfare obtainable from NE. We begin with an explicit expression of agent utility, which we show to be strictly concave, allowing us to characterize the above notions. We provide details and derivation of the lemma in Appendix B.

**Lemma 1.** *By unrolling the auto-regressive price definition in Equation* (1)*, the utility of agent $i$ in instance $\mathcal{I}$ for joint strategy $(\boldsymbol{h}_1, \ldots, \boldsymbol{h}_n)$ is strictly concave in their strategy, and is given by:*

$$u_i(\boldsymbol{h}_i; \boldsymbol{h}_{-i}, \boldsymbol{\lambda}) = f_i(\boldsymbol{h}_i) - \frac{1}{2}\boldsymbol{h}_i^T Q \boldsymbol{h}_i - \sum_{j \neq i}(A\boldsymbol{h}_j)^T \boldsymbol{h}_i - \boldsymbol{s}^T A \boldsymbol{h}_i - p_0(\mathbf{1}^T \boldsymbol{h}_i),$$

*where $Q$ and $A$ are $n \times n$ matrices defined in terms of $\alpha$ and $\beta$, and $Q$ is symmetric and strictly PD.*

Our definitions and statements so far have been framed with respect to pure (deterministic) strategies. In general, strategic agents may use mixed (randomized) strategies, which begs the following question: could mixed strategies appear in NE? In the following lemma (proof in Appendix B), we answer this question in the negative by characterizing an agent's best response – their optimal strategy for a fixed set of others' strategies – as being pure.

**Lemma 2.** *For any fixed game instance $\mathcal{I}$, the best response of any agent $i$ is always unique and deterministic, even when others are playing some mixed (possibly correlated) strategies.*

Next, we turn to the first of our two central results in this section, regarding characterization and computation of the NE, which we address via the following theorem. The proof is technical and stems from casting the equilibrium conditions as a variational inequality, and then proving that the operator for this variational inequality is strongly monotone. This property immediately implies the uniqueness of the NE, and gives us an efficient gradient-based algorithm for computing it. We formalize these results below, with the proof and full algorithm details given in Appendix B.

**Theorem 1.** *For every fixed game instance $\mathcal{I}$, there is a unique pure NE. In addition, the extra-gradient algorithm (Korpelevich, 1976) converges linearly to this equilibrum.*

Interestingly, strong monotonicity of this game's corresponding VI operator implies that in this complete information setting, any coarse correlated equilibrium (CCE) must also be a Nash. Finally, we turn to the question of characterizing the PoA, which we show in general is unbounded. The proof is based on an explicit counterexample, whose intuition is as follows: consider two agents with differing valuation on their final position, where one agent wants to buy and other wants to sell. If they coordinate, they can can provide liquidity to each other, which eliminates trading frictions and allows them to trade a large quantity and obtain high welfare. However, if they agreed to do this, each would have an incentive to cheat by providing less liquidity to the other; by doing so, the trade imbalance would move the price favorably for them, which they could profit from. Because of this, the NE involves both agents trading almost nothing, and achieving very low welfare. We provide full details in Appendix B.

**Theorem 2.** *For any constants $\alpha, \beta, T$, and any $\xi > 0$, there exists an instance $\mathcal{I}$ of the complete information game with PoA ratio at least $\xi$. Therefore, the PoA is unbounded.*

The result above is a worst-case scenario and relies on traders strategically trading in two different directions – one buying and another selling. We prove below (proof in Appendix B), however, that in a subsets of games where all traders are buying to build a positive position, the PoA is always bounded. Such scenarios commonly occur in markets; after an earnings call, for instance, traders may systematically move their positions in a positive direction if earnings were above expectations. In compute markets, agents are looking to cheaply procure a given amount of resources.

**Theorem 3.** *Let $\gamma = \frac{\alpha}{\alpha+\beta}$. Then for position building game instances with the following properties, the Price of Anarchy is upper bounded by $O(n^2 T^2 \gamma^2)$:*

- *All agents only buy. That is: $h_{i,t} \geq 0, \forall i, t$ and $s_t \geq 0$.*
- *Each agent $i$'s goal is to build a position $V_i$ – for all $i$, $\sum_t h_{i,t} = V_i$ is a constraint.*
- *Agent utility $f_i$ depends only on the final position – $f_i(\mathbf{1}^T \boldsymbol{h}_i)$.*

## 3.1 EMPIRICAL SIMULATIONS OF EQUILIBRIUM

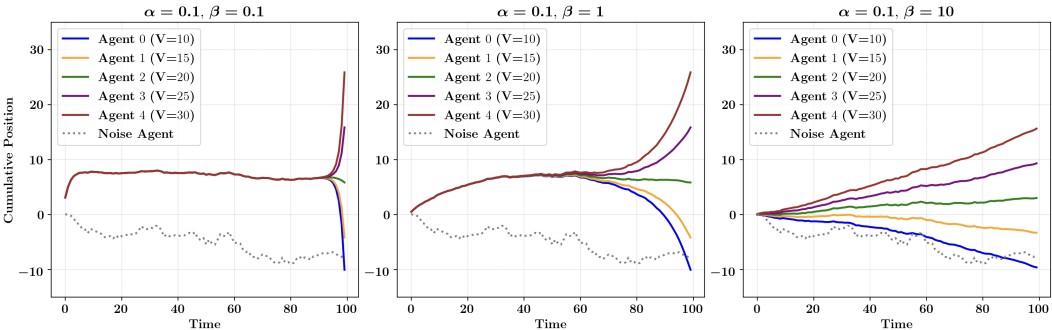

Figure 1: Cumulative position over time for agents in NE, for fixed $\alpha$ and varying $\beta$. The initial price is $p_0 = 2$, the reserve prices are $(4, 5, 6, 7, 8)$ and the constraint values are $V = (10, 15, 20, 25, 30)$.

To further understand the agents' behavior in equilibrium, we empirically compute the NE for a simple, yet practically motivated setting. Consider 5 strategic agents with a linear final position utility $f_i(\boldsymbol{h}_i) = r_i \sum_t h_{i,t}$ for some reserve price $r_i$, and constraints $-V_i \leq \mathbf{1}^T \boldsymbol{h}_i \leq V_i$ (i.e. final position must be in range $[-V, V]$). We randomly sample the actions $s_t$ of the exogenous agent using i.i.d. zero-mean random variables. In Figure 1 we plot the cumulative positions $(\sum_{l=1}^t h_{i,l})$ of each agent $i$, for three different sets of problem parameters, where we fix $\alpha = 0.1$ and vary $\beta \in \{0.1, 1, 10\}$. We observe that, as $\beta$ increases, the total volume traded decreases, which is unsurprising since $\beta$ corresponds to trading frictions. More interestingly, we note a phase transition. For small $\beta$, the NE approaches a pair of block trades – a first at time 0 where all agents purchase an identical quantity of the resource, and a second at time $T$ where all agents buy or sell to reach some final position. For large $\beta$, the NE approaches all agents trading at a constant rate, with some interpolation between these for intermediate $\beta$.

In Appendix F, we augment our synthetic experiments with experiment using publicly available level 1 limit order book-style data for currency exchange markets. Using this data, we are able to estimate supply and demand imbalances and price movements on a tick-by-tick basis, which allows us to run regressions to estimate the market parameter $\alpha, \beta$ and model the exogenous actor $s_t$ as the observed difference between demand and supply. The high-level observations made in the synthetic setting above also emerge in these real-world experiments. We also include, in Appendix G, a detailed analysis on the implications of behaving strategically versus not. This is important, because standard thinking in optimal execution (which classically does not consider competition) is to execute according to Volume-Weighted Average Price (VWAP), which under our model corresponds to trading at a constant rate. This leads to new insights that may be of independent interest. Overall, we see these real-world experimental validations as impactful since, to the best of our knowledge, all prior works who have studied optimal resource acquisition in similar settings have relied on proprietary or synthetic data in stylized environments.

## 4 Partial Information Setting with Common Knowledge of Prior

We now consider the partial-information setting: all agents have common knowledge of the joint distribution $P$ of agent and market types, but each agent only observes their own private information via their type $\theta_i$. For this setting, we only consider characterization and computation of equilibria, since the unbounded PoA for complete information settings automatically carries over here. As discussed in Section 2, we think of the strategy of each agent as an *ex-ante* mapping from each possible type $\theta_i$ to a feasible trading schedule in $G_i(\theta_i)$, which we denote by the function $\boldsymbol{h}_i \in \mathcal{H}_i$. Game play in this Bayesian setting operates via the following sequence of events: (1) each agent decides their *ex-ante* strategy $\boldsymbol{h}_i$; (2) a game instance $\mathcal{I} \sim P$ is sampled and the corresponding type information $\theta_i$ is privately revealed to each agent; and (3) each agent executes their strategy $\boldsymbol{h}_i(\theta_i)$. This setting can be formally studied within the Bayesian game theory framework, using the standard equilibrium notion as follows:

**Definition 5** (Bayesian Nash Equilibrium). *For a Bayesian instance, the strategies $(\boldsymbol{h}_1^{eq}, \ldots, \boldsymbol{h}_n^{eq})$ are in a Bayesian Nash Equilibrium (BNE) if for all agents $i$, all $\boldsymbol{h}_i' \in \mathcal{H}_i$, and all $\theta_i \in \Theta_i$, we have:*
$$\mathbb{E}_{\theta_{-i}, \boldsymbol{\lambda} \sim P|\theta_i}[u_i(\boldsymbol{h}_i^{eq}(\theta_i); \boldsymbol{h}_{-i}^{eq}(\theta_{-i}), \boldsymbol{\lambda})] \geq \mathbb{E}_{\theta_{-i}, \boldsymbol{\lambda} \sim P|\theta_i}[u_i(\boldsymbol{h}_i'(\theta_i); \boldsymbol{h}_{-i}^{eq}(\theta_{-i}), \boldsymbol{\lambda})].$$

As in Section 3, this equilibrium is defined in terms of pure strategies (deterministic trading schedule for each type). The following lemma, which generalizes Lemma 2 to the Bayesian game setting, ensures that this restriction does not restrict the BNE (proof details in Appendix C).

**Lemma 3.** *For any Bayesian instance, the best response of any agent $i$ is always unique and deterministic (meaning trading schedule $\boldsymbol{h}_i(\theta_i)$ for every type $\theta_i$ is deterministic), even if others are playing some mixed (possibly correlated) set of strategies for each type.*

We now present the central result in this setting. Theorem 4 generalizes the result of Theorem 1 to the Bayesian game setting, namely that there is a unique and efficiently computable equilibrium. Similar to the complete information setting, this follows by casting the equilibrium problem as a variational inequality, which we show is strongly monotone; this implies uniqueness of the BNE, and gives an efficient gradient based algorithm for computing it. We provide full details in Appendix C.

**Theorem 4.** *For every Bayesian game instance (given by distribution $P$), there is a unique pure BNE, and the extra-gradient algorithm (Korpelevich, 1976) converges linearly to this BNE.*

### 4.1 Empirical Simulations of Equilibrium

We simulate the BNE for a similar scenario as in Section 3.1, where agents' have linear utility and inequality constraints on their final positions. We use 2 agents here, each with 3 possible types, and the type $\theta_i$ determines the final position bounds $V_i$ and expected reserve price $r_i$ for each agent. We set $\boldsymbol{s} = 0$ in this simulation. As before, we fix $\alpha = 0.1$, and vary $\beta$; in each case, $\beta$ is continuously distributed within some bounded range. We provide full details in Appendix C.

We see similar phase transition dynamics as we vary $\beta$ from small to large as we observed in the complete information setting. However, in this case, since the cumulative positions in each agent's strategy are type-dependent, the strategies do not consistently overlap during early time steps. That is, the partial information induces a richer, type-dependent dynamic. However, we do observe overlap in trade execution between some pairs of types for the two agents, which is interesting and warrants exploration in future work.

## 5 Partial Information Setting with no Prior via Online Learning

We now move to a more realistic setting, where agents neither have complete information, nor any *a priori* knowledge of the prior $P$. Instead, in this setting agents must learn via interaction. Specifically, we consider a mode of repeated game play that occurs over $R \in \mathbb{N}^+$ rounds, where in each round $r$ the game play follows the sequence of events for Bayesian game play described in Section 4, and the *ex-ante* strategy $\boldsymbol{h}_i^r$ that each agent $i$ selects in round $r$ is chosen adaptive to their feedback following rounds 1 through $r - 1$. The feedback that each agent observes after each round is formalized by the following assumption:

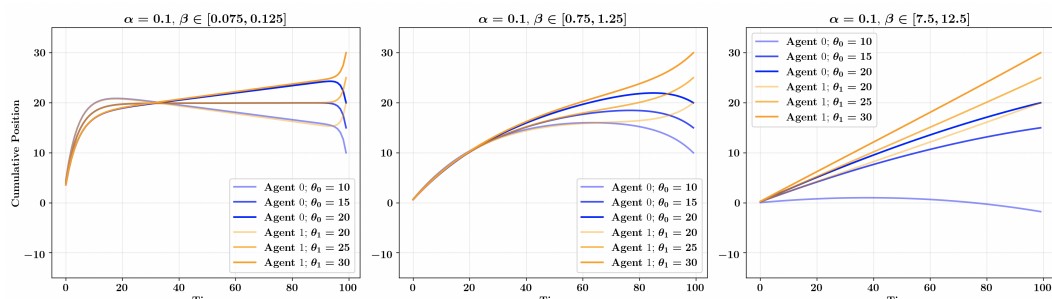

Figure 2: Cumulative position over time for agents under the BNE. Types are distributed uniformly and correspond to the constraint $V$. The type conditioned expected reserves are $(3, 5, 7)$ for agent 1, and $(6, 8, 10)$ for agent 2. $p_0 = 0, \alpha = 0.1$ and the conditional $\beta$ distribution lies in the given range.

**Assumption 2** (End of Round Feedback). *Let $\boldsymbol{h}_i^r$ and $\theta_i^r$ denote the* ex-ante *strategy and sampled type for each agent $i$ in round $r$, and let $\boldsymbol{\lambda}^r$ denote the sampled market type. Then, at the end of round $r$, each agent observes the cost function $c_i^r : \mathcal{H}_i \to \mathbb{R}$ as feedback, given by:* [2]

$$c_i^r(\boldsymbol{h}_i) = -u_i(\boldsymbol{h}_i(\theta_i^r); \boldsymbol{h}_{-i}^r(\theta_{-i}^r), \boldsymbol{\lambda}^r).$$

This feedback is the cost (negative utility) that the agent would have received if they had committed to a different strategy in that round, letting the strategies of all other agents and game instance be fixed. This is feedback is extremely realistic since trades are public in almost all markets and thus agents can observe others' actions and then determine what their counterfactual cost would be if they had acted differently. In practice, this counterfactual cost is inferred by performing some regressions on observed market data to compute the sequence of aggregate outside demands $\sum_{j \neq i} h_{j,t} + s_t$ along with $\alpha$ and $\beta$. Appendix A.3 discusses this process in more detail.

Importantly, many of our algorithms in this setting assume access to gradient $\nabla_{\boldsymbol{h}_i^r(\theta_i^r)} c_i^r(\boldsymbol{h}_i^r)$. This is *strictly weaker* than the counterfactual cost feedback of Assumption 2, since the gradient of this cost function is itself a valid stochastic gradient. Further, our results only require the feedback to be correct in expectation; thus, our theoretical insights are robust to any noise in this feedback, which is to be expected in real market settings.

To obtain concrete guarantees, we impose some boundedness regularity conditions, which are restrictions on the constraint sets, distribution over exogenous actions, and idiosyncratic utilities.

**Assumption 3** (Boundedness). *We assume that there exists some finite, fixed values $B$, $S$, $U$, and $U'$, such that for all agents $i \in [n]$ and strategies $\boldsymbol{h}_i \in \mathcal{H}_i$, the following bounds hold almost surely: (1) $\|\boldsymbol{h}_i(\theta_i)\|_2 \leq B$; (2) $\|s\|_2 \leq S$; (3) $|f_i(\boldsymbol{h}_i(\theta_i))| \leq U$; and (4) $\|\nabla f_i(\boldsymbol{h}_i(\theta_i))\|_2 \leq U'$.*

Next, we define an (unobserved) *population* loss for each agent $i$ in any given round $r$, as follows:

**Definition 6** (Population Loss). *Let $\boldsymbol{h}_i^r$ denote the* ex-ante *strategies of agent $i$ in round $r$ for all $i$. Then, the expected loss for each agent $i$ in round $r$ is a function $\ell_i^r : \mathcal{H}_i \to \mathbb{R}$ given by:*

$$\ell_i^r(\boldsymbol{h}_i) = \mathbb{E}_{\theta_i, \theta_{-i}, \boldsymbol{\lambda} \sim P}[-u_i(\boldsymbol{h}_i(\theta_i); \boldsymbol{h}_i^r(\theta_{-i}), \boldsymbol{\lambda})].$$

In other words, $\ell_i^r(\boldsymbol{h}_i) = \mathbb{E}_{\mathcal{I} \sim P}[c_i^r(\boldsymbol{h}_i)]$ for all $\boldsymbol{h}_i \in \mathcal{H}_i$; likewise for its derivatives. Therefore, the observed losses $c_i^r$ or their gradients can be interpreted as unbiased stochastic estimates of the population losses $\ell_i^r$ or their gradients respectively. It is easy to see that if $\boldsymbol{h}_i^r$ minimizes $\ell_i^r$ for every agent $i$ simultaneously in some round $r$, then the agents are in BNE. Therefore, this intuitively suggests that we could apply existing theory and algorithms for online convex optimization with stochastic feedback to establish convergence to equilibrium. We formalizing this intuition and begin by defining some additional central concepts:

---

[2] $c_i^r$ could be written in a slightly simpler way with domain $G_i(\theta_i^r)$ rather than $\mathcal{H}_i$; the latter, however, is more convenient as it ensures that $c_i^r$ is defined on the same domain for all $r \in [R]$.

**Definition 7** (Regret). *Fix a agent $i \in [n]$ and a sequence of loss functions $\chi_i^r : \mathcal{H}_i \to \mathbb{R}$ for $r \in [R]$. For any sequence of strategies $\boldsymbol{h}_i^r$ for $r \in [R]$, we define their (average)* regret *as*

$$\text{regret}(\boldsymbol{h}_i^1, \ldots, \boldsymbol{h}_i^R; \chi_i^1, \ldots, \chi_i^R) = \frac{1}{R} \sum_{r=1}^{R} \chi_i^r(\boldsymbol{h}_i^r) - \min_{\boldsymbol{h}_i \in \mathcal{H}_i} \frac{1}{R} \sum_{r=1}^{R} \chi_i^r(\boldsymbol{h}_i)$$

**Definition 8** ($\epsilon$-BCCE). *Let $\sigma \in \Delta(\mathcal{H}_1 \times ... \times \mathcal{H}_n)$ be a joint distribution over strategy profiles. Then, $\sigma$ is an $\epsilon$-approximate Bayesian coarse correlated equilibrium ($\epsilon$-BCCE) if and only if for all agents $i$, $\theta_i \in \Theta_i$, and $\boldsymbol{h}_i' \in \mathcal{H}_i$:*

$$\mathbb{E}_{\boldsymbol{h}_i, \boldsymbol{h}_{-i} \sim \sigma, \theta_{-i}, \boldsymbol{\lambda} \sim P|\theta_i}[u_i(\boldsymbol{h}_i(\theta_i), \boldsymbol{h}_{-i}(\theta_{-i}), \boldsymbol{\lambda})] \geq \mathbb{E}_{\boldsymbol{h}_{-i} \sim \sigma, \theta_{-i}, \boldsymbol{\lambda} \sim P|\theta_i}[u_i(\boldsymbol{h}_i'(\theta_i), \boldsymbol{h}_{-i}(\theta_{-i}), \boldsymbol{\lambda})] - \epsilon$$

Regret measures how much we could decrease our average loss by if, retrospectively, we swapped from the actual sequence of chosen strategies to some fixed alternative strategy. *No-regret algorithms* are well studied in the online learning literature; these are algorithms that ensure that the average regret converges to zero under arbitrarily (possibly adversarially) chosen loss functions. On the other hand, approximate BCCE is a weaker equilibrium notion than BNE, in two respects: (1) it allows for correlation between the strategies in equilibrium; and (2) it allows for $\epsilon$-sub-optimality of the chosen strategies (in conditional expectation given $\theta_i$).

Although no-regret algorithms and BCCE may seem like unrelated ideas at first, they are deeply connected since multiple agents simultaneously following no-regret dynamics with a fixed game objective will induce an approximate coarse correlated equilibrium (CCE) in the game. Although we are considering Bayesian games and BCCE rather than CCE, similar reasoning gives us the following theorem (proof in Appendix D).

**Theorem 5.** *Suppose every agent $i \in [n]$ selects their strategy $\boldsymbol{h}_i^r$ at each round $r$ via some online algorithm $\text{Alg}_i$, with the following properties: (1) $\text{Alg}_i$ selects strategy $\boldsymbol{h}_i^r$ at round $i$ only using unbiased stochastic cost-function observations $\tilde{\chi_i^r}$ for some true sequence of cost functions $\chi_i^r \in \mathcal{C}_i$, where $\mathcal{C}_i$ is a set containing all population losses $\ell_i^r$ almost surely; (2) it ensures that $\text{regret}(\boldsymbol{h}_i^1, \ldots, \boldsymbol{h}_i^R; \chi_i^1, \ldots, \chi_i^R) \leq \epsilon_i(R)$ for some $\epsilon_i(R)$ that is independent of the cost functions $\chi_i^r \in \mathcal{C}_i$, which could be adversarially chosen. In addition, let $\sigma^R \in \Delta(\mathcal{H}_1 \times \ldots \times \mathcal{H}_n)$ denote the uniform distribution over $(\boldsymbol{h}_1^r, \ldots, \boldsymbol{h}_n^r)$ across rounds. Then, $\sigma^R$ is an $\epsilon$-BCCE, for some $\epsilon$ that is bounded by $\frac{\epsilon_i(R)}{\Pr(\theta_i)}$ for all $i \in [n], \theta_i \in \Theta_i$.*

We make a few comments on this theorem. First, we note that it is very general, and establishes convergence to equilbria for agents who simultaneously engage in no-regret learning using *any* stochastic-feedback no regret-learning algorithm with thier observed costs $c_i^r$, and the algorithms could be different for each agent. Second, the restriction of losses in the theorem statements to some sets $\mathcal{C}_i$ is necessary since adversarial no-regret will generally be impossible without some bounds on the allowed stochastic/true losses. In general, the constants involved in the regrets $\epsilon_i(R)$ obtainable by a given algorithm may depend on $\mathcal{C}_i$, the choice of which in practice will depend on the bounds we can place on $c_i^r$ and $\ell_i^r$. Finally, the dependence of the result on the worst-case $1/\Pr(\theta_i)$ arises from bounding the conditional sub-optimality in the definition of BCCE uniformly over *all* $i$ and $\theta_i$. For any given $i, \theta_i$, we can bound this sub-optimality by $\epsilon_i(R)/\Pr(\theta_i)$, so the presence of rarely-occurring types don't cause sub-optimality conditional on common types to suffer, and the *average* sub-optimality over all types is bounded by $\epsilon_i(R)$ (see proof for details).

Although the guarantees from Theorem 5 are slightly weak in that they only ensure approximate convergence to a BCCE for the average-iterate strategy, not for the final iterate $\boldsymbol{h}_1^R, \ldots, \boldsymbol{h}_n^R$, we can do better if all agents apply a *doubly optimal* algorithm (Jordan et al., 2024), which is an algorithm that ensures no-regret, as well as last-iterate convergence to a NE if applied by all agents in a strongly monotone game with stochastic gradient feedback. The specific algorithm proposed by Jordan et al. (2024) that obtains this property is online gradient descent (OGD) (Zinkevich, 2003) with a specific stochastic scheme for reducing learning rates. We provide details of this algorithm (Algorithm 2) and its theoretical guarantees in Appendix D.2. The following theorem establishes that, if all agents independently follow this algorithm, their joint strategy profile converges to the BNE.

**Theorem 6.** *Suppose all agents use Algorithm 2 to decide their strategy in each round $r$. Then, letting $k = \max_{i \in [n]} |\Theta_i|$, the final iterate strategies $\boldsymbol{h}_1^R, \ldots, \boldsymbol{h}_n^R$ are an $\epsilon$-approximate BNE, for*

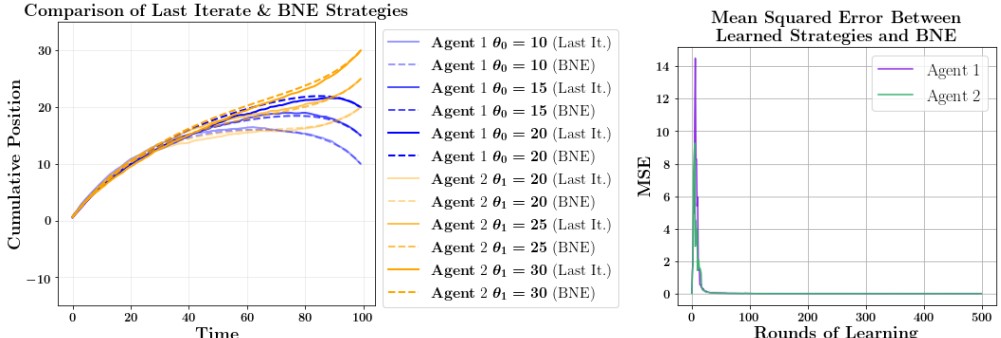

Figure 3: Comparison of Algorithm 2 (over 500 rounds) to exact BNE strategies: (left) we plot the last-iterate strategies returned by Algorithm 2 (solid lines) along with the true BNE (dashed lines) for all agents and types; and (right) we show the convergence in mean-squared error between the strategies from Algorithm 2 and the BNE over the 500 rounds.

*some $\epsilon$ that satisfies the following in expectation over the algorithm's randomness:*

$$\mathbb{E}[\epsilon] = O\left(\frac{\mathrm{poly}(n, T, k, \alpha, \beta, p_0, B, S, U')}{\min_{i \in [n], \theta_i \in \Theta_i} \Pr(\theta_i)} \cdot \frac{\log^{3/2}(R)}{\sqrt{R}}\right)$$

Even though this result only establishes convergence to BNE if all agents follow it, by the *doubly optimal* property discussed above it is no-regret. Thus, the algorithm is also robust to possibly adversarial environments. Compared with the extra-gradient algorithm (which we previously showed can efficiently compute the BNE), the benefits of Algorithm 2 are two-fold: (1) each agent's learning procedure is *prior-independent*, since to update their strategies they only need gradient information about their realized cost; and (2) the procedure is fully decentralized, since agents can run their learning algorithms independently using only the information privately revealed to them. As with Theorem 5, our result depends on the worst-case $1/\Pr(\theta_i)$, but the same comments we made there apply about how this is not a major theoretical limitation. We finally note that the above concept of double optimality is very new, and it is possible that other decentralized algorithms could obtain similar guarantees, but this is a question for future research.

## 5.1 SIMULATIONS

We conclude with an empirical investigation on convergence to equilibrium in actual implementation, simulating repeated play as described in Section 5, where all agents follow Algorithm 2. We use the same Bayesian game instance as in Section 4.1, with 2 agents, each having 3 possible types. We provide full details of this simulation setting in Appendix D.4.

We show the results of this simulation in Figure 3. On the left we directly compare the final iterate strategies from online learning with the BNE. We see that these almost exactly overlap, with only very minor discrepancies, which can be explained by noise in the observed market parameters. On the right we plot the convergence of the agent strategies during online learning to the BNE strategies in terms of mean-squared error (MSE); we see that the MSE very rapidly approaches 0, even faster than guaranteed by our theory. Overall, these results are a strong empirical validation of Theorem 6.

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

## STATEMENT ON USAGE OF LLMS

We used LLM tools for the purpose of checking language use in writing our paper, as well as a tool for suggesting relevant existing results when conducting our research (in particular, for exploring existing results related to uniqueness of Nash Equilibria, for which it suggested reading literature / existing results on variational inequalities.) However, all proofs were derived and written completely by the authors, and the paper was written completely by the authors (outside of the usage of LLMs for language checking as mentioned above.)

# A  ADDITIONAL MODEL DISCUSSION

## A.1  DERIVATION OF PRICE MODEL

First, we discuss in more detail how our price dynamics in Assumption 1 relate to Walrasian price dynamics. As mentioned in Section 2, this model positions that (mean) prices evolve according to the continuous time differential equation

$$dp_t = \alpha(\text{demand}_t - \text{supply}_t)dt\,,$$

for some price-sensitivity factor $\alpha$. Given this, the dynamic model for $p_t^w$ can be viewed as a discretization of this process. In addition, allowing for noise in $s_t$, this turns it into a discretization of the corresponding stochastic differential equation

$$dp_t = \alpha(\text{demand}_t - \text{supply}_t)dt + \sigma dW_t\,,$$

for some noise process $dW_t$ (*e.g.* Brownian motion). The actual price that traders must pay differs from this Walrasian process by amount $\beta(\sum_{i=1}^n h_{i,t} + s_t)$. We can interpret this difference as a temporary (instantaneous) price adjustment from $p_t^w$ driven by the imbalance of supply and demand; when supply and demand are not balanced, the difference must be met by *market makers*, who provide liquidity. These market makers require some spread from $p_t^w$ in order to account for the risk they take by providing liquidity. For example, if demand outstrips supply at time $t$, the market makers will balance this by selling an equal amount at a slight premium; hence, the instantaneous market price $p_t$ will be slightly higher than $p_t^w$. We implicitly assume as part of Assumption 1 that this difference is linear in the imbalance $\sum_{i=1}^n h_{i,t} + s_t$, with coefficient $\beta$.

Alternatively, this model can also be justified from the literature on market impact. For example, the seminal model of Almgren & Chriss (2000), which is the basis for much of the classical theory on (non-strategic) optimal trade execution, posits almost exactly the same model for price impact from trade execution over time, except that they consider a slightly more general offset based on supply and demand imbalance of the kind $\psi(\sum_{i=1}^n h_{i,t} + s_t)$, for some concave function $\psi : \mathbb{R}^+ \to \mathbb{R}^+$. Therefore, our price model is equivalent to theirs in the case of $\psi(x) = \beta x$. We note that empirical research (see *e.g.* Almgren et al. (2005)) suggests power-law models of the kind $\psi(x) = \beta x^\gamma$ with $\gamma \approx 3/5$ to be well supported by real data. Such a model would be more challenging to study under strategic interaction, as it could break strong monotonicity without some additional assumptions on $f_i$ and/or $G_i$; we leave the investigation of alternative price models like this to future work.

## A.2  RELATION OF MODEL TO EXISTING MODELS

Here, we make some concrete comparisons of our model with the models used in the recent lines of work on position building under competition Chriss (2024b;c;a; 2025); Kearns & Shi (2025).

**Relation to Existing Discrete Time Model**  First, consider the special case of our model with no idiosyncratic utilities ($f_i = 0$ for all $i$), and no exogenous actions ($\boldsymbol{s} = 0$). In this case, we can re-formulate the objective for each agent as minimizing a cost function $c_i(\boldsymbol{h}_i, \boldsymbol{h}_{-i})$ given by the negative of the utility $u_i$, which if we unroll the autoregressive price definitions like in the proof of Lemma 1, we can easily verify is given by

$$c_i(\boldsymbol{h}_i; \boldsymbol{h}_{-i}) = \alpha \sum_{t=1}^T h_{i,t} \sum_{l=1}^t \sum_{j=1}^n h_{j,l} + \beta \sum_{t=1}^T h_{i,t} \sum_{j=1}^n h_{j,t} + \sum_{t=1}^T h_{i,t} p_0$$

Now, assume further that the constraints $G_i$ contains a constraint of the kind $\sum_{t=1}^T h_{i,t} = V_i$ for some fixed $V_i$. Then, the third term above can be ignored, as it is always equal to $p_0 V_i$ for any feasible $\boldsymbol{h}_i \in G_i$. Given this, and with some slight re-arranging of terms, we have that the cost structure is given by

$$c_i(\boldsymbol{h}_i; \boldsymbol{h}_{-i}) = \alpha \sum_{t=1}^T h_{i,t} \sum_{j=1}^n x_{j,t-1} + (\alpha + \beta) \sum_{t=1}^T h_{i,t} \sum_{j=1}^n h_{j,t}\,,$$

where we define

$$x_{i,t} = \sum_{l=1}^{t} h_{i,l}$$

as the cumulative position acquired by agent $i$ over the first $t$ time steps. This corresponds exactly to the kind of cost structure assumed in Kearns & Shi (2025), who considered a discrete time version of optimal position building, with cost function

$$c_i^{\mathrm{KS}}(\boldsymbol{h}_i; \boldsymbol{h}_{-i}) = \kappa \sum_{t=1}^{T} h_{i,t} \sum_{j=1}^{n} x_{j,t-1} + \sum_{t=1}^{T} h_{i,t} \sum_{j=1}^{n} h_{j,t}\,.$$

Following terminology for literature on optimal position building, they denote first term as the *permanent-impact cost*, and the second term as the *temporary-impact cost*, with permanent-impact coefficient $\kappa$, and unit temporary-impact coefficient (which is completely general up to normalization of cost). Therefore, if we normalize our cost by $\alpha + \beta$, we see that under the above model restrictions it recovers theirs with $\kappa = \alpha/(\alpha + \beta)$.

Although our model may seem less general given the above reduction, as they allow for any $\kappa \geq 0$ but ours only allows $\kappa \in [0,1]$, we argue that this restriction does not have much or any material impact in practice. First, as discussed in Kearns & Shi (2025), if they decompose their cost into zero-sum and potential (*i.e.* congestion game-style cost) components, the coefficient in front of potential cost becomes negative when $\kappa > 2$. This implies that agents are rewarded rather than punished from congestion of their trading schedule, which therefore encourages agents to behave as aggressively as their constraints will allow (this is reflected *e.g.* in the unstable dynamics they observe when agents play no-regret with $\kappa > 2$). Given this, we would probably wish to restrict to $\kappa < 2$ in such a discrete model in practice. Second, and perhaps more importantly, to the extent that their model is justified as a discretization of the continuous time model discussed below, the convergence of this discretization as we make it more and more fine-grained only works if we let the ratio of temporary-impact-coefficient to permanent-impact-coefficient (*i.e.* $\kappa$) tend towards zero as $T \rightarrow \infty$. Therefore, no matter the target $\kappa$ value in the continuous-time cost $c_i^{\mathrm{NC}}$ defined below, the corresponding $\kappa$ in the discrete-time cost $c_i^{\mathrm{KS}}$ that approximates this will be less than 1 if the discretization is sufficiently fine-grained.

**Relation to Existing Continuous Time Model**     The works by Chriss (2024b;c;a; 2025) consider a continuous-time version of this problem, where the strategies $\boldsymbol{h}_i$ are functions over some continuous time range (which they normalize to be $[0,1]$ without loss of generality). In this setting, we assume the strategies are defined by functions $\boldsymbol{h}_i : [0,1] \rightarrow \mathbb{R}$, where $\boldsymbol{h}_i(t)$ is their instantaneous trading rate at time $t$. We also define $\boldsymbol{x}_i$ implicitly in terms of $\boldsymbol{h}_i$ as the total accumulated position up to time $t$, which is mathematically given by

$$\boldsymbol{x}_i(t) = \int_0^t \boldsymbol{h}_i(l)dl\,.$$

Then, the assumed cost structure is

$$c_i^{\mathrm{NC}}(\boldsymbol{h}_i; \boldsymbol{h}_{-i}) = \kappa \int_0^1 \boldsymbol{h}_i(t) \sum_{j=1}^{n} \boldsymbol{x}_i(t)dt + \int_0^1 \boldsymbol{h}_i(t) \sum_{j=1}^{n} \boldsymbol{h}_j(t)dt\,,$$

which is the continuous-time analogue of the cost structure based on decomposition into permanent-impact cost and temporary-impact cost mentioned above.[3]

Now, suppose we are given a problem instance of this continuous time model, with $\kappa$ given, and time normalized into range $[0,1]$. We can approximate this arbitrarily well with a discrete time model as $T \rightarrow \infty$, by letting the discrete time grid correspond to $\{\frac{1}{T}, \frac{2}{T}, \dots 1\}$ in continuous time. Specifically, we can do this as follows: suppose we are given collection of continuous-time strategies

---

[3]In reality, historically this continuous time model was the original one and the above discrete time version was introduced later, but we present in opposite order since our model is discrete-time.

$\boldsymbol{h}_1^c, \ldots, \boldsymbol{h}_n^c$, and define

$$\boldsymbol{x}_i^c(t) = \int_0^t \boldsymbol{h}_i^c(l)dl \qquad \text{(for continuous } t \in [0, 1]\text{)}$$

$$x_{i,t} = \boldsymbol{x}_i^c\left(\frac{t}{T}\right) \qquad \text{(for discrete } t \in [T]\text{)}$$

$$h_{i,t} = x_{i,t} - x_{i,t-1} \qquad \text{(for discrete } t \in [T])\,.$$

Then, our discrete-time cost structure in terms of these stragegy vectors $\boldsymbol{h}_i$ will be given by

$$c_i^{\alpha,\beta,T}(\boldsymbol{h}_i; \boldsymbol{h}_{-i}) = \alpha \sum_{t=1}^T h_{i,t} \sum_{j=1}^n x_{j,t-1} + (\alpha + \beta) \sum_{t=1}^T h_{i,t} \sum_{j=1}^n h_{j,t}$$

$$= \alpha \sum_{t=1}^T \left\{ \boldsymbol{x}_i^c\left(\frac{t}{T}\right) - \boldsymbol{x}_i^c\left(\frac{t-1}{T}\right) \right\} \sum_{j=1}^n \boldsymbol{x}_j^c\left(\frac{t-1}{T}\right)$$

$$+ (\alpha + \beta) \sum_{t=1}^T \left\{ \boldsymbol{x}_i^c\left(\frac{t}{T}\right) - \boldsymbol{x}_i^c\left(\frac{t-1}{T}\right) \right\} \sum_{j=1}^n \left\{ \boldsymbol{x}_j^c\left(\frac{t}{T}\right) - \boldsymbol{x}_j^c\left(\frac{t-1}{T}\right) \right\}$$

$$= \alpha \sum_{t=1}^T \frac{1}{T} \boldsymbol{h}_i^c\left(\frac{t-\gamma_{i,t}}{T}\right) \sum_{j=1}^n \boldsymbol{x}_j^c\left(\frac{t-1}{T}\right)$$

$$+ (\alpha + \beta) \sum_{t=1}^T \frac{1}{T} \boldsymbol{h}_i^c\left(\frac{t-\gamma_{i,t}}{T}\right) \sum_{j=1}^n \frac{1}{T} \boldsymbol{h}_j^c\left(\frac{t-\gamma_{j,t}}{T}\right),$$

where the final line follows from the mean-value theorem, where $\gamma_{i,t} \in (0, 1)$ for all $i, t$. Therefore, if we consider a sequence of discrete problem instances with $\alpha = \kappa$ and $\beta = T$, we get

$$\lim_{T\to\infty} c_i^{\kappa,T,T}(\boldsymbol{h}_i; \boldsymbol{h}_{-i}) = \lim_{T\to\infty} \kappa \frac{1}{T} \sum_{t=1}^T \boldsymbol{h}_i^c\left(\frac{t-\gamma_{i,t}}{T}\right) \sum_{j=1}^n \boldsymbol{x}_j^c\left(\frac{t-1}{T}\right)$$

$$+ \lim_{T\to\infty} \left(\frac{\kappa+T}{T}\right) \frac{1}{T} \sum_{t=1}^T \boldsymbol{h}_i^c\left(\frac{t-\gamma_{i,t}}{T}\right) \sum_{j=1}^n \boldsymbol{h}_j^c\left(\frac{t-\gamma_{j,t}}{T}\right)$$

$$= \kappa \int_0^1 \boldsymbol{h}_i^c(t) \sum_{j=1}^n \boldsymbol{x}_i^c(t)dt + \int_0^1 \boldsymbol{h}_i^c(t) \sum_{j=1}^n \boldsymbol{h}_j^c(t)dt\,,$$

where first equality plugs in the above result with $\alpha = \kappa$ and $\beta = T$, and the second follows from product of limits and the definition of the Riemann integral. Therefore, under appropriate re-normalization of the ratio $\beta/\alpha$ as we make the discrete-time approximation more fine-grained, our model can approximate the existing continuous time cost structure considered in Chriss (2024b;c;a; 2025) arbitrarily well if we let $T \to \infty$. Therefore, our model on the above restriction on idiosyncratic utilities, constraints, and exogenous actions subsumes theirs up to a vanishing discretization error.

## A.3  Additional Considerations for Application

Here we discuss some additional considerations for applying our model in practice. We separately discuss the considerations for our game theoretic model and the implementability of our learning algorithms in Section 5. To keep this discussion focused, we consider the application of our model to financial settings, where the resource being acquired is (fractional) units of some asset; data-driven optimal resource acquisition is commonly done there in practice, so this is a particularly salient.

**How realistic is our generalized game framework?**  In the absence of other players (*i.e.* when $n = 1$), our setting subsumes the seminal optimal execution setting considered by Almgren &

Chriss (2000). While richer market impact models have been proposed (see *e.g.* Li et al. (2024) for a detailed overview), the simple linear/quadratic model of Almgren & Chriss (2000)—which we adopt—remains widely used today for its practicality.

Our generalized game model makes few limiting assumptions beyond the fact that price impact occurs based on aggregate player actions as in Almgren & Chriss (2000). Information can be imperfect, asymmetric, and have generic structure, and players need not know the game's information structure a priori. Likewise, our constraint and utility structure is flexible: players can target fixed positions or maximize arbitrary utility functions on their final position, and can face realistic constraints (no short selling, limitations on how much players can buy or sell at each time step, upper/lower limits on the allowed position the player can have at any time, *etc.*). All this puts very little limitations on the objectives of the players within the competitive resource acquisition environment. While prior work has explored some of the above missing aspects in isolation, but never together within a sufficiently general framework. Exploring these aspects in isolation often trivializes them; for example, considering imperfect information without consideration of constraints or learning, as in Chriss (2025), leads to methods that cannot extend to more realistic settings. That said, we also acknowledge some limitations of our model in practice.

- First, the price impact and execution model is somewhat stylized. In reality, price impact depends on rich market microstructure features. However, while richer market impact models have been proposed (see *e.g.* Li et al. (2024) for a detailed overview), the simple linear/quadratic model of Almgren & Chriss (2000)—which we adopt—remains widely used in practice and in the literature.

- Second, we assume that agents commit to their full trajectory of behavior $\boldsymbol{h}$ *ex-ante*, and cannot adjust dynamically to the behavior of other agents. We note, however, that dynamic interaction could be approximated by splitting the time up into many smaller horizons, each modeled as a separate instance of our game, which are each short enough that this commitment is more reasonable.

- Finally, our results are restricted to discrete-type spaces. This limitation is mostly technical: any continuous type space can always be approximated via some discretization, and from an agent perspective, arbitrarily large type spaces could be handled by leveraging machine learning (*e.g.* agents could learn a neural network mapping from their type $\theta$ to their trajectory $\boldsymbol{h}$). Some of our results (*e.g.* our average-iterate convergence to BCCE result in Theorem 5) are general enough to accommodate discretization or function approximation approaches, given regret guarantees. Others (in particular the last-iterate convergence to BNE in Theorem 6) cannot currently accommodate such approaches, since they require finite-dimensional action spaces, leaving open an important future direction.

**How implementable are the learning algorithms considered in Section 5 in practice?** Next, we move beyond the realism of the setting itself, and consider how practical our proposed learning algorithms are for actual implementation. First, we consider the basic requirements of such algorithms in terms of required inputs. In general, these algorithms assume (Assumption 2) that the agent observes sufficient information to be able to compute what their utility would have been if they had (counterfactually) instead followed a different trajectory, with the trajectories of all other players fixed. For Algorithm 2, for which we prove last iterate convergence to BNE (Theorem 6), we require something much weaker: stochastic gradient of the agent's expected utility given fixed actions for all others. Clearly, taking the gradient of the counterfactual cost is an unbiased stochastic gradient of their expected Bayesian game utility. So it suffices to reason about the practicality of this counter-factual cost feedback.

In any reasonable market, the trajectories of prices $p_t$ and $p_t^w$ can be directly observed or inferred from limit order book information for electronic exchange-traded assets. Furthermore, without observing the actions of others, it is reasonable in public markets that agents have some reasonable estimates of the current *aggregate* external demand. Given this information, we can proceed as follows: first, let $h_t$ be a single agent's action at a single time, and $d_t$ be the current aggregated external demand from all other agents (strategic or exogenous). Then, the price model in Assumption 1 gives an over-determined system of $2T$ equations for $d_1, \ldots, d_T, \alpha$, and $\beta$, which are $p_t^w - p_{t-1}^w = \alpha(h_t + d_t)$ and $p_t - p_t^w = \beta(h_t + d_t)$ for all $t \in [T]$. First, using the (possibly noisy) estimates of the excess

demands $\hat{\boldsymbol{d}}$, the agent can perform *e.g.* a simple linear regression to obtain estimates $\hat{\alpha}$ and $\hat{\beta}$ of the price impact coefficients from these equations. The estimated total aggregate demand $\hat{\boldsymbol{d}}$ could then optionally be refined by plugging the estimated price impact coefficients. Then, by Lemma 1, the counterfactual total cost function for the single player if they changed their trajectory to $\tilde{\boldsymbol{h}}$ can be estimated as

$$c(\tilde{\boldsymbol{h}}) = -f(\tilde{\boldsymbol{h}}) + \frac{1}{2}\tilde{\boldsymbol{h}}^\top Q(\hat{\alpha},\hat{\beta})\tilde{\boldsymbol{h}} + \left(A(\hat{\alpha},\hat{\beta})\hat{\boldsymbol{d}}\right)^\top \tilde{\boldsymbol{h}} + p_0 \mathbf{1}^\top \tilde{\boldsymbol{h}},$$

where we make the dependence of $Q$ and $A$ on the price impact coefficients explicit. Similarly, the stochastic cost gradient computed at $\tilde{\boldsymbol{h}} = \boldsymbol{h}$ used by Algorithm 2 is then given by

$$\nabla c(\boldsymbol{h}) = -\nabla f(\boldsymbol{h}) + Q(\hat{\alpha},\hat{\beta})\boldsymbol{h} + A(\hat{\alpha},\hat{\beta})\hat{\boldsymbol{d}} + p_0 \mathbf{1}.$$

While, of course, these are not the true counterfactual cost or corresponding gradient given the estimation of $\boldsymbol{d}$, $\alpha$, and $\beta$, our theory only depends on these being stochastic estimates of the corresponding expected Bayesian game cost or cost gradient, so we argue that this is a very minor and technical limitation in practice.

Second, we consider the implication of the mismatch between our idealized mathematical framework and reality, on how the learning algorithm can be implemented. In our framework, agents can trade any fractional asset amount at each time step (at endogenously-determined price $p_t$). On the other hand, in reality, agents must interact with markets via the details of the market microstructure, which introduces some additional challenges. For example, for assets traded in electronic exchanges, agents may need to interact with the corresponding limit order book, and trades may only be possible in particular discrete quantities depending on available bids and offers in the book. However, this is a minor and technical limitation that is commonly addressed in practice for executing orders in electronic market places. For example, an extremely well studied problem is how to execute large orders in real market places at a constant trading rate in volume-weighted time (see for example Donnelly (2022)). Since our mathematical framework for price impact implicitly assumes that time is measured in volume-weighted units (see *e.g.* discussion in Almgren et al. (2005)), these algorithms for interacting with market microstructure for constant trading rate execution could be applied to put our algorithms into practice. In particular, such optimal execution algorithms could be applied within each discrete time step for trading within that time step at a constant rate to acquire the target amount for that step.

Finally, as discussed above in our limitations of the overall game framework, in practice the contextual information contained in player types $\theta$ are unlikely to be discrete, but contain rich continuous-valued information that require some function approximation to handle. This could be handled in several different ways to be able to put our learning algorithms into practice. One possibility is we could do some clustering of this contextual information into discrete bins, and then apply online learning on these clusters exactly as described in Section 5, *e.g.* by applying Algorithm 2 with these discretized types. This kind of discretization approach is popular for online learning with continuous-valued context, see *e.g.* Krishnamurthy et al. (2020) or Sinclair et al. (2023).) Alternatively, as discussed above, we could instantiate our general framework that we provide results for in Theorem 5 with online learning algorithms that can handle continuous-valued context and do function approximation.

## B  PROOFS FOR SECTION 3

PROOF OF LEMMA 1:

We first unroll the auto-regressive nature of the Walrasian price dynamic $p_t^w$. Observe that the following holds:

$$p_1^w = p_0 + \alpha \sum_j h_{j,1} + \alpha s_1 \ ;$$

$$p_2^w = p_0 + \alpha \sum_j h_{j,1} + \alpha s_1 + \alpha \sum_j h_{j,2} + \alpha s_2 \ ; \ \ldots$$

The execution price an agent pays is also influenced by the temporary impact. Combining this with the above, we can write the net cost an agent $i$ faces as follows: $u_i(\boldsymbol{h}_1, \ldots, \boldsymbol{h}_n, \boldsymbol{\lambda})$

$$= f_i(\boldsymbol{h}_i) - \sum_t h_{i,t} p_0 - \alpha \sum_{t=1}^T \sum_{\ell=1}^t h_{i,t} h_{i,\ell} - \alpha \sum_{t=1}^T \sum_{\ell=1}^t h_{i,t} \Big(\sum_{j \neq i} h_{j,\ell} + s_\ell\Big) - \beta \sum_{t=1}^T h_{i,t} \Big( \sum_{j=1}^n h_{j,t} + s_t \Big)$$

$$= f_i(\boldsymbol{h}_i) - \alpha \underbrace{\sum_{t=1}^T \Big( h_{i,t}^2 + h_{i,t} \sum_{\ell=1}^{t-1} h_{i,\ell} \Big) - \beta \sum_{t=1}^T h_{i,t}^2}_{\text{quadratic terms}}$$

$$- \alpha \underbrace{\sum_{t=1}^T h_{i,t} \sum_{\ell=1}^t \sum_{j \neq i} h_{j,\ell} - \beta \sum_{t=1}^T h_{i,t} \sum_{j \neq i} h_{j,t}}_{\text{linear terms} \propto \text{ other agent}} - \alpha \underbrace{\sum_{t=1}^T h_{i,t} \sum_{\ell=1}^t s_\ell - \beta \sum_{t=1}^T h_{i,t} s_t}_{\text{linear term} \propto \text{ exogenous agent}} - \sum_t p_0 h_{i,t}$$

Focusing on the quadratic terms, it suffices to compute the Hessian, denoted by $Q$. Note that $Q_{t,t} = 2\alpha + 2\beta$. As for the off-diagonal values, these are composed entirely of $\alpha$. Indeed, for any $t_1 \neq t_2$, we have that $Q_{t_1,t_2} = \alpha$. Next, we consider the linear terms that are proportional to other agents. We wish to express it in the following form: $\sum_{j \neq i} (A \boldsymbol{h}_j(\theta_j))^T \boldsymbol{h}_i(\theta_i)$. For a given $t$, consider the first of the two linear terms proportional to others. For any $j$, observe that $h_{i,t}(\theta_i)$ is multiplied by $\alpha h_{j,1}(\theta_j), \ldots \alpha h_{j,1}(\theta_j)$. As for the second term, it multiplies $h_{i,t}(\theta_i)$ with $\beta h_{j,t}(\theta_j)$. Hence, we conclude that $A$ is a lower-triangular matrix, whose diagonals are $\alpha + \beta$ and the remaining values are $\alpha$. As for the linear term with respect to the exogenous agent, it follows a similar pattern, and we can express it as $(A\boldsymbol{s})^T \boldsymbol{h}_i$. We thus have the following expression for the matrices $Q$ and $A$:

$$Q_{ij} = \begin{cases} \alpha & \text{if } i < j \\ 2\alpha + 2\beta & \text{if } i = j \\ \alpha & \text{if } i > j \end{cases} \ ; \ A_{ij} = \begin{cases} 0 & \text{if } i < j \\ \alpha + \beta & \text{if } i = j \\ \alpha & \text{if } i > j \end{cases} \ ;$$

Since $Q$ is a symmetric matrix that can be written as $Q = \alpha J + (\alpha + 2\beta)I$ where $J$ is the all 1s matrix and $I$ the identity matrix. Observe that for any $x$, we have that:

$$x^T Q x = (\alpha + 2\beta)(x^T I x) + \alpha(x^T J x) = (\alpha + 2\beta)(x^T x) + \alpha(x^T J x) = (\alpha + 2\beta)||x||_2^2 + \alpha \Big( \sum_{i=1}^T x_i \Big)^2 > 0$$

where the strict inequality holds since the parameters $\alpha, \beta$ are non-negative and $\boldsymbol{x} \neq 0$. In other words, the $Q$ matrix is positive definite and thus the utility of each agent, in terms of their own strategy, is a strictly concave function.

PROOF OF LEMMA 2

*Proof.* Consider an agent $i$ and suppose all other agents are playing a mixed and possibly correlated strategy, denoted by $\sigma_{-i}$. Buyer $i$ can choose to best-respond with a mixed strategy of their own,

denoted $p_i(h)$. Without loss of generality, we set $p_0 = 0$ and observe that:

$$\text{br}_i(\sigma_{-i}) = \underset{p \in \Delta(G_i)}{\arg\max} \int_{\boldsymbol{h}_i} p(\boldsymbol{h}_i) \left( f_i(\boldsymbol{h}_i) - \boldsymbol{s}A\boldsymbol{h}_i - \int_{\boldsymbol{h}_{-i}} \sigma(\boldsymbol{h}_{-i})[\boldsymbol{h}_i^T Q \boldsymbol{h}_i + \sum_{j \neq i} (A\boldsymbol{h}_j)^T \boldsymbol{h}_i] d\boldsymbol{h}_{-i} \right) d\boldsymbol{h}_i$$

$$= \underset{p \in \Delta(G_i)}{\arg\max} \int_{\boldsymbol{h}_i} p(\boldsymbol{h}_i) \left[ f_i(\boldsymbol{h}_i) - \boldsymbol{h}_i^T Q \boldsymbol{h}_i - \boldsymbol{s}A\boldsymbol{h}_i - \underbrace{\left( \int_{\boldsymbol{h}_{-i}} \sigma(\boldsymbol{h}_{-i}) \sum_{j \neq i} (A\boldsymbol{h}_j)^T d\boldsymbol{h}_{-i} \right)}_{\boldsymbol{v}_{-i}^T} \boldsymbol{h}_i \right] d\boldsymbol{h}_i$$

$$= \underset{p \in \Delta(G_i)}{\arg\max} \int_{\boldsymbol{h}_i} p(\boldsymbol{h}_i) \left[ f_i(\boldsymbol{h}_i) - \boldsymbol{h}_i^T Q \boldsymbol{h}_i - \boldsymbol{v}_{-i}^T \boldsymbol{h}_i - \boldsymbol{s}A\boldsymbol{h}_i \right] d\boldsymbol{h}_i$$

$$= \underset{\boldsymbol{h}_i \in G_i}{\arg\max} \left[ f_i(\boldsymbol{h}_i) - \boldsymbol{h}^T Q \boldsymbol{h} - \boldsymbol{v}_{-i}^T \boldsymbol{h}_i - \boldsymbol{s}A\boldsymbol{h}_i \right]$$

where the last equality follows from the linearity of expectation. The resulting maximization expression has a sole quadratic term: $\boldsymbol{h}_i^T Q \boldsymbol{h}_i$. From Lemma 1 we know $Q$ is a PD matrix. Thus, the best-response optimization, to both pure or mixed strategies of others, is strictly concave, and there is a unique solution.

$\square$

### B.1 PROOF OF THEOREM 1

*Proof.* **Uniqueness:** From 2, we know that each agent's best response is a concave optimization problem over a convex region $G_i$. In this proof, we shall express the agent's objective from a cost minimization perspective – i.e. $c_i(\boldsymbol{h}_i; \boldsymbol{h}_{-i}, \boldsymbol{\lambda}) = -u_i(\boldsymbol{h}_i; \boldsymbol{h}_{-i}, \boldsymbol{\lambda})$. As such, the best response objective will be convex. Formally (again setting $p_0 = 0$ for ease of exposition):

$$\text{br}_i(\boldsymbol{h}_{-i}) = \underset{\boldsymbol{h}_i \in G_i}{\arg\min} \frac{1}{2} \boldsymbol{h}_i^T Q \boldsymbol{h}_i + \sum_{j \neq i} (A\boldsymbol{h_j})^T \boldsymbol{h}_i + \boldsymbol{s}A\boldsymbol{h}_i - f_i(\boldsymbol{h}_i) := \underset{\boldsymbol{h}_i \in G_i}{\arg\min} c_i(\boldsymbol{h}_i; \boldsymbol{h}_{-i}, \boldsymbol{\lambda})$$

At a best response for agent $i$, $\boldsymbol{h}_i^*$, it must be that for all $\forall \boldsymbol{h}_i \in G_i : \langle \nabla_{\boldsymbol{h}_i} c_i(\boldsymbol{h}_i^*; \boldsymbol{h}_{-i}), (\boldsymbol{h}_i - \boldsymbol{h}_i^*) \rangle \geq 0$; otherwise, the agent could move in that direction and decrease their net cost. This is a standard equivalence between a convex optimization and variational inequalities (see Rockafellar & Wets (2009)). At a Nash Equilibrium, each buyer $i$ must be playing their best response, given the strategies of other agents. Thus, we are looking for a set of trajectories $(\boldsymbol{h}_1^{eq}, \ldots, \boldsymbol{h}_n^{eq})$ such that:

$$\forall i, \forall (\boldsymbol{h}_1, \ldots, \boldsymbol{h}_n) \in R_1 \times \cdots \times R_n : \langle \nabla_{\boldsymbol{h}_i} c_i(\boldsymbol{h}_i^{eq}; \boldsymbol{h}_{-i}^{eq}), (\boldsymbol{h}_i - \boldsymbol{h}_i^{eq} \rangle \geq 0 \tag{3}$$

Indeed, any tuple $(\boldsymbol{h}_1^{eq}, \ldots, \boldsymbol{h}_n^{eq})$ that satisfies the above must be a Nash Equilibrium. Observe that $c_i$ is the sum of a quadratic function and a convex term. The gradient of $c_i$ is then as follows:

$$\nabla_{\boldsymbol{h}_i} c_i(\boldsymbol{h}_i^*; \boldsymbol{h}_{-i}) = Q\boldsymbol{h}_i^* + \sum_{j \neq i} A\boldsymbol{h}_j + A\boldsymbol{s} - \nabla_{\boldsymbol{h}_i} f_i(\boldsymbol{h}_i)$$

Ignoring the $\nabla_{\boldsymbol{h}_i} f_i(\boldsymbol{h}_i)$ term, the gradient is a linear function of all agent strategies. Denoting $\boldsymbol{x} = [\boldsymbol{h}_1, \ldots, \boldsymbol{h}_n]^T$ as the concatenation of agent strategies and $\boldsymbol{s}$ as a constant (since this is not from a strategic agent), the variational inequality that characterizes the Nash Equilibrium can thus written with an operator $F(\boldsymbol{x}) = M\boldsymbol{x} + b - [\nabla_{\boldsymbol{h}_1} f_i(\boldsymbol{h}_1), \ldots \nabla_{\boldsymbol{h}_n} f_i(\boldsymbol{h}_n)]$. That is, a set of strategies $\boldsymbol{x}^{eq} = [\boldsymbol{h}_1^{eq}, \ldots, \boldsymbol{h}_n^{eq}]$ is a Nash Equilibrium if and only if, for all $\boldsymbol{x} \in R_1 \times \ldots, \times R_n : \langle F(\boldsymbol{x}^{eq}), (\boldsymbol{x} - \boldsymbol{x}^{eq}) \rangle \geq 0$. We note the following:

**Definition 9** (Rockafellar & Wets (2009)). *An operator $F$ is called strongly monotone on a set $\mathcal{X}$ if and only if there exists a scaler $c$ such that:*

$$\langle F(\boldsymbol{x}) - F(\boldsymbol{x}'), (\boldsymbol{x} - \boldsymbol{x}') \rangle \geq c ||\boldsymbol{x} - \boldsymbol{x}'||^2, \quad \forall \boldsymbol{x}_1, \boldsymbol{x}_2 \in \mathcal{X}$$

From Rockafellar & Wets (2009), we note that a variational inequality with a strongly monotone operator has a unique solution. Here, this implies a unique Nash Equilibrium. For our operator $F$:

$$\langle F(\boldsymbol{x}) - F(\boldsymbol{x}'), (\boldsymbol{x} - \boldsymbol{x}') \rangle = \langle M(\boldsymbol{x} - \boldsymbol{x}'), (\boldsymbol{x} - \boldsymbol{x}') \rangle - \sum_{i=1}^n \langle \nabla_{\boldsymbol{h}_i} f_i(\boldsymbol{h}_i) - \nabla_{\boldsymbol{h}_i'} f_i(\boldsymbol{h}_i'), (\boldsymbol{h}_i - \boldsymbol{h}_i') \rangle$$

As such, it suffices to show that for positive $c$: $\langle M(\boldsymbol{x} - \boldsymbol{x}'), (\boldsymbol{x} - \boldsymbol{x}')\rangle \geq c||\boldsymbol{x} - \boldsymbol{x}'||^2$ and for all $i$: $-\langle \nabla_{\boldsymbol{h}_i} f_i(\boldsymbol{h}_i) - \nabla_{\boldsymbol{h}_i'} f_i(\boldsymbol{h}_i'), (\boldsymbol{h}_i - \boldsymbol{h}_i')\rangle \geq 0$. It is known that for convex functions, their gradient is a monotone operator. Thus, $f_i$ being concave (and thus $-f_i$ is convex) and the desired condition immediately holds. As such, it suffices to prove the strong monotonicity of the linear operator $M$ and show that for any $\boldsymbol{x}$: $\langle M\boldsymbol{x}, \boldsymbol{x}\rangle = \boldsymbol{x}^T M \boldsymbol{x} \geq c||\boldsymbol{x}||^2$. The matrix $M$ is given by the following block matrix:

$$M = \begin{bmatrix} Q \in \mathbb{R}^{T \times T} & A \in \mathbb{R}^{T \times T} & \ldots & A \in \mathbb{R}^{T \times T} \\ A \in \mathbb{R}^{T \times T} & Q \in \mathbb{R}^{T \times T} & \ldots & A \in \mathbb{R}^{T \times T} \\ \vdots & \vdots & \vdots & \vdots \\ A \in \mathbb{R}^{T \times T} & A \in \mathbb{R}^{T \times T} & \ldots & Q \in \mathbb{R}^{T \times T} \end{bmatrix} \tag{4}$$

We first note that while our matrix $M$ may not symmetric, we can always write it as the sum of a symmetric component $M_s = \frac{1}{2}(M + M^T)$ and a skew-symmetric component $M_k = \frac{1}{2}(M - M^T)$. By definition, $M_k^T = -M_k$. This means that for any $\boldsymbol{x}$ where $s = \boldsymbol{x}^T M_k \boldsymbol{x}$: $s = s^T = (\boldsymbol{x}^T M_k \boldsymbol{x})^T = -\boldsymbol{x}^T M_k \boldsymbol{x} = -s$. Thus, $\boldsymbol{x}^T M_k \boldsymbol{x} = 0$ and it suffices to only consider the symmetric component $M_s$ for the strong monotonicity condition. Now suppose $M_s$ is a positive definite matrix. Then we can always diagonalize it as $P\Lambda P^T$, where $\Lambda$ is a diagonal matrix of positive eigenvalues and $PP^T = P^T P = I$. Then we can also express any $\boldsymbol{x} = P\boldsymbol{y}$ for some $\boldsymbol{y}$. Then under this PD condition, we have:

$$\boldsymbol{x}^T M \boldsymbol{x} = \boldsymbol{y}^T P^T P \Lambda P^T P \boldsymbol{y} = \boldsymbol{y}^T \Lambda \boldsymbol{y}$$
$$\geq \lambda_{min}(M_s)(P^T \boldsymbol{x})^T (P^T \boldsymbol{x}) = \lambda_{min}(M_s)\boldsymbol{x}^T PP^T \boldsymbol{x} = \lambda_{min}(M_s)||\boldsymbol{x}||^2$$

In other words, if $M_s$ is positive definite, we can choose $c = \lambda_{min}(M_s) > 0$ and satisfy the conditions of strong monotonicity. The matrix $M_s$ is an $n \times n$ block matrix with $Q$ on the diagonal and all other elements being $A_s = \frac{1}{2}(A + A^T)$. This can be succinctly represented using the Kronecker product (recall $J_n$ is an $n \times n$ all 1s matrix):

$$M_s = I_n \otimes (Q - A_s) + J_n \otimes A_s \tag{5}$$

Note that the all 1s matrix is positive-definite with one eigenvalue of $(n-1)$ and all other eigenvalues 0. Therefore, we can write $\Lambda_{J_n} = U^T J_n U$, where $\Lambda_{J_n} = \text{diag}(n, 0, \ldots 0)$. Let $P = U \otimes I_T$, and note that $P^T P = (U^T \otimes I_T)(U \otimes I_T) = U^T U \otimes I_T = I_{nT}$, where we use the mixed product property of Kronecker products. We shall be using $P$ to diagonalize (in the block sense) the matrix $M_s$. Specifically, observe that due to the mixed product rule:

$$P^T M_s P = P^T (I_n \otimes Q - A_s)P + P^T (J_n \otimes A_s)P$$
$$= (U^T \otimes I_T)(I_n \otimes Q - A_s)(U \otimes I_T) + (U^T \otimes I_T)(J_n \otimes A_s)(U \otimes I_T)$$
$$= (U^T I_n U \otimes I_T(Q - A_s)I_T) + (U^T J_n U \otimes I_T A_s I_T)$$
$$= (I_n \otimes (Q - A_s)) + (\Lambda_{J_n} \otimes A_s)$$

The first summand is a block diagonal matrix with $Q - A_s$ in each entry, and the second summand is also block diagonal with $nA_s$ in the first entry and 0 elsewhere. Therefore, $P^T M_S P$ results in a block diagonal matrix $\text{diag}(Q_s + (n-1)A_s, A_s, \ldots, Q - A_s)$. The eigenvalues of $M_s$ are the eigenvalues of this block diagonal matrix, which in turn are the eigenvalues of each matrix in the diagonal. Thus, we need to show that $Q + (n-1)A_s$ and $Q - A_s$ both have positive eigenvalues. Note that $Q = (\alpha + 2\beta)I_T + \alpha J_T$ and $A_s = (\frac{\alpha}{2} + \beta)I_T + \frac{\alpha}{2}J_T$. Thus, for any $\boldsymbol{x} \in \mathbb{R}^T$:

$$\boldsymbol{x}^T(Q - A_s)\boldsymbol{x}^T = \boldsymbol{x}^T\left[(\tfrac{\alpha}{2} + b)I_T + \tfrac{\alpha}{2}J_T\right]\boldsymbol{x} = (\tfrac{\alpha}{2} + b)\boldsymbol{x}^T\boldsymbol{x} + \tfrac{\alpha}{2}\left(\sum_{t=1}^{T} x_t\right)^2 > 0$$

$$\boldsymbol{x}^T(Q + (n-1)A_s)\boldsymbol{x}^T = \boldsymbol{x}^T\left[(n+1)(\tfrac{\alpha}{2} + b)I_T + (n+1)\tfrac{\alpha}{2}J_T\right]\boldsymbol{x}$$

$$= (n+1)(\tfrac{\alpha}{2} + b)\boldsymbol{x}^T\boldsymbol{x} + (n+1)\tfrac{\alpha}{2}\left(\sum_{t=1}^{T} x_t\right)^2 > 0$$

as long as either $\alpha > 0$ or $\beta > 0$. Since these diagonal matrices are positive definite, they have positive eigenvalues, implying $M_s$ has positive eigenvalues, implying strong monotonicity of the simultaneous best response operator, implying uniqueness of the equilibrium.

**Linear Convergence:**

---

**Algorithm 1:** Extra-Gradient Algorithm

---

**Input:** Game Instance $\mathcal{I}$, Variational Operator $F$, step-size $\eta$
Randomly Initialize a feasible joint strategy $\boldsymbol{x}_0 = (\boldsymbol{h}_1, \ldots, \boldsymbol{h}_n)$
**while** $||x_r - x_{r-1}|| \leq \varepsilon$ **do**
$\quad \boldsymbol{x}_{r+1/2} = \text{Proj}_{G_1 \times G_n}(x_r - \eta F(x_r))$
$\quad \boldsymbol{x}_{r+1} = \text{Proj}_{G_1 \times G_n}(x_r - \eta F(x_{r+1/2}))$

---

Theorem 3.4 of Wadia et al. (2024) states that for any $c$-strongly monotone and $L$-Lipshcitz operator, the extragradient algorithm (Algorithm 1) with step-size $\eta = \frac{1}{2(c+L)}$ converges to the fixed point at a linear rate of $1 - \frac{c}{4L}$. We have shown above that our given operator is $c = \lambda_{min}(M_s)$ strongly monotone. As for Lipschitzness, note that our operator can be decomposed as: $F(\boldsymbol{x}) = M\boldsymbol{x} + b - J(\boldsymbol{x})$, where $J(\boldsymbol{x}) = [\nabla_{\boldsymbol{h}_1} f_1, \ldots, \nabla_{\boldsymbol{h}_n} f_n]^T$. Lipschitz constants for the sum of two maps add; thus, it suffices to solve for the Lipschitz constants for the linear operator $M$, $L_M$ and the gradient operator $J$, $L_J$.

Any linear operator is lipschitz – in fact, the Lipschitz constant is just the 2-norm of the matrix $M$. For any matrix, the following is always true: $||M||_2 \leq \sqrt{||M||_1 ||M||_\infty}$, where $||M||_1$ is the largest absolute column sum and $||M||_\infty$ is the largest absolute row sum. In our specific matrix $M$, observe that:

$$||M||_1 = ||M||_\infty = (2\alpha + 2\beta) + (T-1)\alpha + (n-1)[(T-1)\alpha + \beta] = (nT+1)\alpha + (n+1)\beta \geq ||M||_2$$

Thus, $L_M = (nT+1)\alpha + (n+1)\beta$ is a suitable bound for the $M$ operator Lipschitz constant. Secondly, for any $i$, observe that since $f_i$ is concave, the operator $\nabla_{\boldsymbol{h}_i} f_i$ is $L_i = \sup_x \lambda_{max}(-\nabla_{\boldsymbol{h}_i} f_i)$ lipschitz. Further, observe that:

$$||J(\boldsymbol{x}) - J(\boldsymbol{x}')||^2 = \sum_{i=1}^n ||\nabla_{\boldsymbol{h}_i} f_i(\boldsymbol{h}_i) - \nabla_{\boldsymbol{h}_i'} f_i(\boldsymbol{h}_i')||^2 \leq \max_i L_i \sum_{i=1}^n ||\boldsymbol{h}_i - \boldsymbol{h}_i'||^2 = \max_i L_i ||\boldsymbol{x} - \boldsymbol{x}'||^2$$

Therefore, $L_J = \max_i L_i$ and the overall Lipschitz constant is $L_J + L_M = \max_i \sup_x \lambda_{max}(-\nabla_{\boldsymbol{h}_i} f_i) + (nT+1)\alpha + (n+1)\beta$. $\qquad \square$

### B.2 PROOF OF THEOREM 2

*Proof.* Suppose there are $n = 2$ agents and we have some constant values of $\alpha, \beta$ – one can assume, without loss of generality, that $\alpha = \beta = 1$[4]. Let the final position utility for both agents be given by the following linear function: $u_i(\boldsymbol{h}_i) = r_i \sum_t h_{i,t}$, where $r_i$ can be interpreted as the reserve/fair-market price as perceived by agent $i$. Further, the two have box constraints on their cumulative position: $V_i^- \leq \sum_i h_{i,t} \leq V_i^+$. We shall assume the exogenous agent is not present – i.e. $\boldsymbol{s} = \boldsymbol{0}$.

For a positive constant $x$, let the initial price $p_0 = x$ and the reserve prices for the agents be $(r_1 = x, r_2 = x - \varepsilon)$, where $\varepsilon > 0$. We first consider the equilibrium of this game without any constraints. Then each agent's best response is given by:

$$\text{br}_1(\boldsymbol{h}_2) = \arg\max_{\boldsymbol{h}_1} \left\{ -\frac{1}{2}\boldsymbol{h}_1^T Q \boldsymbol{h}_1 - (A\boldsymbol{h}_2)^T \boldsymbol{h}_1 \right\} \tag{6}$$

$$\text{br}_1(\boldsymbol{h}_2) = \arg\max_{\boldsymbol{h}_2} \left\{ -\varepsilon \mathbf{1}^T \boldsymbol{h}_2 - \frac{1}{2}\boldsymbol{h}_2^T Q \boldsymbol{h}_2 - (A\boldsymbol{h}_1)^T \boldsymbol{h}_2 \right\} \tag{7}$$

---

[4]Insofar as $\alpha, \beta$ are constants and not scaling with respect to the $\varepsilon$ all results will hold.

Observe that at the equilibrium of this unconstrained game, the gradient of both agents' best responses must be 0. Since this is a quadratic function, the gradient is linear, and the equilibrium can be uniquely specified by the following system of linear equalities:

$$\underbrace{\begin{bmatrix} Q & A \\ A & Q \end{bmatrix}}_{\text{Matrix } M \in \mathbb{R}^{2T \times 2T}} \begin{bmatrix} \boldsymbol{h}_1^{eq} \\ \boldsymbol{h}_2^{eq} \end{bmatrix} = \underbrace{\begin{bmatrix} \boldsymbol{0} \\ -\boldsymbol{\varepsilon} \end{bmatrix}}_{\boldsymbol{z} \in \mathbb{R}^{2T}}$$

Recall that the matrices $Q, A$ are specified using only the terms $\alpha, \beta$. In lemma 2, we noted that $Q$ is a positive-definite matrix and thus invertible. The matrix $A$ is a lower triangular matrix with $\alpha + \beta$ on the diagonals and is thus also invertible (insofar as $\alpha > 0$ or $\beta > 0$). As such, the matrix $M$ above is invertible and the unconstrained equilibrium strategy is given by $M^{-1}\boldsymbol{z}$. Note that this does not depend on the value of $x$. Further, if $V_i^- \leq -||M^{-1}\boldsymbol{z}||_1$ and $V_i^+ \geq ||M^{-1}\boldsymbol{z}||_1$, then this unconstrained equilibrium is also an equilibrium in the original constrained game. As for the strategy itself, let $m_{ij}$ denote the values of $-M^{-1}$ and note that $m_{ij}$ can be seen as a scaler with respect to $\varepsilon$. Then we have that:

$$h_{1t} = \varepsilon \sum_{j=T}^{2T} m_{t,j} \quad \text{and} \quad h_{2t} = \varepsilon \sum_{j=T}^{2T} m_{T+t,j} \tag{8}$$

Given that the value of the final position is simply the product of the total amount bought and the reserve, the utility of buyer 1 (with reserve $x$) is:

$$u_{eq}^1 = x\varepsilon \underbrace{\sum_{t=1}^{T} \sum_{j=T}^{2T} m_{t,j}}_{\sum_t h_{1t}} - \sum_{t=1}^{T} \left[ \underbrace{\sum_{j=T}^{2T} m_{t,j}\varepsilon \left( x + \alpha\varepsilon \sum_{\tau=1}^{t} \sum_{j=T}^{2T} (m_{\tau,j} + m_{T+\tau,j}) + \beta\varepsilon \sum_{j=T}^{2T} m_{t,j} + m_{T+t,j} \right)}_{\text{price } p_t} \right]$$

$$= \left| \sum_{t=1}^{T} \sum_{j=T}^{2T} m_{t,j}\varepsilon^2 \left( \alpha \sum_{\tau=1}^{t} \sum_{j=T}^{2T} (m_{\tau,j} + m_{T+\tau,j}) + \beta \sum_{j=T}^{2T} (m_{t,j} + m_{T+t,j}) \right) \right| = \Theta(\varepsilon^2)$$

where the absolute value in the second line follows, since utility at equilibrium will always be non-negative (the agents not trading would get utility 0, so utility at equilibrium must be at least 0). A similar analysis leads us to show that the utility of the second agent (with reserve $x - \varepsilon$) is also bounded by $\Theta(\varepsilon^2)$, allowing us to conclude that the welfare at equilibrium is $O(\varepsilon^2)$. Formally:

$$u_2^{eq} = \left| -\varepsilon^2 \sum_{t=1}^{T} \sum_{j=T}^{2T} m_{T+t,j} - \sum_{t=1}^{T} \sum_{j=T}^{2T} m_{T+t,j}\varepsilon^2 \left( \alpha \sum_{\tau=1}^{t} \sum_{j=T}^{2T} (m_{\tau,j} + m_{T+\tau,j}) + \beta \sum_{j=T}^{2T} (m_{t,j} + m_{T+t,j}) \right) \right|$$

We now turn to characterizing the optimal welfare of this instance. For some $\delta > 0$ (to be specified later), consider the following trajectories for each buyer (recall positive values mean buying):

$$\boldsymbol{h}_1 = [x, x, 0, \dots, 0] \quad \text{and} \quad \boldsymbol{h}_2 = [-x - \delta, -x - \delta, 0, \dots, 0] \tag{9}$$

Insofar as $V_i^+ \geq 2x$ and $V_i^- \leq -2x - \delta$, the trajectories above are feasible. Under this strategy, it suffices to consider the prices at rounds $t = 1, 2$, for which we have that: $p_1 = x - \alpha\delta - \beta\delta$ and $p_2 = x - 2\alpha\delta - \beta\delta$. Then the utilities for each buyer is given by:

$$u_1 = x \cdot 2x - x(x - \alpha\delta - \beta\delta) - x(x - 2\alpha\delta - \beta\delta) = 3\alpha\delta x + 2\beta\delta x$$

$$u_2 = (x - \varepsilon)(-2x - \delta) + (x + \delta)(x - \alpha\delta - \beta\delta) + (x + \delta)(x - 2\alpha\delta - \beta\delta)$$

$$= 2\varepsilon x + 2\delta\varepsilon - 3\alpha\delta x - 3\alpha\delta^2 - 2\beta\delta x - 2\beta\delta^2$$

$$\implies u_1^{opt} + u_1^{opt} \geq 2\varepsilon x + 2\delta\varepsilon - 3\alpha\delta^2 - 2\beta\delta^2 = 2x\varepsilon + 2\delta\varepsilon - (3\alpha + 2\beta)\delta^2$$

This gives a concave quadratic (in the unspecified parameter $\delta$) lower bound on the optimal utility. Maximizing it means choosing a $\delta$ such that the gradient is 0:

$$\delta = \frac{\varepsilon}{3\alpha + 2\beta} \implies u_1^{opt} + u_1^{opt} \geq 2x\varepsilon + \frac{2\varepsilon^2}{3\alpha + 2\beta} - \frac{\varepsilon^2}{3\alpha + 2\beta} = 2x\varepsilon + \frac{\varepsilon^2}{3\alpha + 2\beta} = \Theta(x\varepsilon)$$

From here, it is evident that for any constants $\alpha, \beta$ and $x$, we can construct an $\varepsilon > 0$ parametrized instance $\mathcal{I}_\varepsilon$ with box constraints $V_i^- \leq \min(-||M^{-1}\boldsymbol{z}||, -2x - 2\delta)$ and $V_i^+ \geq \max(||M^{-1}\boldsymbol{z}||, 2x)$ with the aforementioned $\delta = \frac{\varepsilon}{3\alpha + 2\beta}$ such that:

$$\text{PoA}(\mathcal{I}_\varepsilon) = \frac{U_{opt}(\mathcal{I}_\varepsilon)}{U_{eq}(\mathcal{I}_\varepsilon)} \geq \frac{\Omega(\varepsilon)}{O(\varepsilon^2)} = \Omega\left(\frac{1}{\varepsilon}\right) \to \infty \quad \text{as } \varepsilon \to 0 \tag{10}$$

$\square$

### B.3 PROOF OF THEOREM 3

*Proof.* We first note that if each agent has a hard constraint $V_i$, then regardless of their strategy, their idiosyncratic utility $f(\cdot)$ will be the same. Thus, it suffices to consider the objective of each agent $i$ as minimizing their cost:

$$c_i(\boldsymbol{h}_i, \boldsymbol{h}_{-i}) = \tfrac{1}{2}\boldsymbol{h}_i^T Q \boldsymbol{h}_i + \sum_{j \neq i} \boldsymbol{h}_i^T A \boldsymbol{h}_j + \boldsymbol{h}_i^T A \boldsymbol{s} \tag{11}$$

**Fact 1.** *For any positive integers $a, b$ and for any $\varepsilon > 0$: $2ab \leq \varepsilon a^2 + \frac{1}{\varepsilon}b^2$ (by AM-GM Inequality).*

**Definition 10** (Roughgarden (2015))**.** *For any two valid and individually rational strategy profile $\boldsymbol{H}^* = (\boldsymbol{h}_1^*, \ldots, \boldsymbol{h}_n^*)$ and $\boldsymbol{H} = (\boldsymbol{h}_1, \ldots, \boldsymbol{h}_n)$ of a cost-minimization game, the game is* smooth *if there exists constants $\lambda > 0$ and $\mu < 1$ such that:*

$$(LHS) \sum_i c_i(\boldsymbol{h}_i^*, \boldsymbol{h}_{-i}) \leq \lambda \sum_i c_i(\boldsymbol{H}^*) + \mu \sum_i c_i(\boldsymbol{H}) \ (RHS) \tag{12}$$

We now show that our game fits within the smooth games framework for any two valid strategies $\boldsymbol{H}^*, \boldsymbol{H}$. First, we recall that the matrix $A$ in the cost function is lower triangular. We define $A_s = \frac{1}{2}(A + A^T)$ as the symmetric version of this matrix. Observe that under the definition of matrix $A$ and $Q$, we have that $A_s = \frac{1}{2}Q$. In addition, let $A_a = A - A_s$ denote the remaining component, which we note by construction is always skew-symmetric. Finally, we define $\tilde{A}_a = Q^{-1/2}A_aQ^{-1/2}$, and $\kappa_A = ||\tilde{A}_a||_2$.

Let $c_{total}(\boldsymbol{H}) = \sum_i c_i(\boldsymbol{H})$. Further, let $\boldsymbol{z}_i = Q^{1/2}\boldsymbol{h}_i$ and $\boldsymbol{z}^* = Q^{1/2}\boldsymbol{h}_i^*$. Since $Q$ is symmetric and positive definite, we note that $Q^{1/2}$ is symmetric. Then observe that:

$$c_{total}(\boldsymbol{H}) = \boldsymbol{h}_i^T A \boldsymbol{s} + \sum_i \tfrac{1}{2}\boldsymbol{h}_i^T Q \boldsymbol{h}_i + \sum_{(i \neq j)} \boldsymbol{h}_i^T A \boldsymbol{h}_j$$

$$= \boldsymbol{h}_i^T A \boldsymbol{s} + \sum_i \tfrac{1}{2}\boldsymbol{h}_i^T Q \boldsymbol{h}_i + \sum_{i < j} \boldsymbol{h}_i^T A \boldsymbol{h}_j + \boldsymbol{h}_j^T A \boldsymbol{h}_i$$

$$= \boldsymbol{h}_i^T A \boldsymbol{s} + \tfrac{1}{2}\sum_{i=1}^n ||\boldsymbol{z}_i||_2^2 + \sum_{i < j} \boldsymbol{h}_i^T(A + A^T)\boldsymbol{h}_j = \boldsymbol{h}_i^T A \boldsymbol{s} + \tfrac{1}{2}\sum_{i=1}^n ||\boldsymbol{z}_i||_2^2 + \sum_{i < j} \boldsymbol{h}_i^T Q \boldsymbol{h}_j$$

$$= \boldsymbol{h}_i^T A \boldsymbol{s} + \tfrac{1}{2}\sum_{i=1}^n ||\boldsymbol{z}_i||_2^2 + \sum_{i < j} \boldsymbol{z}_i^T \boldsymbol{z}_j$$

Importantly, since all strategy vectors $\boldsymbol{h}_i$ are positive and $\boldsymbol{s}$ is positive, we can state the following for any two joint strategies $\boldsymbol{H}^*$ and $\boldsymbol{H}$:

$$c_{total}(\boldsymbol{H}^*) \geq \boldsymbol{h}_i^{*T} A \boldsymbol{s} + \tfrac{1}{2}\sum_{i=1}^n ||\boldsymbol{z}_i^*||_2^2 \quad \text{and} \quad c_{total}(\boldsymbol{H}) \geq \tfrac{1}{2}\sum_{i=1}^n ||\boldsymbol{z}_i||_2^2 \tag{13}$$

As for the (LHS), we observe the following:

$$
\text{LHS} = \boldsymbol{h}_i^{*T} A \boldsymbol{s} + \tfrac{1}{2} \sum_{i=1}^n \boldsymbol{h}_i^{*T} Q \boldsymbol{h}_i^* + \sum_{i=1} \sum_{i \neq j} \boldsymbol{h}_i^{*T} A \boldsymbol{h}_j
$$

$$
= \boldsymbol{h}_i^{*T} A \boldsymbol{s} + \tfrac{1}{2} \sum_{i=1}^n ||\boldsymbol{z}_i^*||^2 + \sum_{i=1} \sum_{j \neq i} \boldsymbol{h}_i^{*T} (A_s + A_a) \boldsymbol{h}_j
$$

$$
= \boldsymbol{h}_i^{*T} A \boldsymbol{s} + \tfrac{1}{2} \sum_{i=1}^n ||\boldsymbol{z}_i^*||^2 + \tfrac{1}{2} \sum_{i=1}^n \sum_{j \neq i} \boldsymbol{h}_i^{*T} Q \boldsymbol{h}_j + \sum_{i=1}^n \sum_{j \neq i} \boldsymbol{z}_i^{*T} Q^{-1/2} A_a Q^{-1/2} \boldsymbol{z}_j
$$

$$
= \boldsymbol{h}_i^{*T} A \boldsymbol{s} + \tfrac{1}{2} \sum_{i=1}^n ||\boldsymbol{z}_i^*||^2 + \tfrac{1}{2} \sum_{i=1}^n \sum_{j \neq i} \boldsymbol{z}_i^{*T} \boldsymbol{z}_j + \sum_{i=1}^n \sum_{i \neq j} \boldsymbol{z}_i^{*T} \tilde{A}_a \boldsymbol{z}_j
$$

$$
\leq \boldsymbol{h}_i^{*T} A \boldsymbol{s} + \tfrac{1}{2} \sum_{i=1}^n ||\boldsymbol{z}_i^*||^2 + \tfrac{1}{2} \sum_{i=1}^n \sum_{j \neq i} ||\boldsymbol{z}_i^*|| \cdot ||\boldsymbol{z}_j|| + \sum_{i=1}^n \sum_{i \neq j} ||\boldsymbol{z}_i^*|| \cdot ||\boldsymbol{z}_j|| \cdot \kappa_A
$$

$$
\leq \boldsymbol{h}_i^{*T} A \boldsymbol{s} + \tfrac{1}{2} \sum_{i=1}^n ||\boldsymbol{z}_i^*||^2 + (\tfrac{1}{2} + \kappa_A) \sum_{i=1}^n \sum_{j \neq i} ||\boldsymbol{z}_i^*|| \cdot ||\boldsymbol{z}_j||
$$

$$
\leq \boldsymbol{h}_i^{*T} A \boldsymbol{s} + \tfrac{1}{2} \sum_{i=1}^n ||\boldsymbol{z}_i^*||^2 + (\tfrac{1}{2} + \kappa_A) \sum_{i=1}^n \sum_{j \neq i} \left( \tfrac{\varepsilon}{2} ||\boldsymbol{z}_i^*||^2 + \tfrac{1}{2\varepsilon} ||\boldsymbol{z}_j||^2 \right) \quad \text{(due to Fact 1)}
$$

$$
\leq \boldsymbol{h}_i^{*T} A \boldsymbol{s} + \tfrac{1}{2} \sum_{i=1}^n ||\boldsymbol{z}_i^*||^2 + (\tfrac{1}{2} + \kappa_A) \sum_{i=1}^n \sum_{j \neq i} \left( \tfrac{\varepsilon}{2} ||\boldsymbol{z}_i^*||^2 + \tfrac{1}{2\varepsilon} ||\boldsymbol{z}_j||^2 \right)
$$

$$
\leq \boldsymbol{h}_i^{*T} A \boldsymbol{s} + \tfrac{1}{2} \underbrace{\left[ 1 + \varepsilon(0.5 + \kappa_A)(n-1) \right]}_{\lambda > 1} \sum_{i=1}^n ||\boldsymbol{z}_i^*||^2 + \tfrac{1}{2} \underbrace{\frac{(0.5 + \kappa_A)(n-1)}{\varepsilon}}_{\mu} \sum_{i=1}^n ||\boldsymbol{z}_i||^2
$$

$$
\leq \lambda \left[ \boldsymbol{h}_i^{*T} A \boldsymbol{s} + \tfrac{1}{2} \sum_{i=1}^n ||\boldsymbol{z}_i^*||^2 \right] + \mu \tfrac{1}{2} \sum_{i=1}^n ||\boldsymbol{z}_i||^2 \leq \lambda c_{total}(\boldsymbol{H}^*) + \mu c_{total}(\boldsymbol{H}) = \text{RHS}
$$

We note that for smooth games, $\mu < 1$, and we thus need to set $(n-1)(0.5 + \kappa_A) < \varepsilon$. From Roughgarden (2015), we know that the PoA in smooth games is upper bounded by $\frac{\lambda}{1-\mu}$. Letting $a = (0.5 + \kappa_A)$, we have that the PoA as a function of $\varepsilon$ is:

$$
\text{PoA}_\varepsilon = \frac{\varepsilon(1 + \varepsilon a(n-1))}{\varepsilon - a(n-1)} \tag{14}
$$

By taking the derivative and solving for $\varepsilon$ to get the critical point, we have that :

$$
\varepsilon^* = (n-1)a + \sqrt{(n-1)^2 a^2 + 1} \leq (n-1)a + 1 + \sqrt{(n-1)^2 a^2} \leq 2a(n-1) + 1.
$$

Letting $b = a(n-1)$ – and thus $\varepsilon^* = 2b + 1$ – we can plug in this expression of $\varepsilon^*$ to our PoA. We get:

$$
\text{PoA} = \frac{(2b+1)(1 + b(2b+1))}{2b+1-b} = \frac{(2b+1)(2b^2 + b + 1)}{(b+1)} = \frac{4b^3 + 4b^2 + 3b + 1}{b+1}
$$

$$
= 4b^2 + 3 - \frac{2}{b-1} \leq 4b^2 + 3 \leq 4a^2(n-1)^2 + 3 \leq (1 + 2\kappa_A)^2 (n-1)^2 + 3
$$

Lastly, we note that: $||A_a||_2 \leq \sqrt{||A_a||_1 \cdot ||A_a||_\infty} = (T-1)\frac{\alpha}{2}$. Then we have that:

$$
\kappa_A = ||Q^{-1/2} A Q^{-1/2}||_2 \leq ||Q^{-1/2}||_2^2 ||A_a||_2 \leq \frac{T-1}{2} \frac{\alpha}{\alpha + \beta} = \frac{T-1}{2} \gamma \tag{15}
$$

Plugging everything in, we have that:

$$
\text{PoA} \leq (1 + \gamma(T-1))^2 (n-1)^2 + 3 = O(n^2 T^2 \gamma^2) \tag{16}
$$

$\square$

## C    PROOFS AND DETAILS FOR SECTION 4

### C.1    PROOF OF LEMMA 3

*Proof.* In the most general sense, observe that agent $i$'s best response for a realized type $\theta_i$ allows them to play a mixed strategy over all valid strategies: $p_i(\boldsymbol{h}_i|\theta_i)$, where $\boldsymbol{h}_i$ is a vector in $\mathbb{R}^T$ since the probability is already conditioned on $\theta_i$. Suppose the remaining agents are playing some mixed, possibly correlated strategy $\sigma_{-i}$, where $\sigma_{-i}(\boldsymbol{h}_{-i}|\theta_{-i})$ denotes the probability that the remaining agents play strategy $\boldsymbol{h}_{-i} \in R_{-i}$ when their joint type realization is some $\theta_{-i}$. We can then express agent $i$'s best response problem as follows (we use $G_i$ to denote $G_i(\theta_i)$):

$$\mathrm{br}_i(\theta_i, \sigma_{-i}) = \operatorname*{arg\,max}_{p_i(\boldsymbol{h}_i|\theta_i) \in \Delta(G_i)} \int_{\boldsymbol{h}_i} p_i(\boldsymbol{h}_i; \theta_i) \sum_{\theta_{-i}} \int_{\boldsymbol{s}, \alpha, \beta} P(\theta_{-i}, \boldsymbol{s}, \alpha, \beta|\theta_i) \int_{\boldsymbol{h}_{-i}} \sigma_{-i}(\boldsymbol{h}_{-i}|\theta_{-i}) u_i(\boldsymbol{h}_i; \boldsymbol{h}_{-i}, \boldsymbol{\lambda})$$

The linearity of the integral and the fact that $\int_{\boldsymbol{h}_i} p_i(\boldsymbol{h}_1|\theta_i) d\boldsymbol{h}_i = 1$ means that a maximum must exists at a vertex/pure strategy. If multiple pure strategies are optimal, then any linear combination (a mixed strategy) would also be a best-response. However, if there is a unique pure strategy maximizing this, then it means any mixed strategy must be strictly sub-optimal. In other words, it suffices to show that the pure-strategy best-response is unique even when others' play mixed and correlated strategies. This pure best-response problem is given by:

$$\mathrm{br}_i(\theta_i, \sigma_{-i}) = \operatorname*{arg\,max}_{\boldsymbol{h}_i \in G_i} \sum_{\theta_{-i}} \int_{\boldsymbol{\lambda}} P(\theta_{-i}, \boldsymbol{\lambda}|\theta_i) \int_{\boldsymbol{h}_{-i}} \sigma_{-i}(\boldsymbol{h}_{-i}|\theta_{-i}) u_i(\boldsymbol{h}_i; \boldsymbol{h}_{-i}, \boldsymbol{\lambda})$$

$$= \operatorname*{arg\,max}_{\boldsymbol{h}_i \in G_i} \sum_{\theta_{-i}} \int_{\boldsymbol{\lambda}} P(\theta_{-i}, \boldsymbol{\lambda}|\theta_i) \int_{\boldsymbol{h}_{-i}} \sigma_{-i}(\boldsymbol{h}_{-i}|\theta_{-i}) \left[ f_i(\boldsymbol{h}_i) - \boldsymbol{h}_i^T Q \boldsymbol{h}_i - \sum_{j \neq i} \boldsymbol{h}_j^T A \boldsymbol{h}_i - \boldsymbol{s} B \boldsymbol{h}_i \right]$$

$$= \operatorname*{arg\,max}_{\boldsymbol{h}_i \in G_i} \int_{f_i \in F} f_i(\boldsymbol{h}_i) d\mu(f_i) - \boldsymbol{h}_i^T Q \boldsymbol{h}_i$$

$$- \underbrace{\left[ \sum_{\theta_{-i}} \int_{\boldsymbol{s}, \alpha, \beta} P(\theta_{-i}, \boldsymbol{s}, \alpha, \beta|\theta_i) \int_{\boldsymbol{h}_{-i}} \sigma_{-i}(\boldsymbol{h}_{-i}|\theta_{-i}) \sum_{j \neq 1} \boldsymbol{h}_j^T A + \boldsymbol{s} B \right]}_{\boldsymbol{w}^T(\cdot)} \boldsymbol{h}_i$$

$$= \operatorname*{arg\,max}_{\boldsymbol{h}_i \in G_i} f_i^*(\boldsymbol{h}_i) - \boldsymbol{h}_i^T Q \boldsymbol{h}_i - \boldsymbol{w}^T(\cdot) \boldsymbol{h}_i$$

where $\mu(f_i)$ is any finite non-negative measure on the function space $F$, and $f^*$ is the result of the integral. The concavity of the function class $F$ and non-negativity of measure $\mu$ ensure that $f^*$ is concave Rockafellar & Wets (2009). Next, we note that $\boldsymbol{w}^T(\cdot)$ is a $T$ dimensional vector that does not depend on the $\boldsymbol{h}_i$. Thus, the objective faced by buyer $i$ is strictly concave (since $Q$ is a PD matrix – see Lemma 2) and there is a unique solution. This immediately implies that a mixed strategy will always be a sub-optimal best-response. $\qquad\square$

### C.2    PROOF OF THEOREM 4

*Proof.* As in Theorem 1, we express the results from a minimization perspective. That is, each agent's best-response for type realization $\theta_i$ is: $\operatorname*{arg\,min}_{\boldsymbol{h}_i \in G_i} \mathbb{E}_{\boldsymbol{\theta}_{-i}, \boldsymbol{\lambda}|\theta_i}[c_i(\boldsymbol{h}_i, \boldsymbol{h}_{-i}, \boldsymbol{\lambda})]$, where $c_i(\boldsymbol{h}_i, \boldsymbol{h}_{-i}, \boldsymbol{\lambda}) = -u_i(\boldsymbol{h}_i, \boldsymbol{h}_{-i}, \boldsymbol{\lambda})$. Also from 1, we note that the necessary and sufficient conditions for an $n$ agent BNE with $k$ discrete types and pure strategies for each type, can be interpreted as follows:

$$\forall i \in [n], \forall \theta_i \in [k], \forall \boldsymbol{h}' \in \mathcal{H}_i : \mathbb{E}_{\boldsymbol{\theta}_{-i}, \boldsymbol{\lambda}}[c_i(\boldsymbol{h}_i^{eq}(\theta_i), \boldsymbol{h}_{-i}^{eq}(\boldsymbol{\theta}_{-i}), \boldsymbol{\lambda}) - c_i(\boldsymbol{h}', \boldsymbol{h}_{-i}^{eq}(\boldsymbol{\theta}_{-i}), \boldsymbol{\lambda})] \leq 0$$

Since expected utility is a smooth function, as in the Nash setting, the simultaneous conditions can be expressed as a variational inequality of the cost derivatives: there can exist no feasible direction at equilibrium at which cost is decreasing. Importantly, this characterization is exact even if the derivatives are scaled by a distinct constant. Formally, a set of strategies are at a BNE if and only if

the following holds for any choice of $\gamma_{i\ell}$ – we will choose $\gamma_{i,\ell} = P(\theta_i)$, the marginal probability of an agent $i$ being of type $\theta_i \in [k]$ – recall $\boldsymbol{h}_i(\theta_i) \in G_i(\theta_i)$ is the strategy used upon realization $\theta_i$:

$$\forall i \in [n], \forall \theta_i \in [k], \forall \boldsymbol{h}'(\theta_i) \in G_i : \langle \gamma_{i,\theta_i} \nabla_{\boldsymbol{h}_i(\theta_i)} \mathbb{E}_{\boldsymbol{\theta}_{-i},\boldsymbol{\lambda}}[c_i(\boldsymbol{h}_i^{eq}(\theta_i), \boldsymbol{h}_{-i}^{eq}(\boldsymbol{\theta}_{-i}), \boldsymbol{\lambda})], (\boldsymbol{h}'(\theta_i) - \boldsymbol{h}_i^{eq}(\theta_i)) \rangle \geq 0$$

With our choice of scaling $\gamma_{i,\ell}$, and switching the order of gradients and expectation, we have that for any $i, \theta_i, \gamma_{i,\theta_i} \nabla_{\boldsymbol{h}_i(\theta_i)} \mathbb{E}_{\boldsymbol{\theta}_{-i},\boldsymbol{\lambda}}[c_i(\boldsymbol{h}_i^{eq}(\theta_i), \boldsymbol{h}_{-i}^{eq}(\boldsymbol{\theta}_{-i}), \boldsymbol{\lambda})]$

$$= P(\theta_i) \Bigg( \sum_{\boldsymbol{\theta}_{-i}} \int_{\boldsymbol{\lambda}} Q_{\alpha,\beta} \boldsymbol{h}_i(\theta_i) P(\boldsymbol{\theta}_{-i}, \boldsymbol{\lambda}|\theta_i)$$

$$+ \sum_{\boldsymbol{\theta}_{-i}} \int_{\boldsymbol{\lambda}} \Bigg[ \sum_{j \neq i} A_{\alpha,\beta} \boldsymbol{h}_j(\theta_j) + B_{\alpha,\beta} s \Bigg] P(\boldsymbol{\theta}_{-i}, \boldsymbol{\lambda}|\theta_i) - \nabla_{\boldsymbol{h}_i(\theta_i)} \int_{\boldsymbol{\lambda}} f_i(\boldsymbol{h}_i(\theta_i)) d\mu(f_i|\theta_i) \Bigg)$$

$$= P(\theta_i) \Bigg[ \int_{\alpha,\beta} Q_{\alpha,\beta} P(\alpha, \beta|\theta_i) \Bigg] \boldsymbol{h}_i(\theta_i) + P(\theta_i) \sum_{j \neq i} \sum_{\theta_j} \int_{\alpha,\beta} A_{\alpha,\beta} \boldsymbol{h}_j(\theta_j) \sum_{\boldsymbol{\theta}_{-(i,j)}} \int_s P(\theta_j, \boldsymbol{\theta}_{-(i,j)}, \boldsymbol{\lambda}|\theta_i)$$

$$+ \underbrace{\int_{\alpha,\beta,s} B_{\alpha,\beta} s P(\boldsymbol{\lambda}, \theta_i)}_{b_{i,\theta_i}} - P(\theta_i) \nabla_{\boldsymbol{h}_i(\theta_i)} \underbrace{\int_{f_i \in F} f_i(\boldsymbol{h}_i(\theta_i)) d\mu(f_i|\theta_i)}_{f_{i,\theta_i}^*(\boldsymbol{h}_i(\theta_i))}$$

$$= P(\theta_i) \Bigg[ \underbrace{\int_{\alpha,\beta} Q_{\alpha,\beta} P(\alpha, \beta|\theta_i)}_{Q_{i,\theta_i}^* \in \mathbb{R}^{T \times T}} \Bigg] \boldsymbol{h}_i(\theta_i) + \sum_{j \neq i} \sum_{\theta_j} P(\theta_j, \theta_i) \Bigg[ \underbrace{\int_{\alpha,\beta} A_{\alpha,\beta} P(\alpha, \beta|\theta_j, \theta_i)}_{A_{i,\theta_i,j,\theta_j}^* \in \mathbb{R}^{T \times T}} \Bigg] \boldsymbol{h}_j(\theta_j)$$

$$+ b_{i,\theta_i} - P_i(\theta_i) \cdot \nabla_{\boldsymbol{h}_i(\theta_i)} f_{i,\theta_i}^*(\boldsymbol{h}_i(\theta_i))$$

where in the last transition, we use the fact that:

$$P(\theta_j, \alpha, \beta|\theta_i) \cdot P(\theta_i) = P(\alpha, \beta, \theta_j, \theta_i) = P(\alpha, \beta|\theta_i, \theta_j) P(\theta_i, \theta_j)$$

We note that $\mu(f_i|\theta_i)$ is a finite non-negative measure on the function space $F$, and $f_{i,\theta_i}^*$ is the result of the functional integral. The concavity of the function class $F$ and non-negativity of measure $\mu$ ensure that $f_{i,\theta_i}^*$ is a concave function Rockafellar & Wets (2009). Next, let $k_p = \prod_{i=1}^n k_i$ and define $\boldsymbol{x} = [\boldsymbol{h}_{1,1}, \ldots, \boldsymbol{h}_{1,k_1}, \ldots, \boldsymbol{h}_{n,1}, \ldots, \boldsymbol{h}_{n,k_n}] \in \mathbb{R}^{nk_p}$ denote a complete strategy profile (strategy for each agent for each type). At a high-level, our goal is to show that this operator, denoted by $F$, is strictly monotone, which implies the uniqueness of the solution to the equilibrium variational inequality. That is, we want to show that for all $\boldsymbol{x}$, $\langle F(\boldsymbol{x}) - F(\boldsymbol{x}'), (\boldsymbol{x} - \boldsymbol{x}') \rangle \geq m||\boldsymbol{x} - \boldsymbol{x}'||^2$.

We can write this operator as follows: $F(\boldsymbol{x}) = M_{\text{bayes}} \boldsymbol{x} + \boldsymbol{b} - J(\boldsymbol{x})$, where $b \in \mathbb{R}^{k_p T}$ has $b_{i,\theta_i} \in \mathbb{R}^T$ as index $(i, \theta_i)$. Observe that this vector is a constant with respect to the agent strategies. Similarly, $J(\boldsymbol{x}) \in \mathbb{R}^{k_p T}$, where at index $i, \theta_i$, we have $P(\theta_i) \nabla_{\boldsymbol{h}_i(\theta_i)} f_{i,\theta_i}^*(\boldsymbol{h}_i(\theta_i)) \in \mathbb{R}^T$. As for the $M_{\text{bayes}} \in \mathbb{R}^{k_p T \times k_p T}$, we can write this as an $n \times n$ block matrix, where each block $(i \in [n], j \in [n])$ is a $k_i T \times k_j T$ matrix), defined as follows:

$$M_{\text{bayes}}^{ii} = \begin{bmatrix} P(\theta_i = 1)Q_{i,1}^* & 0 & \ldots & 0 \\ 0 & P(\theta_i = 2)Q_{i,2}^* & \ldots & 0 \\ \vdots & \vdots & \ldots & \vdots \\ 0 & 0 & \ldots & P(\theta_i = k)Q_{i,k}^* \end{bmatrix}$$

$$M_{\text{bayes}}^{ij} = \begin{bmatrix} P(\theta_i = 1, \theta_j = 1)A_{i,1,j,1}^* & P(\theta_i = 1, \theta_j = 2)A_{i,1,j,2}^* & \ldots & P(\theta_i = 1, \theta_j = k_j)A_{i,1,j,k}^* \\ P(\theta_i = 2, \theta_j = 1)A_{i,2,j,1}^* & P(\theta_i = 2, \theta_j = 2)A_{i,2,j,2}^* & \ldots & P(\theta_i = 2, \theta_j = k_j)A_{i,2,j,k}^* \\ \vdots & \vdots & \ldots & \vdots \\ P(\theta_i = k_i, \theta_j = 1)A_{i,k,j,1}^* & P(\theta_i = k, \theta_j = 2)A_{i,k,j,2}^* & \ldots & P(\theta_i = k_i, \theta_j = k_j)A_{i,k,j,k}^* \end{bmatrix}$$

Observe that each $f_{i,\theta_i}^*$ is a concave function. Since for all convex functions, their gradient is a monotone operator, it is immediate that $-\langle J(\boldsymbol{x}) - J(\boldsymbol{x}') \rangle \geq 0$. And since $\boldsymbol{b}$ is a constant, it suffices

to show that the matrix $M$ is positive definite. That is, we want to show that $\boldsymbol{x}^T M \boldsymbol{x} \geq m||\boldsymbol{x}||^2$. Observe that:

$$
\begin{aligned}
\boldsymbol{x}^T M_{\text{bayes}} \boldsymbol{x} &= \sum_{i=1}^n \sum_{\theta_i} P(\theta_i) \boldsymbol{h}_i^T(\theta_i) Q_{i,\theta_i}^* \boldsymbol{h}_i(\theta_i) + \sum_{i \neq j} \sum_{\theta_i, \theta_j} P_{ij}(\theta_i, \theta_j) \boldsymbol{h}_i^T(\theta_i) A_{i,\theta_i,j,\theta_j} \boldsymbol{h}_j(\theta_j) \\
&= \sum_{i=1}^n \sum_{\theta_i} \int_{\alpha,\beta} \boldsymbol{h}_i^T(\theta_i) Q_{\alpha,\beta} \boldsymbol{h}_i(\theta_i) \sum_{\boldsymbol{\theta}_{-i}} P(\theta_i, \boldsymbol{\theta}_{-i}, \alpha, \beta) \\
&\quad + \sum_{i \neq j} \sum_{\theta_i, \theta_j} \int_{\alpha,\beta} \boldsymbol{h}_i^T(\theta_i) A_{\alpha,\beta} \boldsymbol{h}_j(\theta_j) \sum_{\boldsymbol{\theta}_{-(i,j)}} P(\theta_i \theta_j, \boldsymbol{\theta}_{-(i,j)} \alpha, \beta) \\
&= \sum_{\boldsymbol{\theta}} \int_{\alpha,\beta} P(\boldsymbol{\theta}, \alpha, \beta) \left[ \sum_{i=1}^n \boldsymbol{h}_i^T(\theta_i) Q_{\alpha,\beta} \boldsymbol{h}_i(\theta_i) + \sum_{j \neq i} \boldsymbol{h}_i^T(\theta_i) A_{\alpha,\beta} \boldsymbol{h}_j(\theta_j) \right] \\
&= \mathbb{E}_{\boldsymbol{\theta},\alpha,\beta} \left[ \sum_{i=1} \boldsymbol{z}_{i,\theta_i}^T Q_{\alpha,\beta} \boldsymbol{z}_{i,\theta_i} + \sum_{i \neq j} \boldsymbol{z}_{i,\theta_i}^T A_{\alpha,\beta} \boldsymbol{z}_{j,\theta_j} \right] = \mathbb{E}_{\boldsymbol{\theta},\alpha,\beta}[\boldsymbol{z}_{\boldsymbol{\theta}}^T M_{\alpha,\beta} \boldsymbol{z}_{\boldsymbol{\theta}}]
\end{aligned}
$$

where $\boldsymbol{z}_{i,\theta_i} = \sum_\ell \mathbb{1}[\theta_i = \ell] \boldsymbol{h}_i(\ell)$ is a random vector of length $T$ and for a realization $\boldsymbol{\theta} \in [k]^n$, $\boldsymbol{z}_{\boldsymbol{\theta}} = [\boldsymbol{z}_{1,\theta_1}, \ldots, \eta_{n,\theta_n}]^T$ is a concatenation of these $n$ random vectors (of size $nT$). Further, $M_{\alpha,\beta}$ is a random matrix which, for any realization of $\alpha, \beta$, is the same as the $M$ matrix used in the complete information setting. From Theorem 1, we also note that for any $\alpha, \beta$, the symmetric component of $M_{\alpha,\beta}$, denoted $M_{\alpha,\beta}^s$ is positive definite; thus, by choosing $c = \lambda_{min}(M_{\alpha,\beta}^s)$ ensures the strong monotonicity condition on the operator $M_{\alpha,\beta}$. Thus, for any $\boldsymbol{z}_{\boldsymbol{\theta}}$ and any realization realization of $(\alpha, \beta)$, there exists a $c_{\alpha,\beta}$ such that $\boldsymbol{z}_{\boldsymbol{\theta}}^T M_{\alpha,\beta} \boldsymbol{z}_{\boldsymbol{\theta}} \geq c_{\alpha,\beta}||\boldsymbol{z}_{\boldsymbol{\theta}}||^2$, when $\boldsymbol{z}_{\boldsymbol{\theta}} \neq \boldsymbol{0}$.

To determine a uniform bound on $c$ across the randomness of $(\boldsymbol{\theta}, \alpha, \beta)$, let each agent's type realization $\theta_i = \ell$ occur with non-zero probability[5]. Then letting $P_{min} = \min_{i,\ell} P(\theta_i = \ell)$ be the smallest probability, and $c_{min} = \min_{\alpha,\beta} \lambda_{min}(M_{\alpha,\beta}^s)$ the smallest eigenvalue possible in the distribution support of $\alpha, \beta$:

$$
\begin{aligned}
\boldsymbol{x}^T M_{\text{bayes}} \boldsymbol{x} &= \mathbb{E}_{\boldsymbol{\theta},\alpha,\beta}[\boldsymbol{z}_{\boldsymbol{\theta}}^T M_{\alpha,\beta} \boldsymbol{z}_{\boldsymbol{\theta}}] = \sum_{\boldsymbol{\theta}|\boldsymbol{z}_{\boldsymbol{\theta}} \neq \boldsymbol{0}} \int_{\alpha,\beta} P(\boldsymbol{\theta}, \alpha, \beta) c_{\alpha,\beta} ||\boldsymbol{z}_{\boldsymbol{\theta}}||^2 \\
&\geq c_{min} \sum_{i=1}^n \sum_{\ell=1}^k P(\theta_i = \ell) ||\boldsymbol{z}_{i,\theta_i}||^2 = \underbrace{c_{min} P_{min}}_{m} ||\boldsymbol{x}||^2
\end{aligned}
$$

We recall from Theorem 1 that for any $c$-strongly monotone and $L$-Lipshictz operator $F$, the extra gradient algorithm converges linearly to the unique solution of the variational inequality. We have already shown the operator to be $c$-strongly monotone. Further, since the operator is of the form $M_{\text{bayes}} \boldsymbol{x} + b - J(\boldsymbol{x})$, it suffices to show Lipschitzness of each term. The linear operator $M_{\text{bayes}}$ is always Lipschitz, with the constant depending on the norm of this matrix. Since each $f \in F$ is smooth, $J(\boldsymbol{x})$ is composed of the gradient of some smooth concave function. Therefore, this is also Lipschitz, with the constant depending on the Hessian of this function. □

## C.3 EXPERIMENTAL SETUP

Our experimental setting for the Bayesian Simulations is as follows. There are 2 agents and 3 possible types for each agent. The type of an agent $i$, $\theta_i$ is a positive real number that is equal to the constraint. That is, an agent $i$ of type $\theta_i$ has a constraint $-\theta_i \leq \boldsymbol{1}^T \boldsymbol{h}_i(\theta_i) \leq \theta_i$. We have $\theta_1 \in [10, 15, 20]$ and $\theta_2 \in [20, 25, 30]$. The joint type distribution $P(\theta_1, \theta_2)$ is uniform over the 9 possible outcomes.

---

[5]Note that if a type realization occurs with probability 0, it can be removed from the support without loss of generality.

Each agent's idiosyncratic utility $f_i$ is a linear function: $f_i(\boldsymbol{h}_i(\theta_i)) = r_i \boldsymbol{1}^T \boldsymbol{h}_i(\theta_i)$. The linearity of this function means it suffices to consider $\mathbb{E}[r_i|\theta_i]$. For agent 1, the type conditioned expected reserves are $(3, 5, 7)$, and for agent 2, we set $(6, 8, 10)$.

Lastly, the variational inequality characterizing the BNE has linear dependence on the $\alpha, \beta$. As such, it suffices to consider the expected value of these market parameters, conditioned on the joint type realization. We set $\mathbb{E}[\alpha|\theta_1, \theta_2] = 0.1$ and $\mathbb{E}[\beta|\theta_1, \theta_2] = \frac{c}{400}(\theta_1 + \theta_2)$, where we have $c = 1, 10, 100$ for the left, middle and right panel. These numbers were chosen to ensure the $\beta$ values were in a comparable range to those used in the experiments for Section 3. This exact setup, with $c = 10$, is used for the online learning experiments for Section 5.

# D    PROOFS AND DETAILS FOR SECTION 5

In what follows we primarily use the cost notation $c_i(\cdot)$ (recall that $c_i(\boldsymbol{h}_i(\theta_i); \boldsymbol{h}_{-i}(\theta_{-i}), \boldsymbol{\lambda}) = -u_i(\boldsymbol{h}_i(\theta_i); \boldsymbol{h}_{-i}(\theta_{-i}), \boldsymbol{\lambda}))$.

## D.1    PROOF OF THEOREM 5

**Proposition 1.** *Let $P$ be a joint distribution over game instances $\mathcal{I}$. Let $\sigma \in \Delta(\mathcal{H}_1 \times \cdots \times \mathcal{H}_n)$ be a distribution over strategy profiles. Suppose $\sigma$ satisfies for all $i$, for all $\theta_i$, for all $\boldsymbol{h}_i'(\theta_i)$:*

$$\mathbb{E}_{\boldsymbol{h} \sim \sigma} \mathbb{E}_{\theta, \boldsymbol{\lambda} \sim P} \big[ c_i(\boldsymbol{h}_i(\theta_i), \boldsymbol{h}_{-i}(\theta_{-i}), \boldsymbol{\lambda}) - c_i(\boldsymbol{h}_i'(\theta_i), \boldsymbol{h}_{-i}(\theta_{-i}), \boldsymbol{\lambda}) \big] \leq \epsilon.$$

*Then, $\sigma$ is an approximate Bayesian coarse correlated equilibrium, satisfying for all $i$, for all $\theta_i$, for all $\boldsymbol{h}_i'(\theta_i)$:*

$$\mathbb{E}_{\boldsymbol{h} \sim \sigma} \mathbb{E}_{\theta_{-i}, \boldsymbol{\lambda} \sim P \mid \theta_i} \big[ c_i(\boldsymbol{h}_i(\theta_i), \boldsymbol{h}_{-i}(\theta_{-i}), \boldsymbol{\lambda}) - c_i(\boldsymbol{h}_i'(\theta_i), \boldsymbol{h}_{-i}(\theta_{-i}), \boldsymbol{\lambda}) \big] \leq \frac{\epsilon}{\Pr(\theta_i)}.$$

*Notice that when $\sigma$ is a singleton distribution, this corresponds to an approximate Bayesian Nash equilibrium.*

*Proof.* We show the contrapositive. Suppose for some agent $i$, there is a type $\theta_i'$ and action $\boldsymbol{h}_i'(\theta_i')$ such that

$$\mathbb{E}_{\boldsymbol{h} \sim \sigma} \mathbb{E}_{\theta_{-i}, \boldsymbol{\lambda} \sim P \mid \theta_i'} \big[ c_i(\boldsymbol{h}_i(\theta_i'), \boldsymbol{h}_{-i}(\theta_{-i}), \boldsymbol{\lambda}) - c_i(\boldsymbol{h}_i'(\theta_i'), \boldsymbol{h}_{-i}(\theta_{-i}), \boldsymbol{\lambda}) \big] > \frac{\epsilon}{\Pr(\theta_i')}.$$

For each $\theta_i$, define

$$\boldsymbol{h}_i^*(\theta_i) \in \arg\min_{\boldsymbol{h}_i} \mathbb{E}_{\boldsymbol{h} \sim \sigma} \mathbb{E}_{\theta_{-i}, \boldsymbol{\lambda} \sim P \mid \theta_i} \big[ c_i(\boldsymbol{h}_i, \boldsymbol{h}_{-i}(\theta_{-i}), \boldsymbol{\lambda}) \big].$$

By optimality of $\boldsymbol{h}_i^*(\theta_i')$,

$$\mathbb{E}_{\boldsymbol{h} \sim \sigma} \mathbb{E}_{\theta_{-i}, \boldsymbol{\lambda} \sim P \mid \theta_i'} \big[ c_i(\boldsymbol{h}_i^*(\theta_i'), \boldsymbol{h}_{-i}(\theta_{-i}), \boldsymbol{\lambda}) \big] \leq \mathbb{E}_{\boldsymbol{h} \sim \sigma} \mathbb{E}_{\theta_{-i}, \boldsymbol{\lambda} \sim P \mid \theta_i'} \big[ c_i(\boldsymbol{h}_i'(\theta_i'), \boldsymbol{h}_{-i}(\theta_{-i}), \boldsymbol{\lambda}) \big].$$

Thus,

$$\mathbb{E}_{\boldsymbol{h} \sim \sigma} \mathbb{E}_{\theta_{-i}, \boldsymbol{\lambda} \sim P \mid \theta_i'} \big[ c_i(\boldsymbol{h}_i(\theta_i'), \boldsymbol{h}_{-i}(\theta_{-i}), \boldsymbol{\lambda}) - c_i(\boldsymbol{h}_i^*(\theta_i'), \boldsymbol{h}_{-i}(\theta_{-i}), \boldsymbol{\lambda}) \big] > \frac{\epsilon}{\Pr(\theta_i')}.$$

Now consider the gain by a unilateral deviation to $\boldsymbol{h}_i^*(\theta_i)$ for all $\theta_i$:

$$\mathbb{E}_{\boldsymbol{h} \sim \sigma} \mathbb{E}_{\theta, \boldsymbol{\lambda} \sim P} \big[ c_i(\boldsymbol{h}_i(\theta_i), \boldsymbol{h}_{-i}(\theta_{-i}), \boldsymbol{\lambda}) - c_i(\boldsymbol{h}_i^*(\theta_i), \boldsymbol{h}_{-i}(\theta_{-i}), \boldsymbol{\lambda}) \big]$$
$$= \sum_{\theta_i} \Pr(\theta_i) \, \mathbb{E}_{\boldsymbol{h} \sim \sigma} \mathbb{E}_{\theta_{-i}, \boldsymbol{\lambda} \sim P \mid \theta_i} \big[ c_i(\boldsymbol{h}_i(\theta_i), \boldsymbol{h}_{-i}(\theta_{-i}), \boldsymbol{\lambda}) - c_i(\boldsymbol{h}_i^*(\theta_i), \boldsymbol{h}_{-i}(\theta_{-i}), \boldsymbol{\lambda}) \big].$$

By optimality of $\boldsymbol{h}_i^*(\theta_i)$ for every $\theta_i$, each summand is non-negative. Furthermore, the summand corresponding to $\theta_i'$ exceeds $\Pr(\theta_i') \cdot \frac{\epsilon}{\Pr(\theta_i')} = \epsilon$. Hence the whole sum is $> \epsilon$, contradicting the hypothesis. $\qquad\square$

**Lemma 4.** *For all $i \in [n]$, let $V_i(\boldsymbol{h}) = \nabla_{\boldsymbol{h}_i} \ell_i(\boldsymbol{h}_i, \boldsymbol{h}_{-i}; P) = \nabla_{\boldsymbol{h}_i} \mathbb{E}_{\theta, \boldsymbol{\lambda} \sim P}[c_i(\boldsymbol{h}_i(\theta_i), \boldsymbol{h}_{-i}(\theta_{-i}), \boldsymbol{\lambda})] \in \mathbb{R}^{k_i \times T}$ and $V(\boldsymbol{h}) = (V_1(\boldsymbol{h}), ..., V_n(\boldsymbol{h})) \in \mathbb{R}^{n \times k \times T}$. The operator $V$ is $m$-strongly monotone, i.e. $\langle V(\boldsymbol{h}') - V(\boldsymbol{h}), \boldsymbol{h}' - \boldsymbol{h} \rangle \geq m \|\boldsymbol{h}' - \boldsymbol{h}\|^2$ for all $\boldsymbol{h}, \boldsymbol{h}'$, where $m$ is the strong monotonicity constant of Theorem 4. Consequently, for all $i$, $\ell_i(\boldsymbol{h}_i, \boldsymbol{h}_{-i}; P)$ is $m$-strongly convex in $\boldsymbol{h}_i$.*

*Proof.* Recall that Theorem 4 shows that the operator $W(\boldsymbol{h}) \in \mathbb{R}^{n \times k \times T}$, defined by:

$$W_i(\theta_i)(\boldsymbol{h}) = \Pr(\theta_i) \cdot \nabla_{\boldsymbol{h}_i(\theta_i)} \mathbb{E}_{\theta_{-i}, \boldsymbol{\lambda} \sim P|\theta_i}[c_i(\boldsymbol{h}_i(\theta_i), \boldsymbol{h}_{-i}(\theta_{-i}), \boldsymbol{\lambda})] \in \mathbb{R}^T$$

in the entry corresponding to agent $i$ and type $\theta_i$, is $m$-strongly monotone for some positive $m$.

Now, for every $i$, we can write:

$$V_i(\boldsymbol{h}) = \nabla_{\boldsymbol{h}_i} \left( \sum_{\theta_i} \Pr(\theta_i) \cdot \mathbb{E}_{\theta_{-i}, \boldsymbol{\lambda} \sim P|\theta_i}[c_i(\boldsymbol{h}_i(\theta_i), \boldsymbol{h}_{-i}(\theta_{-i}), \boldsymbol{\lambda})] \right)$$

Fix a agent $i$ and a type $\theta_i^*$. Since each agent has finitely many types, we can write $V$ as a vector of size $nkT$, where the entry of $V$ corresponding to agent $i$ and type $\theta_i^*$ is:

$$\begin{aligned}
V_i(\theta_i^*)(\boldsymbol{h}) &= \nabla_{\boldsymbol{h}_i(\theta_i^*)} \left( \sum_{\theta_i} \Pr(\theta_i) \cdot \mathbb{E}_{\theta_{-i}, \boldsymbol{\lambda} \sim P|\theta_i}[c_i(\boldsymbol{h}_i(\theta_i), \boldsymbol{h}_{-i}(\theta_{-i}), \boldsymbol{\lambda})] \right) \\
&= \nabla_{\boldsymbol{h}_i(\theta_i^*)} \left( \Pr(\theta_i^*) \cdot \mathbb{E}_{\theta_{-i}, \boldsymbol{\lambda} \sim P|\theta_i^*}[c_i(\boldsymbol{h}_i(\theta_i^*), \boldsymbol{h}_{-i}(\theta_{-i}), \boldsymbol{\lambda})] \right) \\
&\qquad \text{(since all terms not involving } \theta_i^* \text{ can be treated as constants)} \\
&= \Pr(\theta_i^*) \cdot \nabla_{\boldsymbol{h}_i(\theta_i^*)} \mathbb{E}_{\theta_{-i}, \boldsymbol{\lambda} \sim P|\theta_i^*}[c_i(\boldsymbol{h}_i(\theta_i^*), \boldsymbol{h}_{-i}(\theta_{-i}), \boldsymbol{\lambda})] \\
&= W_i(\theta_i^*)(\boldsymbol{h})
\end{aligned}$$

Thus $V = W$, and $V$ is $m$-strongly monotone, i.e. $\langle V(\boldsymbol{h}') - V(\boldsymbol{h}), \boldsymbol{h}' - \boldsymbol{h} \rangle \geq m \|\boldsymbol{h}' - \boldsymbol{h}\|^2$ for all $\boldsymbol{h}, \boldsymbol{h}'$. For every $i$, $m$-strong convexity then follows by definition, by considering $\boldsymbol{h}, \boldsymbol{h}'$ that are the same in all coordinates except $i$. $\square$

*Proof of Theorem 5.* For each $i$ and $\boldsymbol{h}_i'$, by the regret guarantees of $\text{Alg}_i$:

$$\frac{1}{R} \sum_{r=1}^{R} (\ell_i^r(\boldsymbol{h}_i^r) - \ell_i^r(\boldsymbol{h}_i')) = \mathbb{E}_{\boldsymbol{h} \sim \sigma^R} \mathbb{E}_{\theta, \boldsymbol{\lambda} \sim P}[c_i(\boldsymbol{h}_i(\theta_i), \boldsymbol{h}_{-i}(\theta_{-i}), \boldsymbol{\lambda}) - c_i(\boldsymbol{h}_i'(\theta_i), \boldsymbol{h}_{-i}(\theta_{-i}), \boldsymbol{\lambda})] \leq \epsilon_i(R)$$

Applying Proposition 1, we can conclude that for all $\theta_i$ and all $\boldsymbol{h}_i'(\theta_i)$:

$$\mathbb{E}_{\boldsymbol{h} \sim \sigma^R} \mathbb{E}_{\theta_{-i}, \boldsymbol{\lambda} \sim P|\theta_i}[c_i(\boldsymbol{h}_i(\theta_i), \boldsymbol{h}_{-i}(\theta_{-i}), \boldsymbol{\lambda}) - c_i(\boldsymbol{h}_i'(\theta_i), \boldsymbol{h}_{-i}(\theta_{-i}), \boldsymbol{\lambda})] \leq \frac{\epsilon_i(R)}{\Pr(\theta_i)}$$

$\square$

## D.2 ALGORITHM DETAILS

Here we present the algorithm of Jordan et al. (2024) and state its guarantees.

**Theorem 7** (Theorem 3.7 of Jordan et al. (2024)). *Consider a game $\mathcal{G}$ among $n$ agents, each with a convex and bounded action set $\mathcal{H}_i \subseteq \mathbb{R}^{d_i}$ and a cost function $\ell_i : \prod_{i=1}^{n} \mathcal{H}_i \to \mathbb{R}$ satisfying: (i) $\ell_i(\boldsymbol{h}_i, \boldsymbol{h}_{-i})$ is continuous in $(\boldsymbol{h}_i, \boldsymbol{h}_{-i})$ and continuously differentiable in $\boldsymbol{h}_i$; (ii) $\nabla_{\boldsymbol{h}_i} \ell_i(\boldsymbol{h}_i, \boldsymbol{h}_{-i})$ is continuous in $(\boldsymbol{h}_i, \boldsymbol{h}_{-i})$; (iii) $\|\boldsymbol{h} - \boldsymbol{h}'\| \leq D$ for all $\boldsymbol{h}, \boldsymbol{h}' \in \prod_{i=1}^{n} \mathcal{H}_i$; and (iv) $\mathcal{G}$ is $m$-strongly monotone. Suppose at every round $r \in [R]$, each agent observes an unbiased and bounded gradient $\tilde{\nabla}_{\boldsymbol{h}_i^r}^r$ satisfying $\mathbb{E}[\tilde{\nabla}_{\boldsymbol{h}_i^r}^r | \boldsymbol{h}^r] = \nabla_{\boldsymbol{h}_i^r} \ell_i(\boldsymbol{h}_i^r, \boldsymbol{h}_{-i}^r)$ and $\mathbb{E}[\|\tilde{\nabla}_{\boldsymbol{h}_i}^r\|^2 | \boldsymbol{h}^r] \leq M$. Then, if all agents run Algorithm 2, the final iterate satisfies:*

$$\mathbb{E}\left[\|\boldsymbol{h}^R - \boldsymbol{h}^*\|^2\right] \leq O\left( \frac{D^2 M (1 + \exp(1/(m^2 \log R))) \log(nR) \log^2(R)}{R} \right)$$

*where $\boldsymbol{h}^*$ is the Nash equilibrium of $\mathcal{G}$, i.e. for all $i \in [n]$, for all $\boldsymbol{h}_i \in \mathcal{H}_i$, $\ell_i(\boldsymbol{h}_i^*, \boldsymbol{h}_{-i}^*) \leq \ell_i(\boldsymbol{h}_i, \boldsymbol{h}_{-i}^*)$.*

---

**Algorithm 2:** AdaOGD (Algorithm 1 of Jordan et al. (2024))

---

**Input:** Strategy space $\mathcal{H}_i$

Initialize $\boldsymbol{h}_i^1 \in \mathcal{H}_i$

Let $z_0 = \frac{1}{\log(R+10)}$

**for** $r = 1, ..., R$ **do**

    Sample $M^r \sim \text{Geometric}(z_0)$

    Let $\eta^{r+1} = \frac{r+1}{\sqrt{1+\max\{M^1, ..., M^r\}}}$

    Update $\boldsymbol{h}_i^{r+1} = \arg\min_{\boldsymbol{h}_i \in \mathcal{H}_i}\{(\boldsymbol{h}_i - \boldsymbol{h}_i^r)^\top \tilde{\nabla}_i^r + \frac{\eta^{r+1}}{2}\|\boldsymbol{h}_i - \boldsymbol{h}_i^r\|^2\}$

---

### D.3 PROOF OF THEOREM 6

*Proof.* We define the game $\mathcal{G}$ where each agent $i$ chooses a strategy map $\boldsymbol{h}_i \in \mathcal{H}_i$ and suffers cost:

$$\ell_i(\boldsymbol{h}_i, \boldsymbol{h}_{-i}; P) = \mathbb{E}_{\theta, \boldsymbol{\lambda} \sim P}[c_i(\boldsymbol{h}_i(\theta_i), \boldsymbol{h}_{-i}(\theta_{-i}), \boldsymbol{\lambda})]$$

We verify the conditions of Theorem 7 on this game $\mathcal{G}$. Since for all type profiles $\theta$, $c_i(\boldsymbol{h}_i(\theta_i), \boldsymbol{h}_{-i}(\theta_{-i}), \boldsymbol{\lambda})$ is continuous in $(\boldsymbol{h}_i(\theta_i), \boldsymbol{h}_{-i}(\theta_{-i}))$ and continuously differentiable in $\boldsymbol{h}_i(\theta_i)$, we have that $\ell_i(\boldsymbol{h}_i, \boldsymbol{h}_{-i}; P)$ is continuous in $(\boldsymbol{h}_i, \boldsymbol{h}_{-i})$ and continuously differentiable in $\boldsymbol{h}_i$. Under Assumption 3, $\|\boldsymbol{h} - \boldsymbol{h}'\| \leq B\sqrt{nk}$ for all $\boldsymbol{h}, \boldsymbol{h}'$. Furthermore, by Lemma 4, $\mathcal{G}$ is $m$-strongly monotone, where $m$ is the strong monotonicity of Theorem 4.

To apply Theorem 7, it remains to establish that agents observe unbiased and bounded gradient feedback. Recall at each round $r \in [R]$, agent $i$ receives as feedback: $\tilde{\nabla}_{\boldsymbol{h}_i}^r = \nabla_{\boldsymbol{h}_i} c_i(\boldsymbol{h}_i^r(\theta_i^r), \boldsymbol{h}_{-i}^r(\theta_{-i}^r), \boldsymbol{\lambda}^r)$. Since $\theta^r, \boldsymbol{\lambda}^r \sim P$ are sampled independently from the strategies chosen at round $r$, we have that $\mathbb{E}[\tilde{\nabla}_{\boldsymbol{h}_i}^r | \boldsymbol{h}^r] = \mathbb{E}[\tilde{\nabla}_{\boldsymbol{h}_i}^r] = \mathbb{E}[\nabla_{\boldsymbol{h}_i} c_i(\boldsymbol{h}_i^r(\theta_i^r), \boldsymbol{h}_{-i}^r(\theta_{-i}^r), \boldsymbol{\lambda}^r)] = \nabla_{\boldsymbol{h}_i} \ell_i(\boldsymbol{h}_i^r, \boldsymbol{h}_{-i}^r; P)$, i.e. the gradient is unbiased. Moreover, we can compute, for any $\boldsymbol{h}_i, \boldsymbol{h}_{-i}, \theta, \boldsymbol{\lambda}$:

$$\|\nabla_{\boldsymbol{h}_i} c_i(\boldsymbol{h}_i(\theta_i), \boldsymbol{h}_{-i}(\theta_{-i}), \boldsymbol{\lambda})\|$$

$$= \|p_o \cdot 1_T + \alpha J \boldsymbol{h}_i(\theta_i) + \alpha M\left(\sum_{j \neq i} \boldsymbol{h}_j(\theta_j) - \boldsymbol{s}\right) + \beta\left(2\boldsymbol{h}_i(\theta_i) + \sum_{j \neq i} \boldsymbol{h}_j(\theta_j) - \boldsymbol{s}\right) - \nabla_{\boldsymbol{h}_i} f_i(\boldsymbol{h}_i(\theta_i))\|$$

$$\leq |p_0|\sqrt{T} + \alpha T B + \alpha T((n-1)B + S) + \beta((n+1)B + S) + U' \qquad \text{(by Assumption 3)}$$

where $J \in \mathbb{R}^{T \times T}$ is the matrix with $J_{tt} = 2$ for all $t \in [T]$ and 1 everywhere else, and $M \in \mathbb{R}^{T \times T}$ is the matrix with $M_{ts} = 1$ for $s \leq t$ and 0 everywhere else. Hence,

$$\mathbb{E}[\|\tilde{\nabla}_{\boldsymbol{h}_i}^r\|^2 | \boldsymbol{h}^r] = \mathbb{E}[\|\tilde{\nabla}_{\boldsymbol{h}_i}^r\|^2]$$

$$= \mathbb{E}[\|\nabla_{\boldsymbol{h}_i} c_i(\boldsymbol{h}_i^r(\theta_i^r), \boldsymbol{h}_{-i}^r(\theta_{-i}^r), \boldsymbol{\lambda}^r)\|^2]$$

$$\leq \left(p_{0_{max}}\sqrt{T} + \alpha_{max} T B + \alpha_{max} T((n-1)B + S) + \beta_{max}((n+1)B + S) + U'\right)^2$$

$$= \text{poly}(n, T, \alpha, \beta, p_0, B, S, U')$$

Above, $\alpha_{max} = \arg\max_{\alpha \in \text{supp}(P)}\{\alpha\}$, $\beta_{max} = \arg\max_{\beta \in \text{supp}(P)}\{\beta\}$, and $p_{0_{max}} = \arg\max_{p_0 \in \text{supp}(P)}\{|p_0|\}$

Therefore, by Theorem 7, the final iterate produced by Algorithm 2 satisfies:

$$\mathbb{E}\left[\|\boldsymbol{h}^R - \boldsymbol{h}^*\|^2\right] \leq O\left(\frac{D^2 M(1 + \exp(1/m^2 \log R))\log(nR)\log^2(R)}{R}\right)$$

where $\boldsymbol{h}^*$ is the Nash equilibrium of $\mathcal{G}$, and the expectation is taken over the randomness of the algorithm. Here, we have $D = \text{poly}(n, k, B)$ and $M = \text{poly}(n, T, \alpha, \beta, p_0, B, S, U')$.

Next we show that $\ell_i$ is Lipschitz in the $\ell_2$ norm, which will allow us to argue that since $\boldsymbol{h}^R$ and $\boldsymbol{h}^*$ are close in $\ell_2$ distance, they must also incur similar cost. Observe that by Assumption 3, for any $j \neq i$, for any $\boldsymbol{h}_i, \boldsymbol{h}_{-i}, \theta, \boldsymbol{\lambda}$:

$$\|\nabla_{\boldsymbol{h}_j} c_i(\boldsymbol{h}_i(\theta_i), \boldsymbol{h}_{-i}(\theta_{-i}), \boldsymbol{\lambda})\| = \|\alpha M^\top \boldsymbol{h}_i(\theta_i) + \beta \boldsymbol{h}_i(\theta_i)\| \leq (\alpha T + \beta) B$$

and so:

$$\sup_{\boldsymbol{h}} \|\nabla_{\boldsymbol{h}} c_i(\boldsymbol{h}_i(\theta_i), \boldsymbol{h}_{-i}(\theta_{-i}), \boldsymbol{\lambda})\|$$
$$\leq \underbrace{|p_0|\sqrt{T} + \alpha T B + \alpha T((n-1)B + S) + \beta((n+1)B + S) + U' + n(\alpha T + \beta)B}_{=:L'(p_0, \alpha, \beta)}$$

Therefore for all $\boldsymbol{h}, \boldsymbol{h}', \theta, \boldsymbol{\lambda}$, $c_i$ is $L'(p_0, \alpha, \beta)$-Lipschitz in $\boldsymbol{h}$, i.e.:

$$|c_i(\boldsymbol{h}'_i(\theta_i), \boldsymbol{h}'_{-i}(\theta_{-i}), \boldsymbol{\lambda}) - c_i(\boldsymbol{h}_i(\theta_i), \boldsymbol{h}_{-i}(\theta_{-i}), \boldsymbol{\lambda})| \leq L'\|\boldsymbol{h}' - \boldsymbol{h}\|$$

Taking expectations, we have that $\ell_i$ is $L$-Lipschitz in $\boldsymbol{h}$, i.e. for all $\boldsymbol{h}, \boldsymbol{h}'$:

$$|\ell_i(\boldsymbol{h}'_i, \boldsymbol{h}'_{-i}; P) - \ell_i(\boldsymbol{h}_i, \boldsymbol{h}_{-i}; P)| = |\mathbb{E}_{\theta, \boldsymbol{\lambda} \sim P}[c_i(\boldsymbol{h}'_i(\theta_i), \boldsymbol{h}'_{-i}(\theta_{-i}), \boldsymbol{\lambda}) - c_i(\boldsymbol{h}_i(\theta_i), \boldsymbol{h}_{-i}(\theta_{-i}), \boldsymbol{\lambda})]|$$
$$\leq \mathbb{E}_{\theta, \boldsymbol{\lambda} \sim P}[|c_i(\boldsymbol{h}'_i(\theta_i), \boldsymbol{h}'_{-i}(\theta_{-i}), \boldsymbol{\lambda}) - c_i(\boldsymbol{h}_i(\theta_i), \boldsymbol{h}_{-i}(\theta_{-i}), \boldsymbol{\lambda})|]$$
$$\leq L\|\boldsymbol{h}' - \boldsymbol{h}\|$$

where $L \leq \max_{p_0, \alpha, \beta} L'(p_0, \alpha, \beta) = \text{poly}(n, T, \alpha, \beta, p_0, B, S, U')$.

Thus, the cost evaluated at $\boldsymbol{h}^R$ is close to the cost evaluated at $\boldsymbol{h}^*$:

$$\mathbb{E}\left[\ell_i(\boldsymbol{h}^*_i, \boldsymbol{h}^*_{-i}; P) - \ell_i(\boldsymbol{h}^R_i, \boldsymbol{h}^R_{-i}; P)\right] \leq L \cdot \mathbb{E}\left[\|\boldsymbol{h}^R - \boldsymbol{h}^*\|\right] \qquad \text{(by } L\text{-Lipschitzness)}$$
$$\leq L \cdot O\left(\sqrt{\frac{D^2 M(1 + \exp(1/m^2 \log R)) \log(nR) \log^2(R)}{R}}\right)$$

In the second inequality, we use that fact that $\mathbb{E}\left[\|\boldsymbol{h}^R - \boldsymbol{h}^*\|\right]^2 \leq \mathbb{E}\left[\|\boldsymbol{h}^R - \boldsymbol{h}^*\|^2\right]$ by Jensen's inequality. Furthermore, since the entries of $\|\boldsymbol{h}^R - \boldsymbol{h}^*\|$ are non-negative, we also have that for any $\boldsymbol{h}_i \in \mathcal{H}_i$:

$$\mathbb{E}\left[\ell_i(\boldsymbol{h}_i, \boldsymbol{h}^R_{-i}; P) - \ell_i(\boldsymbol{h}_i, \boldsymbol{h}^*_{-i}; P)\right] \leq L \cdot \mathbb{E}\left[\|\boldsymbol{h}^R_{-i} - \boldsymbol{h}^*_{-i}\|\right] \qquad \text{(by } L\text{-Lipschitzness)}$$
$$\leq L \cdot \mathbb{E}\left[\|\boldsymbol{h}^R - \boldsymbol{h}^*\|\right]$$
$$\leq L \cdot O\left(\sqrt{\frac{D^2 M(1 + \exp(1/m^2 \log R)) \log(nR) \log^2(R)}{R}}\right)$$

Combining the above, we can show that $\boldsymbol{h}^R$ is an approximate Nash equilibrium of $\mathcal{G}$. In particular, for any $i$, for any $\boldsymbol{h}_i \in \mathcal{H}_i$:

$$\mathbb{E}\left[\ell_i(\boldsymbol{h}^R_i, \boldsymbol{h}^R_{-i}; P) - \ell_i(\boldsymbol{h}_i, \boldsymbol{h}^R_{-i}; P)\right]$$

$$\leq \mathbb{E}\left[\ell_i(\boldsymbol{h}^*_i, \boldsymbol{h}^*_{-i}; P) - \ell_i(\boldsymbol{h}_i, \boldsymbol{h}^R_{-i}; P)\right] + L \cdot O\left(\sqrt{\frac{D^2 M(1 + \exp(1/m^2 \log R)) \log(nR) \log^2(R)}{R}}\right)$$

$$\leq \mathbb{E}\left[\ell_i(\boldsymbol{h}^*_i, \boldsymbol{h}^*_{-i}; P) - \ell_i(\boldsymbol{h}_i, \boldsymbol{h}^*_{-i}; P)\right] + 2L \cdot O\left(\sqrt{\frac{D^2 M(1 + \exp(1/m^2 \log R)) \log(nR) \log^2(R)}{R}}\right)$$

$$\leq 2L \cdot O\left(\sqrt{\frac{D^2 M(1 + \exp(1/m^2 \log R)) \log(nR) \log^2(R)}{R}}\right)$$

$$\qquad \text{(by the fact that } \boldsymbol{h}^* \text{ is a Nash equilibrium)}$$

Thus, applying Proposition 1, we can conclude that $\boldsymbol{h}^R$ is an approximate Bayesian Nash equilibrium. Specifically, for all $i$, for all $\theta_i$, and for all $\boldsymbol{h}_i(\theta_i)$:

$$\mathbb{E}[\mathbb{E}_{\theta_{-i}, \boldsymbol{\lambda} \sim P|\theta_i}[c_i(\boldsymbol{h}_i^R(\theta_i), \boldsymbol{h}_{-i}^R(\theta_{-i}), \boldsymbol{\lambda}) - c_i(\boldsymbol{h}_i(\theta_i), \boldsymbol{h}_{-i}^R(\theta_{-i}), \boldsymbol{\lambda})]]$$

$$\leq \frac{2L}{\Pr(\theta_i)} \cdot O\left(\sqrt{\frac{D^2 M(1 + \exp(1/m^2 \log R)) \log(nR) \log^2(R)}{R}}\right) \quad \text{(by Proposition 1)}$$

$$\leq O\left(\frac{\text{poly}(n, T, k, \alpha, \beta, p_0, B, S, U')}{\Pr(\theta_i)} \cdot \frac{\log^{3/2}(R)}{\sqrt{R}}\right)$$

as desired. $\qquad\square$

### D.4 EXPERIMENTAL DETAILS

The online learning experimental setup follows that of Section 4. Here we provide more details on the conditional distributions of agent types and market parameters used. Recall that $\theta_1 \in [10, 15, 20]$ and $\theta_2 \in [20, 25, 30]$, and the joint type distribution $P(\theta_1, \theta_2)$ is uniform over the 9 possible type profiles. Agent 1's linear utility coefficient $r_1$ is drawn from a conditional Gaussian distribution: $r_1|\theta_1 \sim \mathcal{N}(\mu(\theta_1), 1)$ with $\mu(10) = 3, \mu(15) = 5$, and $\mu(20) = 7$. Similarly, Agent 2's linear utility coefficient $r_2$ is drawn from a conditional Gaussian distribution: $r_2|\theta_2 \sim \mathcal{N}(\mu(\theta_2), 1)$ with $\mu(20) = 6, \mu(25) = 8$, and $\mu(30) = 10$. Thus the type conditioned expected reserves are $(3, 5, 7)$ for agent 1 and $(6, 8, 10)$ for agent 2. Finally, we fix $p_0 = 0$, $\alpha = 0.1$, $\beta = \frac{1}{40}(\theta_1 + \theta_2)$, and draw $\boldsymbol{s}$ from the Gaussian distribution: for all $t \in [T]$, $s_t \sim \mathcal{N}(0, 1)$.

## E  LEARNING IN THE BAYESIAN GAME WITHOUT STOCHASTIC FEEDBACK

Here we relax the assumption that agents have access to online algorithms with no-regret guarantees under *stochastic gradient* feedback. Instead, we assume access to no-regret algorithms that, given cost function feedback, can learn over an agent's strategy space conditional on any type. Recall that $G_i(\theta_i)$ is the set of feasible strategies $\boldsymbol{h}_i(\theta_i)$ for agent $i$ and type $\theta_i$. Given a sequence of costs $c_i^r$, an algorithm $\texttt{Alg}$ achieves average regret bounded by $\epsilon(R)$ if:

$$\frac{1}{R}\sum_{r=1}^{R} c_i^r(\boldsymbol{h}_i^r(\theta_i)) - \min_{\boldsymbol{h}_i(\theta_i) \in G_i(\theta_i)} \frac{1}{R}\sum_{r=1}^{R} c_i^r(\boldsymbol{h}_i(\theta_i)) \leq \epsilon(R)$$

Mirroring Theorem 5, we show how agents can converge to Bayesian coarse correlated equilibrium by running separate instances of such no-regret algorithms, one for each type. The procedure is described in Algorithm 3 and mirrors the setup used by Hartline et al. (2015).

---

**Algorithm 3:** No-Regret Learning Protocol Without Stochastic Feedback

---

**Input:** No-regret algorithms $\texttt{Alg}_i$
**Output:** Joint distribution of strategy profiles $\sigma \in \Delta(\mathcal{H}_1 \times ... \times \mathcal{H}_n)$
Each agent $i$ initializes an instance of $\texttt{Alg}_i$ over the action space $G_i(\theta_i)$ for every $\theta_i \in \Theta_i$. We
  denote the instance corresponding to $\theta_i$ by $\texttt{Alg}_i(\theta_i)$.
**for** $r = 1, ..., R$ **do**
  **for** $i = 1, ..., n$ **do**
    For each $\theta_i$, let $\boldsymbol{h}_i^r(\theta_i) \in G_i(\theta_i)$ be the output of $\texttt{Alg}_i(\theta_i)$.
    Observe $\theta_i^r$ and take action $\boldsymbol{h}_i^r(\theta_i^r)$.
    Receive cost function $c_i^r$ and update $\texttt{Alg}_i(\theta_i^r)$ with $c_i^r$. Update all other $\texttt{Alg}_i(\theta_i)$,
    $\theta_i \neq \theta_i^r$, with $c_i^r = 0$.

Output empirical distribution over strategy profiles $\{\boldsymbol{h}^1, ..., \boldsymbol{h}^R\}$.

---

**Theorem 8.** *Fix a joint distribution $P$ over types $\theta$ and instances $\boldsymbol{\lambda}$. Fix $\delta \in (0, 1)$. For every $i \in [n]$, suppose there is an algorithm $\mathrm{Alg}_i$ that, given any $G_i(\theta_i)$, and against any sequence $c_i^1, ..., c_i^R$, obtains average regret bounded by $\epsilon_i(R)$ after $R$ rounds. Let $\sigma^R$ be the output of the Bayesian no-regret learning protocol (Algorithm 3), when given as input algorithms $\mathrm{Alg}_i$. Then, for every agent $i$, for every type $\theta_i \in \Theta_i$, and every $\boldsymbol{h}_i'(\theta_i) \in G_i(\theta_i)$, with probability at least $1 - \delta$:*

$$\mathbb{E}_{\boldsymbol{h} \sim \sigma^R} \mathbb{E}_{\theta_{-i}, \boldsymbol{\lambda} \sim P|\theta_i}[c_i\big(\boldsymbol{h}_i(\theta_i), \boldsymbol{h}_{-i}(\theta_{-i}), \boldsymbol{\lambda}\big) - c_i\big(\boldsymbol{h}_i'(\theta_i), \boldsymbol{h}_{-i}(\theta_{-i}), \boldsymbol{\lambda}\big)] \leq \frac{\epsilon_i(R) + 2H\sqrt{\frac{2\ln\frac{2}{\delta}}{R}}}{\Pr(\theta_i)}$$

*Here, $H = Bp_{0_{max}}\sqrt{T} + \alpha_{max}BT(nB + S) + \beta_{max}B(nB + S) + U$, where $\alpha_{max} = \arg\max_{\alpha \in supp(P)}\{\alpha\}$, $\beta_{max} = \arg\max_{\beta \in supp(P)}\{\beta\}$, and $p_{0_{max}} = \arg\max_{p_0 \in supp(P)}\{|p_0|\}$.*

First, we show in the following lemma a concentration bound: on any sequence a type $\theta_i^*$ was observed, the agent $i$'s cost under the empirically observed types and game instances concentrate around their expected cost.

**Lemma 5.** *Fix a agent $i$ and a type $\theta_i^* \in \Theta_i$. Fix $\delta \in (0, 1)$. Suppose costs are bounded between $[-H, H]$ uniformly over all types and strategies. Let $\boldsymbol{h}^1, ..., \boldsymbol{h}^R$ be any sequence of strategy profiles, and let $\sigma^R$ denote the empirical distribution over $\boldsymbol{h}^1, ..., \boldsymbol{h}^R$. Then, with probability at least $1 - \delta$:*

$$\left| \frac{1}{R}\sum_{r=1}^{R} \mathbf{1}[\theta_i^r = \theta_i^*]\, c_i\big(\boldsymbol{h}_i^r(\theta_i^r), \boldsymbol{h}_{-i}^r(\theta_{-i}^r), \boldsymbol{\lambda}\big) - \mathbb{E}_{\boldsymbol{h} \sim \sigma^R}\mathbb{E}_{\boldsymbol{\theta}, \boldsymbol{\lambda} \sim P}\big[\mathbf{1}[\theta_i = \theta_i^*]\, c_i\big(\boldsymbol{h}_i(\theta_i^*), \boldsymbol{h}_{-i}(\theta_{-i}), \boldsymbol{\lambda}\big)\big] \right| \leq H\sqrt{\frac{2\ln\frac{1}{\delta}}{R}}.$$

*Proof.* For convenience, let $Y_i(\boldsymbol{h}, \theta_{-i}, \boldsymbol{\lambda}) = c_i\big(\boldsymbol{h}_i(\theta_i^*), \boldsymbol{h}_{-i}(\theta_{-i}), \boldsymbol{\lambda}\big)$ and $I_i^r = \mathbf{1}[\theta_i^r = \theta_i^*]$. Hence we can write:

$$\frac{1}{R}\sum_{r=1}^{R}\mathbf{1}[\theta_i^r = \theta_i^*]\, c_i\big(\boldsymbol{h}_i^r(\theta_i^r), \boldsymbol{h}_{-i}^r(\theta_{-i}^r), \boldsymbol{\lambda}\big) = \frac{1}{R}\sum_{r=1}^{R} I_i^r\, Y_i(\boldsymbol{h}^r, \theta_{-i}^r, \boldsymbol{\lambda}^r).$$

and, letting $p(\theta_i^*) = \Pr_P[\theta_i^*]$:

$$\mathbb{E}_{\boldsymbol{h} \sim \sigma^R}\mathbb{E}_{\boldsymbol{\theta}, \boldsymbol{\lambda} \sim P}\big[\mathbf{1}[\theta_i = \theta_i^*]\, c_i\big(\boldsymbol{h}_i(\theta_i^*), \boldsymbol{h}_{-i}(\theta_{-i}), \boldsymbol{\lambda}\big)\big] = \frac{1}{R}\sum_{r=1}^{R}\mathbb{E}_{\boldsymbol{\theta}, \boldsymbol{\lambda} \sim P}\big[\mathbf{1}[\theta_i = \theta_i^*]\, c_i\big(\boldsymbol{h}_i^r(\theta_i^*), \boldsymbol{h}_{-i}^r(\theta_{-i}), \boldsymbol{\lambda}\big)\big]$$

$$= \frac{1}{R}\sum_{r=1}^{R} p(\theta_i^*)\mathbb{E}_{\theta_{-i}, \boldsymbol{\lambda} \sim P|\theta_i^*}[Y_i(\boldsymbol{h}^r, \theta_{-i}, \boldsymbol{\lambda})].$$

Thus we want to bound the quantity $\left| \frac{1}{R}\sum_{r=1}^{R} I_i^r Y_i(\boldsymbol{h}^r, \theta_{-i}^r, \boldsymbol{\lambda}^r) - \frac{1}{R}\sum_{r=1}^{R} p(\theta_i^*)\mathbb{E}_{\theta_{-i}, \boldsymbol{\lambda} \sim P|\theta_i^*}[Y_i(\boldsymbol{h}^r, \theta_{-i}, \boldsymbol{\lambda})] \right|$.

Let $F_{\leq r}$ denote the sequence $\{I_i^s Y_i(\boldsymbol{h}^s, \theta_{-i}^s, \boldsymbol{\lambda}^s)\}_{s \leq r}$. Since types are drawn independently each round, $I_i^r$ is independent of $F_{\leq r-1}$. Moreover, $\boldsymbol{h}^r$ is chosen prior to the draw of $\boldsymbol{\theta}^r, \boldsymbol{\lambda}^r$, so $\boldsymbol{\theta}^r, \boldsymbol{\lambda}^r$ are independent of $\boldsymbol{h}^r$. Thus:

$$\mathbb{E}[I_i^r Y_i(\boldsymbol{h}^r, \theta_{-i}^r, \boldsymbol{\lambda}^r)|F_{\leq r-1}] = \mathbb{E}\big[\mathbb{E}[I_i^r Y_i(\boldsymbol{h}^r, \theta_{-i}^r, \boldsymbol{\lambda}^r)|F_{\leq r-1}, \theta_i^r]|F_{\leq r-1}\big]$$

$$= \mathbb{E}\big[\mathbf{1}[\theta_i^r = \theta_i^*]\mathbb{E}[Y_i(\boldsymbol{h}^r, \theta_{-i}^r, \boldsymbol{\lambda}^r)|F_{\leq r-1}, \theta_i^r]|F_{\leq r-1}\big]$$

$$= \mathbb{E}\big[\mathbf{1}[\theta_i^r = \theta_i^*]\mathbb{E}_{\boldsymbol{h}_{-i}, \boldsymbol{\lambda} \sim P|\theta_i^r}[Y_i(\boldsymbol{h}^r, \theta_{-i}, \boldsymbol{\lambda})|F_{\leq r-1}]|F_{\leq r-1}\big]$$

$$= \mathbb{E}\big[\mathbf{1}[\theta_i^r = \theta_i^*]\mathbb{E}_{\boldsymbol{h}_{-i}, \boldsymbol{\lambda} \sim P|\theta_i^r}[Y_i(\boldsymbol{h}^r, \theta_{-i}, \boldsymbol{\lambda})]|F_{\leq r-1}\big]$$

$$= \mathbb{E}_{\boldsymbol{h}_{-i}, \boldsymbol{\lambda} \sim P|\theta_i^*}[Y_i(\boldsymbol{h}^r, \theta_{-i}, \boldsymbol{\lambda})] \cdot \Pr[\mathbf{1}[\theta_i^r = \theta_i^*]|F_{\leq r-1}]$$

$$= \mathbb{E}_{\boldsymbol{h}_{-i}, \boldsymbol{\lambda} \sim P|\theta_i^*}[Y_i(\boldsymbol{h}^r, \theta_{-i}, \boldsymbol{\lambda})] \cdot \Pr[\mathbf{1}[\theta_i^r = \theta_i^*]]$$

$$= p(\theta_i^*)\mathbb{E}_{\boldsymbol{h}_{-i}, \boldsymbol{\lambda} \sim P|\theta_i^*}[Y_i(\boldsymbol{h}^r, \theta_{-i}, \boldsymbol{\lambda})].$$

By Azuma's inequality:

$$\Pr\left[\left| \frac{1}{R}\sum_{r=1}^{R} I_i^r Y_i(\boldsymbol{h}^r, \theta_{-i}^r, \boldsymbol{\lambda}^r) - \frac{1}{R}\sum_{r=1}^{R}\mathbb{E}[I_i^r Y_i(\boldsymbol{h}^r, \theta_{-i}^r, \boldsymbol{\lambda}^r)|F_{\leq r-1}] \right| \geq m\right] \leq 2\exp\left(\frac{-m^2 R}{2H^2}\right).$$

Plugging in $m \geq H\sqrt{\frac{2\ln\frac{1}{\delta}}{R}}$, we have that with probability at least $1 - \delta$:

$$\left| \frac{1}{R}\sum_{r=1}^{R} I_i^r Y_i(\boldsymbol{h}^r, \theta_{-i}^r, \boldsymbol{\lambda}^r) - \frac{1}{R}\sum_{r=1}^{R} \mathbb{E}[I_i^r Y_i(\boldsymbol{h}^r, \theta_{-i}^r, \boldsymbol{\lambda}^r)|F_{\leq r-1}] \right|$$

$$= \left| \frac{1}{R}\sum_{r=1}^{R} I_i^r Y_i(\boldsymbol{h}^r, \theta_{-i}^r, \boldsymbol{\lambda}^r) - \frac{1}{R}\sum_{r=1}^{R} p(\theta_i^*)\mathbb{E}_{\theta_{-i}, \boldsymbol{\lambda}\sim P|\theta_i^*}[Y_i(\boldsymbol{h}^r, \theta_{-i}, \boldsymbol{\lambda})] \right|$$

$$= \left| \frac{1}{R}\sum_{r=1}^{R} \mathbf{1}[\theta_i^r = \theta_i^*]\, c_i\big(\boldsymbol{h}_i^r(\theta_i^r), \boldsymbol{h}_{-i}^r(\theta_{-i}^r), \boldsymbol{\lambda}\big) - \mathbb{E}_{\boldsymbol{h}\sim\sigma^R}\mathbb{E}_{\boldsymbol{\theta},\boldsymbol{\lambda}\sim P}\big[\mathbf{1}[\theta_i = \theta_i^*]\, c_i\big(\boldsymbol{h}_i(\theta_i^*), \boldsymbol{h}_{-i}(\theta_{-i}), \boldsymbol{\lambda}\big)\big] \right|$$

$$\leq H\sqrt{\frac{2\ln\frac{1}{\delta}}{R}},$$

as claimed. $\qquad\qquad\qquad\qquad\qquad\qquad\qquad\qquad\qquad\qquad\qquad\qquad\qquad\qquad\square$

Now we prove the theorem.

*Proof.* Let $\sigma^R$ be the empirical distribution over $\boldsymbol{h}^1, ..., \boldsymbol{h}^R$, the history of strategy profiles output by the learning protocol. Consider a agent $i$, and fix a type $\theta_i^* \in \Theta_i$ and any action $\boldsymbol{h}_i'(\theta_i) \in G_i(\theta_i)$. By the regret guarantee of $\texttt{Alg}_i(\theta_i^*)$, we have that:

$$\frac{1}{R}\sum_{r=1}^{R} c_i\big(\boldsymbol{h}_i^r(\theta_i^r), \boldsymbol{h}_{-i}^r(\theta_{-i}^r), \boldsymbol{\lambda}\big) - \frac{1}{R}\sum_{r=1}^{R} c_i\big(\boldsymbol{h}_i'(\theta_i), \boldsymbol{h}_{-i}^r(\theta_{-i}^r), \boldsymbol{\lambda}\big) \leq \epsilon_i(R).$$

By construction, on the rounds where $\theta_i^r \neq \theta_i^*$, $c_i^r = 0$ for all actions in $G_i(\theta_i)$, and so we equivalently have:

$$\underbrace{\frac{1}{R}\sum_{r=1}^{R} \mathbf{1}[\theta_i^r = \theta_i^*]\, c_i\big(\boldsymbol{h}_i^r(\theta_i^r), \boldsymbol{h}_{-i}^r(\theta_{-i}^r), \boldsymbol{\lambda}\big)}_{(1)} - \underbrace{\frac{1}{R}\sum_{r=1}^{R} \mathbf{1}[\theta_i^r = \theta_i^*]\, c_i\big(\boldsymbol{h}_i'(\theta_i), \boldsymbol{h}_{-i}^r(\theta_{-i}^r), \boldsymbol{\lambda}\big)}_{(2)} \leq \epsilon_i(R).$$

Before analyzing this expression, we bound the magnitude of costs. For any $\boldsymbol{h}_i, \boldsymbol{h}_{-i}, \theta, \boldsymbol{\lambda}$, we have, by applying Cauchy-Schwarz and Assumption 3:

$$\big|c_i\big(\boldsymbol{h}_i(\theta_i), \boldsymbol{h}_{-i}(\theta_{-i}), \boldsymbol{\lambda}\big)\big|$$

$$\leq |p_0|\|\boldsymbol{h}_i(\theta_i)\|_1 + \alpha\|M\boldsymbol{h}_i(\theta_i)\|\left\|\sum_{j=1}^{n}\boldsymbol{h}_j(\theta_j) - \boldsymbol{s}\right\| + \beta\|\boldsymbol{h}_i(\theta_i)\|\left\|\sum_{j=1}^{n}\boldsymbol{h}_j(\theta_j) - \boldsymbol{s}\right\| - f_i\big(\boldsymbol{h}_i(\theta_i)\big)$$

$$\leq B|p_0|\sqrt{T} + \alpha BT(nB+S) + \beta B(nB+S) + U,$$

where $M$ is the lower triangular matrix. We set $H = Bp_{0_{max}}\sqrt{T} + \alpha_{max}BT(nB+S) + \beta_{max}B(nB+S) + U$.

Then, to analyze term $(1)$: using Lemma 5 and the fact that:

$$\mathbb{E}_{\boldsymbol{h}\sim\sigma^R}\mathbb{E}_{\boldsymbol{\theta},\boldsymbol{\lambda}\sim P}\big[\mathbf{1}[\theta_i = \theta_i^*]\, c_i\big(\boldsymbol{h}_i(\theta_i^*), \boldsymbol{h}_{-i}(\theta_{-i}), \boldsymbol{\lambda}\big)\big] = p(\theta_i^*)\cdot\mathbb{E}_{\boldsymbol{h}\sim\sigma^R}\mathbb{E}_{\boldsymbol{\theta},\boldsymbol{\lambda}\sim P|\theta_i^*}\big[c_i\big(\boldsymbol{h}_i(\theta_i^*), \boldsymbol{h}_{-i}(\theta_{-i}), \boldsymbol{\lambda}\big)\big],$$

we have that with probability at least $1 - \frac{\delta}{2}$:

$$\left| \frac{1}{R}\sum_{r=1}^{R} \mathbf{1}[\theta_i^r = \theta_i^*]\, c_i\big(\boldsymbol{h}_i^r(\theta_i^r), \boldsymbol{h}_{-i}^r(\theta_{-i}^r), \boldsymbol{\lambda}\big) - \mathbb{E}_{\boldsymbol{h}\sim\sigma^R}\mathbb{E}_{\boldsymbol{\theta},\boldsymbol{\lambda}\sim P}\big[\mathbf{1}[\theta_i = \theta_i^*]\, c_i\big(\boldsymbol{h}_i(\theta_i^*), \boldsymbol{h}_{-i}(\theta_{-i}), \boldsymbol{\lambda}\big)\big] \right|$$

$$= \left| \frac{1}{R}\sum_{r=1}^{R} \mathbf{1}[\theta_i^r = \theta_i^*]\, c_i\big(\boldsymbol{h}_i^r(\theta_i^r), \boldsymbol{h}_{-i}^r(\theta_{-i}^r), \boldsymbol{\lambda}\big) - p(\theta_i^*)\cdot\mathbb{E}_{\boldsymbol{h}\sim\sigma^R}\mathbb{E}_{\boldsymbol{\theta},\boldsymbol{\lambda}\sim P|\theta_i^*}\big[c_i\big(\boldsymbol{h}_i(\theta_i^*), \boldsymbol{h}_{-i}(\theta_{-i}), \boldsymbol{\lambda}\big)\big] \right|$$

$$\leq H\sqrt{\frac{2\ln\frac{2}{\delta}}{R}}.$$

Similarly, for term (2): we apply Lemma 5 on the sequence where for all $r \in [R]$, $\boldsymbol{h}_i^r(\theta_i) = \boldsymbol{h}_i'(\theta_i)$ for all $\theta_i \in \Theta_i$, and $\boldsymbol{h}_{-i}^r$ remains unchanged. We have that with probability at least $1 - \frac{\delta}{2}$:

$$\left| \frac{1}{R} \sum_{r=1}^R \mathbf{1}[\theta_i^r = \theta_i^*] \, c_i\big(\boldsymbol{h}_i'(\theta_i), \boldsymbol{h}_{-i}^r(\theta_{-i}^r), \boldsymbol{\lambda}\big) - \mathbb{E}_{\boldsymbol{h} \sim \sigma^R} \mathbb{E}_{\boldsymbol{\theta}, \boldsymbol{\lambda} \sim P}\big[\mathbf{1}[\theta_i = \theta_i^*] \, c_i\big(\boldsymbol{h}_i'(\theta_i), \boldsymbol{h}_{-i}(\theta_{-i}), \boldsymbol{\lambda}\big)\big] \right|$$

$$= \left| \frac{1}{R} \sum_{r=1}^R \mathbf{1}[\theta_i^r = \theta_i^*] \, c_i\big(\boldsymbol{h}_i'(\theta_i), \boldsymbol{h}_{-i}^r(\theta_{-i}^r), \boldsymbol{\lambda}\big) - p(\theta_i^*) \cdot \mathbb{E}_{\boldsymbol{h} \sim \sigma^R} \mathbb{E}_{\boldsymbol{\theta}, \boldsymbol{\lambda} \sim P | \theta_i^*}\big[c_i\big(\boldsymbol{h}_i'(\theta_i), \boldsymbol{h}_{-i}(\theta_{-i}), \boldsymbol{\lambda}\big)\big] \right|$$

$$\leq H \sqrt{\frac{2 \ln \frac{2}{\delta}}{R}}.$$

Thus, we can conclude, with probability at least $1 - \delta$:

$$p(\theta_i^*) \cdot \mathbb{E}_{\boldsymbol{h} \sim \sigma^R} \mathbb{E}_{\theta_{-i}, \boldsymbol{\lambda} \sim P | \theta_i^*}\big[c_i\big(\boldsymbol{h}_i(\theta_i), \boldsymbol{h}_{-i}(\theta_{-i}), \boldsymbol{\lambda}\big)\big] - p(\theta_i^*) \cdot \mathbb{E}_{\boldsymbol{h} \sim \sigma^R} \mathbb{E}_{\theta_{-i}, \boldsymbol{\lambda} \sim P | \theta_i^*}\big[c_i\big(\boldsymbol{h}_i'(\theta_i), \boldsymbol{h}_{-i}(\theta_{-i}), \boldsymbol{\lambda}\big)\big]$$

$$\leq \frac{1}{R} \sum_{r=1}^R \mathbf{1}[\theta_i^r = \theta_i^*] \, c_i\big(\boldsymbol{h}_i^r(\theta_i^r), \boldsymbol{h}_{-i}^r(\theta_{-i}^r), \boldsymbol{\lambda}\big) + \frac{1}{R} \sum_{r=1}^R \mathbf{1}[\theta_i^r = \theta_i^*] \, c_i\big(\boldsymbol{h}_i'(\theta_i), \boldsymbol{h}_{-i}^r(\theta_{-i}^r), \boldsymbol{\lambda}\big) + 2H \sqrt{\frac{2 \ln \frac{2}{\delta}}{R}}$$

$$\leq \epsilon_i(R) + 2H \sqrt{\frac{2 \ln \frac{2}{\delta}}{R}},$$

and:

$$\mathbb{E}_{\boldsymbol{h} \sim \sigma^R} \mathbb{E}_{\theta_{-i}, \boldsymbol{\lambda} \sim P | \theta_i^*}\big[c_i\big(\boldsymbol{h}_i(\theta_i), \boldsymbol{h}_{-i}(\theta_{-i}), \boldsymbol{\lambda}\big)\big] - \mathbb{E}_{\boldsymbol{h} \sim \sigma^R} \mathbb{E}_{\theta_{-i}, \boldsymbol{\lambda} \sim P | \theta_i^*}\big[c_i\big(\boldsymbol{h}_i'(\theta_i), \boldsymbol{h}_{-i}(\theta_{-i}), \boldsymbol{\lambda}\big)\big]$$

$$\leq \frac{\epsilon_i(R) + 2H \sqrt{\frac{2 \ln \frac{2}{\delta}}{R}}}{\Pr(\theta_i^*)}.$$

This completes the proof. $\qquad\square$

# F  EXPERIMENTS ON REAL MARKET DATA

Our model assumes that trade volume has a permanent and temporary impact on prices in the market. This is consistent with/motivated by both established literature in economics and by financial models (see *e.g.* Almgren & Chriss (2000); Almgren et al. (2005); Chriss (2024b); Kearns & Shi (2025)). We now motivate this with publicly available real market data. Specifically, we extract market data for the Canadian Dollar (CAD) to U.S. Dollar (USD) forex exchange from Dukascopy, a Swiss financial services firm, which provides tick-level data on the following columns: bid, ask, bid volume, and ask volume. We first use outline a simple approach to extract the permanent and temporary impact coefficients from such coarse data, which may be of independent interest. We then use the extracted market coefficients and simulate equilibrium strategies for traders within this real market.

## ESTIMATING $\alpha$ – THE PERMANENT IMPACT COEFFICIENT

Permanent impact (*i.e.* the coefficient represented by $\alpha$) represents the non-transient effect on price due to the imbalance of supply and demand. One can approximate the excess demand by taking the difference between the bid and ask volumes (actual execution data is rarely available publicly). As for the price, we model it by computing the mid-price – the mid-point of the bid and ask prices at each time step. We regress the next step mid-price (we use the average over a window of 100 ticks) based on the average excess volume in the current step.

Market data routinely demonstrates periodic and temporal effects. To control for this, we build a training data set with tick data from 10am-11am, 11am-12pm, and 12pm-1pm (Eastern time) for the

9 Tuesdays between September 2, 2025 and November 4, 2025. Each of these hour-long intervals consist of roughly 6000 ticks. As for the test set, we look to predict the mid-price change for the same three hour-long time windows on Tuesday, Nov 11. The results below in Figure 4.

We first note that we *expect* the data here to be extremely noisy. A very strong signal/correlation of how future mid-price is affected by present supply-demand imbalance would present a meaningful signal to profit from, and would be discovered by market participants (*i.e.* would violate principle of no arbitrage). What we hope for, and observe, is a faint but consistent signal of the expected relationship.

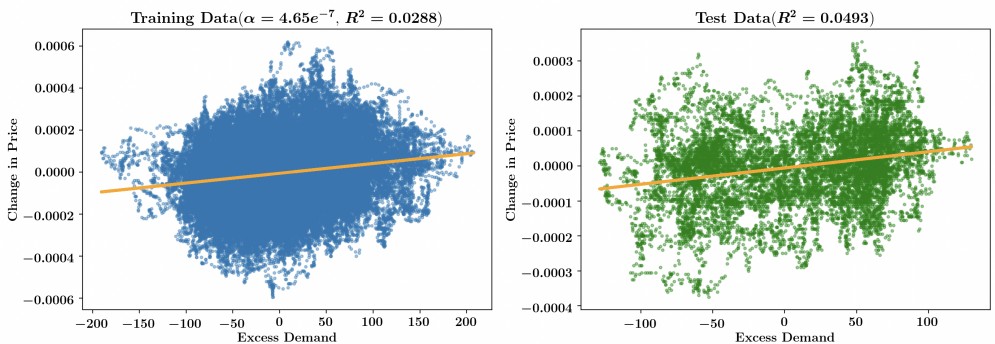

Figure 4: Predicted permanent impact coefficient $\alpha = 4.65e^{-7}$ on our training set using OLS regression. We demonstrate the performance on the test set on the right.

### ESTIMATING $\beta$ – THE TEMPORARY IMPACT COEFFICIENT

Estimating $\beta$, the temporary impact coefficient, is harder using such a simple publicly available dataset. One approach is as follows: $\beta$ essentially captures the premium that traders pay at execution time due to a large imbalance of buy/sell volume present. One proxy of this premium is the ask-bid spread – the larger this value, the higher amount market makers can charge to execute an order. Imbalance of either, excess supply or excess demand, leads to a higher spread and thus a higher execution cost. Based on this, we predict the ask-bid spread at the next time step (more specifically a window of ticks), based on the absolute volume imbalance at the current time-step (window of ticks). We present the results below in Figure 5. As before, we observe a faint but consistent signal on the estimation of the temporary impact coefficient, noting again, that a strong correlation is unrealistic in live market data. Lastly, we observe that in this market, $\beta$ is roughly 10x larger than $\alpha$, which corresponds to the middle plot in all our synthetic experiments.

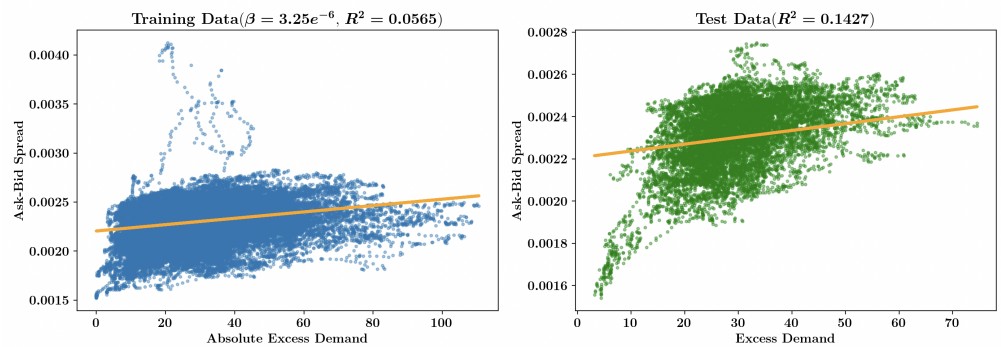

Figure 5: Predicted temporary impact coefficient $\beta = 3.25e^{-6}$ on our training set using OLS regression. We demonstrate the performance on the test set on the right.

EQUILIBRIUM SIMULATIONS UNDER REAL MARKET

Equipped with an estimate of the real market parameters for a foreign exchange trading market, we can simulate the equilibrium strategies within such markets. For simplicity, we focus on the plots for the complete information, noting that the remaining settings will exhibit similar phenomena insofar as real-market effects are concerned.

Consider the 10am-11am (EST) trading window on October 7th, 2025, which falls within the periods over which $\alpha$ and $\beta$ were estimated above. The initial price of 1 USD was 1.395 CAD in this window (thus $p_0 = 1.395$). We use the difference between the extracted bid and ask volume data to model the exogenous agent – note that, as before, positive values here denote excess demand. As for the strategic agents, we consider a setup similar to other experiments we run. There are 5 agents with heterogenous linear final position utility $f_i(\boldsymbol{h}_i) = r_i \sum_t h_{i,t}$ for some reserve price $r_i$, and constraints $-V_i \leq \mathbf{1}^T \boldsymbol{h}_i \leq V_i$ (i.e. final position must be in range $[-V_i, V_i]$). For computational ease, we discretize time to 1-minute interval and adjust all parameters accordingly (each $s_t$ for the minute interval is the sum of the excess demands within that minute). The results are presented below in Figure 6.

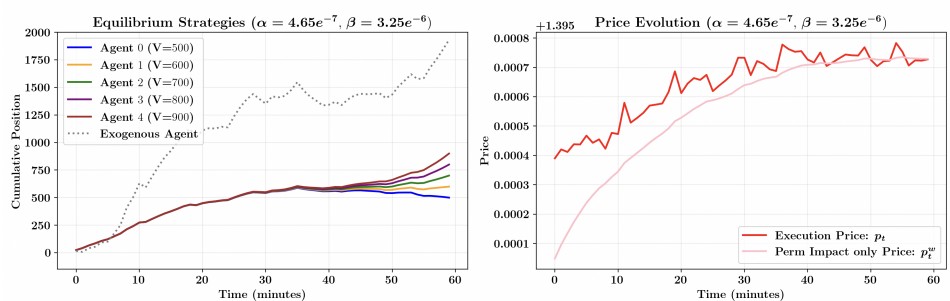

Figure 6: Cumulative position over time for agents in NE (left) and the price evolution in NE (right) for given $\alpha$ and $\beta$. The initial price is $p_0 = 1.395$, the reserve prices are $(1.40, 1.405, 1.41, 1.415, 1.42)$ and the constraint values are $V = (500, 600, 700, 800, 900)$.

We first observe that the exogenous agent here is primarily applying pressure on the demand side and is not random. This is a point of departure from our synthetic experiments (although none of our results made any assumption about $s_t$). That said, the core pattern of how strategic agents trade – similar strategies with divergence two third of the way – is similar to the middle panel of our synthetic experiments, where the $\alpha, \beta$ ratio was similar. This suggest the salient feature of the market is the ratio between these market parameters.

# G    COMPARING EQUILIBRIUM STRATEGIES TO A COMMON BASELINE

Our model considers the dynamics of $n$ traders looking to execute a position within some markets governed by parameters $\alpha, \beta$, signifying the permanent and temporary impact of trading volume on price. We take a game-theoretic approach, where each trader's strategy is dictated by an equilibrium between all players. It is, however, instructive to compare such an approach with the widely used execution strategy known as *VWAP - Volume Weighted Average Price*.

The VWAP strategy is simple and doesn't require strategic consideration: at any given time interval, each agent trades proportional to the historical volume of trade that occurred at that interval. Larger trades are placed when there is expected to be large volume in the market, and smaller-sized trades are placed when the market volume is low. While our model does not explicitly model the historical market volume, this can be easily remedied by reinterpreting our model's time dimension. That is, instead of treating each time step $[t, t+1]$ as a fixed period of wall-clock time, we instead interpret it as *volume weighted time*. That is, based on past historical market data, we dilate each interval (shrink or expand) such that an equal amount of volume is traded within each interval. This may mean, for

instance, that one interval corresponds to wall-clock time [9:30, 9:35] (high volume at the start of the day) and another to wall-clock time [11:00, 12:00] (lower volume at mid-day). Indeed, the price impact model (Almgren & Chriss, 2000) that our work is based on implicitly already assumes that time is defined as "volume-weighted time" exactly like this (see *e.g.* Almgren et al. (2005) for a detailed discussion of this issue.) Further, changing the definition of how time is measured does not change any of our results.

In volume-weighted time, the VWAP strategy is simple: trade a constant amount at each interval. So, a trader who wishes to build a position $V_i$ simply executes $V_i/T$ at each interval. Given this preamble, our core question is: *How does the VWAP strategy compare to the equilibrium strategy. Further, what are the dynamics of a market that include both equilibrium traders and VWAP traders?*

We explore this question in the context of our running synthetic example outlined in Section 3.1. Suppose we have 5 traders who want to build (hard constraint) positions of $[10, 15, 20, 25, 30]$. Suppose further that we set $\alpha = 0.1$, and set the exogenous player to be the same as in Figure 1 (which we recall was based on sampling *i.i.d.* mean-zero random noise variables at each time step). Then, in Figure 7, we plot the joint strategies where agents 2 and 3 follow VWAP, and the remaining agents follow the corresponding complete-information 3 player Nash Equilibrium given agents 2 and 3 following VWAP.

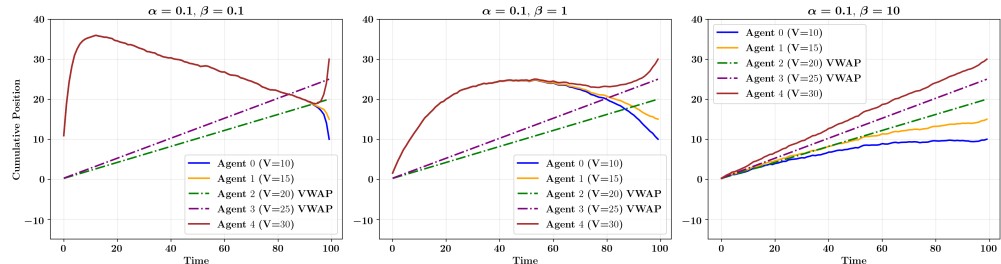

Figure 7: The strategies of the 5 players when agents 2 and 3 are playing VWAP and the remaining agents play the induced equilibrium of this setting.

Even though agents $0, 1, 4$ are strategic in both settings (Figures 1 and 7), the fact that agents 2 and 3 have now shifted to a VWAP strategy changes the equilibrium strategy of these 3. Note, however, that for the large $\beta = 10$ setting, the agent behaviors do not change much (both for those who deviate and those who don't). This is intuitive since when the temporary impact is large, agents generally want to spread out their trades regardless of other factors.

We next ask, how does this shift affect the cost (i.e. negative utility) incurred by all agents? In Figure 8, we see that the agents who switch from being strategic to playing VWAP pay a *higher* cost for doing so. However, as $\beta$ becomes larger, this becomes less consequential, as the ratio tends to 1. The effect on the remaining three players, however, is the opposite. These players end up paying a *lower* cost when agents (2,3) are following VWAP versus when they are strategic. This suggests that players who switch from being strategic to playing VWAP end up paying a higher cost at VWAP. Agents who are always strategic can exploit those who follow VWAP. As $\beta$ becomes large, however, these effects become small.

Our results suggest a transition to VWAP to strategic (or vice versa) can have both a positive or negative impact depending on the player. This begs the question of how this affects *total welfare*, *i.e.* the (negative) sum of all costs incurred by agents. In Figure 9, we plot the cumulative cost as a function of $\beta$. We plot 6 curves, where the $k^{th}$ curve corresponds to a subset of $k$ agents playing the VWAP strategy (we in fact consider all combinations of $k$ agents playing VWAP and take the average). Interestingly, we observe that the cumulative cost is almost always lower when a subset of agents are playing VWAP as compared to the all-strategic cumulative cost. This suggests that VWAP strategies could have better social welfare properties than all-strategic. We believe that this is an interesting finding that should be considred by market designers and regulators.

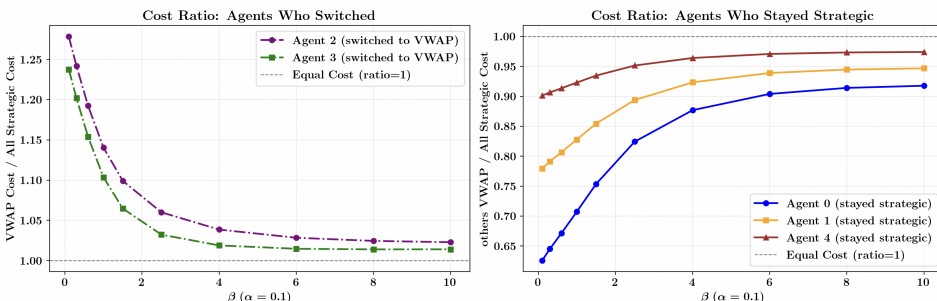

Figure 8: On the left is the ratio of cost between playing VWAP and playing strategically for agents 2 and 3. On the right is the ratio of costs for the remaining 3 agents (0,1,4) between when agents 2,3 were playing VWAP and when agents 2 and 3 were strategic.

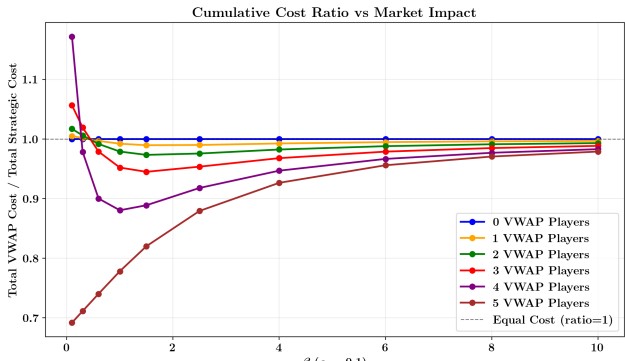

Figure 9: Ratio of cumulative costs between a subset of agents playing VWAP and all agents being strategic.

