# OpenReview forum: "Learning To Acquire Resources in Competition"
_ICLR.cc/2026/Conference — Submitted to ICLR 2026_

### Official Review · Reviewer_ZZ8N · 2025-10-18

**Soundness:** 3
**Presentation:** 3
**Contribution:** 2
**Rating:** 6
**Confidence:** 4

**Summary:**

This paper studies how multiple strategic agents acquire a costly divisible resource over time. It proposes a general discrete-time model with convex action set and concave idiosyncratic utilities, and a simple linear price dynamics with permanent and temporary impact. In the complete-information game there exists a unique pure NE, computable via a strongly-monotone VI with extragradient convergence; however the PoA is unbounded. In a Bayesian setting with finite type spaces and common prior, there is a unique pure BNE and can be obtained by extra-gradient algorithm. In a repeated game setting without a common prior, no-regret learning leads to an $epsilon$-BCCE and using the doubly-optimal OGD schedule gives last-iterate convergence to the approximate BNE under strong monotonicity.

**Strengths:**

1. This paper forms a general model with convex constraints, concave utilities and linear price dynamics with permanent and temporary impact.
2. Prove the unique, efficiently computable NE and BNE via strongly monotone VIs and extra-gradient in the first two settings.
3. For the learning without priors case, show that the average convergence to an approximate BCCE under generic no-regret, and last-iterate convergence to an approximate BNE under AdaOGD.

**Weaknesses:**

1. Assuming access to full counterfactual cost or unbiased gradients is somehow unrealistic in markets. The last-iterate result requires all agents to use the specified OGD schedule and assumes strong monotonicity and access to unbiased gradient/cost feedback. It’s unclear how robust this is under misspecified $\alpha, \beta$.
2. Given PoA is unbounded, the paper would benefit from identifying conditions leading to a bounded PoA.
3. The linear price-dynamics model is fairly simple, and the main proofs rely on standard VI arguments for strong monotonicity and well-known no-regret learning results. The learning component largely instantiates existing theory in this setting.

**Questions:**

1. How sensitive are the results to the choices of $\alpha$ and $\beta$? In particular, how do estimation errors in these parameters influence the convergence rate and the validity of the uniqueness/monotonicity assumptions?
2. In realistic settings, agents observe realized prices, not full gradients. Can your results extend to bandit feedback setting?

---

> ### Author Response · Authors · 2025-11-23
>
> We thank you for your detailed review. We are glad you found our model to be general and the appreciated our technical results. We address your specific comments in detail below:
>
> > Regarding the cost feedback in the online setting
>
> We address this broadly in the global response, but to summarize, since trades are public in almost all markets, agents can observe aggregate actions of others’ and then determine what their cost would be if they had acted differently. This kind of counterfactual cost computation is very standard in finance (often evaluated through backtesting) and is formalized in assumption 2. Crucially, observations of stochastic cost gradients as required by Algorithm 2 are strictly weaker than this cost observation outlined in Assumption 2 -- one can take the gradient of this couterfactual cost to obtain a valid stochastic gradient. Furthermore, our theory only assumes that the counterfactual cost or gradient is correct in expectation, so our results are automatically robust to any noise or perturbations.
>
> While it is valid to consider our online learning model with bandit feedback, we did not analyze this primarily because (1) a noisy version of much richer feedback is generally available in most practical settings and (2) it would lead to worse bounds. That said, such bandit-based approaches, or hybrid approaches that lie in-between bandits approaches and ours, could be an interesting avenue for future research for certain problem domains.
>
> > Regarding Price of Anarchy
>
> We sincerely thank you for pointing out the nuance surrounding Price of Anarchy (PoA). PoA is inherently a worse case notion and, as discussed in our global response, our revised draft includes a novel technical result (Theorem 3) that upper bounds the PoA for a class of instances where all agents want to buy (or respectively all want to sell) the resource to build a given position. In contrast, our worst case PoA relied on some strategic traders buying and others selling. We think this is an important contribution and highlights the nuance of the market/game.
>
> > Regarding robustness to mis-specified $\alpha$ and $\beta$
>
> This is an interesting question since market parameters are rarely observed perfectly. That said, we believe that our imperfect information setting, where agents hold some belief about the parameter distribution, already deals with this. Going further, in our learning setting, players are not assumed to have even a prior distribution over $\alpha$ or $\beta$; rather, they only have noisy observations of them after interacting in each round. In fact, the noise in an agent’s observation can depend on their private type and we make no assumptions about this relation. Ultimately, we are guaranteeing that agents learn to behave as optimally as they can given their available information, which can involve several degrees of uncertainty.
>
> > Regarding novelty of our learning results
>
> We expand on this point, and our core contribution, in the global response. To summarize, while some of the results do follow from existing works upon establishing certain properties, we believe that several are novel and require a nuanced and careful analysis. Moreover, we view our core contribution as providing a general model to capture an important problem that, in practice, has many interweaving properties. Our goal was to provide novel insights within this under-studied framework. From this perspective,  clarity and interpretability of our results is a clear advantage as it allows practitioners to engage and build atop this foundation.

---

### Official Review · Reviewer_Ntpc · 2025-10-29

**Soundness:** 4
**Presentation:** 4
**Contribution:** 2
**Rating:** 2
**Confidence:** 4

**Summary:**

The paper introduces a model of resource acquisition games and analyse it under various information conditions. Under complete information, the paper shows existence and uniqueness of a pure Nash equilibrium which can be efficiently computed but has worst-case unbounded price of anarchy. When information is partial but agents share a common prior, there exists a unique Bayesian Nash equilibrium, which can also be compute efficiently. When no common prior is available the paper studies the case in which agents gather information while playing by following an online learning algorithm. The paper provides sufficient conditions for convergence to a Bayesian CCE and last-iterate convergence to the BNE.

**Strengths:**

The model studied in the paper is interesting and well motivated by practical applications in finance and other markets. The paper is generally well written and clear. Technical results appear to be correct. Experiments, despite being on very simple instances, hint at interesting behaviors that may open up new directions of research on this model.

**Weaknesses:**

My main concern is with the strength of the technical contributions. Once equipped with Lemma 1, Section 3 and 4 rely on fairly standard tools for these kind of problems. In particular, the positive results are kind of expected given the specific structure of the problem being considered.

Section 5 is the one I find most interesting. However, Theorem 4 follows fairly standard ideas and is largely expected given similar results on convergence to BCCE in other games. Theorem 5  is largely based on proving that the problem being considered meets the requirements to apply the theorem by Jordan ed al. (2024).

Moreover, given the nice connection to practical applications, I would have appreciated a longer discussion about the feedback available to the online algorithm, and whether it makes sense in practice. From the discussion in paragraph starting at 359 this is not entirely clear. For instance, is assuming access to stochastic gradient feedback reasonable in practice? I would probably expect to have something closer to bandit feedback. Some discussion on this would be a useful addition.

Finally, simulations display some interesting behaviours (eg phase transitions) but they are all carried out on extremely simplified settings and it is difficult to extrapolate general insights from them. Extending the experimental analysis to richer synthetic or real-world settings would also be a nice addition to the current set of results.

Typos: ““proprety” line 239; “stochstic” line 420

**Questions:**

See question on feedback above.

---

> ### Author Response · Authors · 2025-11-23
>
> We thank you for the detailed review and comments. We appreciate that you found out model interesting and well motivated by practical applications in finance and markets. This was indeed our goal from the the start as we believe the phenomena we model has wide-ranging applicability. We address below the concerns you raised and hope this may improve your perception of our work.
>
> > Regarding the strength of our technical contributions
>
> In the global response, we give a broad discussion of our contributions, both in terms of modeling and technical novelty. As for the specific, you are right that Lemma 1 is used to establish strong monotonicity of the game, from which our results in Section 3 largely follow. However, this is mostly a warm up to our most important technical sections, especially Sections 4 and 5, in which we believe the results are more technically interesting. In particular, in the Bayesian version of the game, strong monotonicity is much less trivial to establish.
>
> While we agree that Theorem 4 (5 in updated pdf) is somewhat expected given similar results for convergence to BCCE in the literature, this still involves handling some subtle details – such as the continuous-valued parts of the market type that are not part of the discrete player types – that make it not a trivial application of standard theory, and require some careful handling. Further, we politely disagree with your view of Theorem 5 (6 in updated pdf); the algorithm of Jordan et al. (2024) is designed for full-information games, and applying it here for learning under imperfect information is both novel, and involves dealing with some subtle details.
>
> Beyond uniqueness and learning of equilibria, we also comment on the quality of equilibria through a Price of Anarchy analysis. In the revised version, we supplement the lower bound (which can be unbounded in the worst case) with an upper bound for a class of games that are practical interest (Theorem 3).
>
> Overall, our main contribution is to identify and analyze a class of resource acquisition games that are both practically relevant and analytically tractable—key for practitioners to interpret and apply our findings. We believe there are novel technical aspects in several of our results, but more importantly, there are major insights that are of practical relevance.
>
> > Regarding Experimentation around Real World Applications
>
> In general, experimentation with real world data is extremely difficult, since suitable real world market data is generally proprietary. However, as part of our revision, we take a major step by performing novel analysis on publicly available level 1 limit order book-style data to extract real market parameters for our model (Appendix F). We analyze the equilibrium and price dynamics, as specified by our model, within this real market setting. In this vein, we also include an empirical comparison to a commonly used execution algorithm used in practice and derive some novel insights (Appendix G). We believe this is a very important and meaningful contribution of practical importance; thank you for this suggestion.
>
> In addition, we added a very detailed discussion of the practical applicability of our broader model (both the game theoretic and learning aspects, including our feedback model) in Appendix A3. See our global response for a detailed discussion of this. To summarize, since trades are public in almost all markets, agents can observe aggregate actions of others’ and then determine what their cost would be if they had acted differently. This kind of counterfactual cost computation is very standard in finance (often evaluated through backtesting) and is formalized in assumption 2. Crucially, observations of stochastic cost gradients as required by Algorithm 2 are strictly weaker than this cost observation outlined in Assumption 2 -- one can take the gradient of this couterfactual cost to obtain a valid stochastic gradient. Furthermore, our theory only assumes that the counterfactual cost or gradient is correct in expectation, so our results are automatically robust to any noise or perturbations.

---

> > ### Comment · Reviewer_Ntpc · 2025-11-27
> >
> > Thank you for the detailed response and for the substantial updates in the revised version of the paper.
> >
> > I appreciate the authors’ efforts to strengthen the submission, and I agree that the revisions move the work in a positive direction.
> >
> > However, I continue to think that the technical contributions are not particularly strong for the reasons outlined in the review. Regarding the authors’ reply on Theorem 5 (now Theorem 6): although the authors state that there are “subtle details” involved, these remain unclear in the current version. If there are indeed significant challenges in adapting the algorithm of Jordan et al. (2024), these should be spelled out more explicitly in the paper.
> >
> > I fully agree with the authors that “more importantly, there are major insights that are of practical relevance”, and I believe that a stronger version of the submission would place greater emphasis on these practical aspects (potentially by integrating some of the newly added results from the appendix into the main text).
> >
> > I will confirm my original score, as I believe the paper still has considerable room for improvement.

---

> ### Author Response · Authors · 2025-12-02
>
> Thank you for your response and engagement. We respectively disagree with your perspective on Theorem 5 (now Theorem 6). This result relies on the key (but subtle) observation that the repeated Bayesian game can be re-formulated as a full-information game with stochastic feedback, via the “population loss” (Definition 6). To the best of our knowledge, such algorithms have not previously been used to obtain convergence guarantees to Bayesian Nash equilibrium, especially in settings that allow for (possibly continuous) game types. It is only given this observation that the result of this theorem stems from showing our problem meets the requirements to apply the theorem by Jordan et al. We also highlight that this is only one of several technical results in the paper.

---

### Official Review · Reviewer_i74L · 2025-10-31

**Soundness:** 4
**Presentation:** 4
**Contribution:** 3
**Rating:** 6
**Confidence:** 2

**Summary:**

This paper analyzes a class of games involving agents attempting to acquire resources. The class of games is shown to be strongly monotone, which leads to a standard family of results including fast convergence of extra-gradient descent to Nash equilibria, among others.

**Strengths:**

The paper is well written, and the results are rather strong, basically completely answering the question of what happens with learning dynamics in this class of games.

**Weaknesses:**

This paper's main argument is essentially: "Here is an interesting class of games. They are strongly monotone (Theorems 1 and 3). Thus, applying known results for strongly monotone games verbatim, good things happen". The strong monotonicity results seem pretty straightforward from the definitions, so the strength of the paper boils down completely to understanding how important this class of games is. I do not feel well placed to evaluate that---I come from a computer science background, and I am admittedly relatively unfamiliar with the more economic side of this paper. Thus, I will not attempt to do so, in the hope that at least one other reviewer has a better understanding here.

Minor note on the comment at L829 about CCE collapsing to NE in this class of games: I do not think that this follows from the preceding argument. It is not true that strict concavity of utilities implies that CCEs collapse to NEs. Instead, Theorem 1 (monotonicity of the associated VI operator) is what shows that CCE collapses to Nash---or, at least, that the marginals of any CCE form a Nash. (I did not carefully check the proof(s) of monotonicity, though it seems believable.) Perhaps the comment should be moved to after Theorem 1 instead.

**Questions:**

I have no specific questions, though I invite any comments on anything I said above.

---

> ### Author Response · Authors · 2025-11-23
>
> We thank the reviewer for their helpful review. We are glad they found our results strong and complete. We comment below on the technical strengths of our result; during the revision, we also formulated new technical results and experiments (see global response) that we believe gives additional practical insights into this real-world problem faced in trading and markets.
>
> > Strength of Our Technical Contributions
>
> In the global response, we give broader context on our contribution (carefully modeling a practical problem in markets to wide generality) and a detailed discussion about our paper’s most important technical contributions and novelty. Regarding the specific criticism that our arguments mostly boil down to simple arguments when we apply strong monotonicity verbatim: while we acknowledge that some of our initial results do have this flavor, we also feel that there is more nuance to it:
>
> - First, while we agree that strong monotonicity is fairly straightforward to show in the simple perfect information version of our game, this section was mostly provided for presentation purposes and to build to our most important contributions in section 4 and 5. Operating under imperfect information, strong monotonicity was not very straightforward to establish, and required some very careful technical analysis, as provided in our appendix.
> - Second, while some of our results (mostly those around existence and uniqueness of equilibria) are proven with strong monotonicity, not all of them are. In addition to these results, we have e.g. novel results on price of anarchy (PoA) and on last iterate convergence to BNE, which are interesting and novel technical results that are independent of this technique. In the revised version, we supplement the existing PoA lower bound with a novel upper bound based on the smooth games framework.
> - Lastly and as mentioned in the global response, we believe an important contribution of our work is the careful modeling and analysis of a generalized resource acquisition game within a strategic context. This setup is both practically relevant and provides the foundation for new work in this direction.
>
> > Regarding CCE collapsing to NE
>
> You are correct that CCE collapses to NE in any strongly monotone game. Thank you for pointing this out! In the revised version, we have moved this statement as you suggested.

---

> ### Comment · Reviewer_i74L · 2025-11-24
>
> Thank you. Having read the other reviews, responses, and the revised version, my overall opinion of the paper remains positive, and I will maintain my score.

---

### Official Review · Reviewer_5a9t · 2025-11-02

**Soundness:** 3
**Presentation:** 2
**Contribution:** 2
**Rating:** 4
**Confidence:** 3

**Summary:**

The paper extends a model studied by previous work to study resource acquisition under competition. The paper extends the model of previous work by allowing utility functions of agents to be concave functions of the acquired resource quantity. The model also extends to include a non-strategic or exogenous agent.

With these extensions, the paper shows that the equilibrium of the game defined remains unique and the paper demonstrates how to compute this equilibrium. They also study a version of the game with incomplete information, where agents have uncertainty about types and game parameters, holding beliefs over them rather than knowing exact values. The paper shows that there is a unique Bayesian Nash equilibrium. Finally, the paper provides classes of algorithms that allow agents to converge to a Bayesian CCE or the Bayesian Nash equilibrium.

**Strengths:**

The paper studies various extensions of previous work and shows that there remains a unique equilibrium. Given uniqueness, the equilibrium seems like a plausible solution concept for this setting.

**Weaknesses:**

The extensions are not well-motivated. Given that the differences between this work and previous work are the extensions, it would be useful to provide more information about how the extensions are important for the motivating applications to understand why we should find the extensions important. Additionally, if the main contribution of the work is extending the setting of the game over theoretical contributions, I think there should be more justification on why the extensions are important.

The novel technical contributions are not clear. At first glance, it appears like going from linear utilities to concave utilities are standard extensions that work out in many standard settings . It is unclear how much technical novelty there is in implementing these extensions. Likewise, in the section on learning dynamics for convergence to equilibrium, it seems like the results are applications of standard results on equilibrium convergence without any new technical contributions.

**Questions:**

Can you describe some of the technical challenges for your results compared to results of previous work? Can you describe the novel technical contributions your work makes?

---

> ### Author Response · Authors · 2025-11-23
>
> We thank you for the detailed review and helpful comments. We address your comments in detail below.
>
> > Regarding the model and extensions we consider
>
> With regard to your main critique, we first note that there is limited works on exploring strategic trading from a computational and learning perspective. Of those that exist, our work accomodates 4 main components that are not considered together in these prior works. We provide more detail on each:
> - General Constraints: constraints on execution behavior are very important and common in practical application. For example, practitioners often enforce constraints such as no short selling, limits on how quickly they buy or sell, and maximum / minimum limits on their overall position at any time. Such constraints are often important for controlling risk and being compliant with regulation. While some papers (in particular Chriss (2024c) and Kearns and Shi (2025)) have considered such constraints, they only considered them from the point of view of computing best responses under perfect information. Until our work, it was an open question for example on the impact these would have on the existence and uniqueness of equilibria, nor how to handle these for computing equilibria or working under imperfect information.
> - General Utilities: the prior work has only considered the strategic aspects when every agent is acquiring a fixed target position, but this is very trivializing, since it means only the costs to build the position need to be considered. In practice, agents also need to consider the utility of the position they are wanting to acquire (in terms of e.g. reservation price and/or position-based risks), and trade this off against the costs in acquiring the position. This leads to a much richer game than what prior work has considered. Furthermore, this prior work has generally depended on the very simple linear/quadratic cost function that is obtained without such additional utility functions, so including these introduces additional technical challenges.
> - Imperfect Information: With the exception of Chriss (2025) and Kearns and Shi (2025), all of this prior work has assumed perfect information, which is extremely unrealistic in practice. Furthermore, of these two methods, the first considers approaches that only work under many other trivializations of the problem (no constraints, no additional utility, and all players having common knowledge of player type distribution prior), and the second only works for learning under repeated play of the exact same game (i.e. with same players, same alpha/beta, etc.), which is also very trivializing.
> - Learning from Feedback: With the exception of Kearns and Shi (2025), none of these prior works have considered the problem of learning how to compete in any kind of data-driven way; they instead have only considered how to compute equilibria. Furthermore, as mentioned above, Kearns and Shi (2025) only considers learning in a very simple setting where the exact same game is repeated, so there is no learning how to compete based on varying game dynamics and contextual information. However, in real application, where there is imperfect information and no common prior on player types, there is no way to avoid needing to learn how to compare based on feedback, so this extension is of vital significance.
>
> > Regarding techical novelty
>
> We expand on this point in our global response. To summarize: (1) the core contributions of our paper revolve around modeling an important problem in wide generality, and (2) the novel insights and technical contributions we provide within this model (e.g. price of anarchy, how we approach imperfect information, etc). Given the practicality of the model, we perceive the clarity of our results as a clear advantage.

---

### Author Response · Authors · 2025-11-23

We thank the reviewers for their thoughtful feedback, and for acknowledging the technical soundness and clarity of our paper. Below we address the concerns regarding (i) the nature of our contribution and technical novelty, and (ii) how realistic the algorithms in section 5 are for practice. We also discuss here the new results and insights added to the paper draft during the rebuttal period.

## Our main contribution and technical novelty
We view our primary contribution to be in the modeling and problem formulation. In particular, we introduce a general framework for resource acquisition problems that:
- Formalizes price dynamics grounded in both market equilibrium theory and the market microstructure literature.
- Considers strategic aspects, which have been very underexplored in our target applications.
- Captures important aspects of the problem missing from prior work: (a) sufficiently general constraints; (b) utilities that capture e.g. risk aversion and inventory/holding costs; \(c) incomplete information; (d) computability and learning from interactions.

Prior work has explored some of the above missing aspects in isolation, but never together within a sufficiently general framework. Exploring these aspects in isolation often trivializes them; for example, considering imperfect information without consideration of constraints or learning, as in Chriss (2025), leads to methods that cannot extend to more realistic settings. These extensions are important, because they align closely with the practical reality.

While some of our results about this generalized framework do follow from standard results after establishing properties such as strong monotonicity, we argue that our overall analysis goes beyond this. First, we provide a price of anarchy analysis of this general game framework (which we further extend in our revised paper as described below.) This is a novel and non-trivial contribution that is very relevant (e.g. for market design and regulatory perspectives). Second, we argue that our analysis in sections 4 and 5 is not so straightforward. While some works (e.g. Hartline et al. (2015)) have considered this kind of hybrid Bayesian game + online learning framework before for convergence to BCC, it is by no means extremely common, and we believe our inclusion of both discrete player types and continuous-valued game types (defined by $\alpha$ $\beta$, and $s$) within this framework is novel, and doing this correctly is non-trivial. Furthermore, our application of doubly optimal learning (which was designed for non-Bayesian problems) to this setting to establish last iterate convergence to BNE is also quite novel, and involves many subtle and non-trivial details.

Overall our technical results are intended as first steps towards understanding strategic behavior and learning within this framework. We show that, despite our model’s generality, it remains analytically and computationally tractable; we see this as a strength of our work as it offers clean tractable insights to practitioners for what is, fundamentally, a real-world problem. In our revised version (see updated PDF) we provide supplemental technical and empirical results along this direction (see below).


## Realism of the feedback model for online learning

We thank the reviewers for raising this concern, as it is a very important consideration. In short, since trades are public in almost all markets, agents can observe aggregate actions of others’ and then determine what their cost would be if they had acted differently (their counterfactual cost). This is formalized in Assumption 2 where we consider an agent having access to this counterfactual cost. The stochastic gradient feedback assumed by Algorithm 2 is strictly weaker than this fairly natural assumption since the gradient of this cost function is itself a valid stochastic gradient. Thus, it is sufficient to consider the realism of Assumption 2. We clarified this in the revised version.

In the revised version, we add an additional appendix that provides a thorough discussion of considerations of applying our overall mathematical game framework to real settings. In particular, we provide a detailed discussion of the realism of assumption 2 there. We discuss how the counterfactual cost function can be estimated given: (a) observation of price trajectories; and (b) (potentially noisy) estimates of the total excess demand from all other agents at each time. For example, in financial applications, these are very realistic to observe for market participants who interact directly with electronic exchanges and observe full market microstructure information. So, even though agents might not receive gradient feedback “for free,” they can often reconstruct it. Furthermore, our theory only assumes that the counterfactual cost or gradient is correct in expectation, so our results are automatically robust to noise due to reconstruction of this kind.

---

### Author Response · Authors · 2025-11-23

## New technical and empirical results added to the paper
> Experiments on real market data and resulting equilibrium (Appendix F)

We provide a novel real data-based setting for running experiments on resource acquisition under competition. We believe this is very impactful, because to the best of our knowledge all prior works who have studied optimal resource acquisition in similar settings (based e.g. on Almgren and Chriss model) have relied on proprietary data (see e.g. Almgren et al. (2005)), so there is currently a lack of publicly available real data-based environments to experiment with.

For this setting, we take publicly available level 1 limit order book-style data for currency exchange markets. From this data, we are able to estimate supply and demand imbalances and price movements on a tick-by-tick basis, which allows us to run linear regressions to estimate the permanent $\alpha$ and temporary $\beta$ price impact components of our model. Furthermore, the observed excess demands provide an estimate of the exogenous agent actions, $s_t$ in our model, and together these allow for real world data-based simulations. We provide a detailed analysis of this setting in our revision.

> Comparing strategic and non-strategic behavior (Appendix G)

We provide a detailed analysis of the implications of our model for the impact of behaving strategically versus not. This is important, because standard thinking in optimal execution (which classically does not consider competition) is to execute according to Volume-Weighted Average Price (VWAP), which under our model corresponds to trading at a constant rate.

In this analysis, we compare the behavior and costs of non-strategic traders, who execute according to VWAP and strategic traders who play their resulting equilibrium strategies. We empirically show that as $\beta$ increases, strategic traders’ equilibrium strategies converge to VWAP. In mixed settings where a subset of traders play VWAP while the remaining play in equilibrium (treating the VWAP players as non-strategic), a single trader is generally worse off playing VWAP than acting strategically. However, we find that total welfare can increase as more traders adopt VWAP.

Overall, these findings highlight the importance of explicitly modeling strategic behavior, given its nontrivial implications for both individual costs and aggregate welfare. Furthermore, they are very interesting and relevant from the point of view of market design and regulation.

> Additional Price of Anarchy Analysis (Theorem 3)

Although we already provided a general price of anarchy analysis for the full game, which shows that it is in general completely unbounded, the counterexample for unboundedness relied on some agents wanting to buy and some agents wanting to sell the resource, and therefore providing supply to each other. However, this does not rule out possible price of anarchy bounds in settings where all agents want to buy (or respectively all want to sell) the resource. This is especially pertinent in several market settings: after an earnings call, for instance, traders may systematically move their positions in a positive direction if earnings were above expectations.

In the revised paper draft, we include a new technical result (Theorem 3) that upper bounds the PoA in such settings. More specifically, we leverage the smooth games framework of [Roughgarden, 2015] to show that PoA is upper bounded by $O(n^2T^2\gamma^2)$, where $\gamma$ is a ratio between $\alpha$ and $\beta$. We thank reviewer ZZ8N for bringing this up.

---

### Author Response · Authors · 2025-12-02
**Summarizing Remarks**

We firstly thank all reviewers and ACs for their hard work in reviewing and helping us improve the paper during a challenging process. Resource acquisition in a competitive, market driven setting, is a problem of deep practical relevance, not only in financial markets, but also in growing environments like compute/GPU markets. Our goal in this work was to faithfully model and capture the intricacies of this problem in broad generality, noting that existing literature herein is sparse. Our work gives an extensive and complete suite of technical, empirical, and real-world results, including:
- equilibrium characterization (existence, uniqueness, etc) under different information structures and corresponding empirical validation
- detailed price of anarchy (PoA) analysis (strengthened during the rebuttal due to helpful reviews)
- regret analysis of equilibrium learning from online feedback
- experiments on real markets data and comparison to real baseline/strategies used in trading (added during the rebuttal due to helpful reviews)

While many of our results, like the PoA analysis, are novel, some of them appeal to existing techniques and results upon non-trivial reformulations. Since the problem is very real but understudied, such results allow engagement from practitioners and are thus worth disseminating. In the revised version, we further expound on the practicality of our setting by discussing, among others, how the online feedback model is quite natural and precisely explaining how this manifests in real markets.

---

### Meta-Review · Area_Chair_9Snk · 2026-01-05

**Summary:**

The primary concerns regarding this paper focus on its technical novelty and the practicality/realism of the proposed model. On technical novelty, reviewers questioned whether the work is a straightforward application of existing models and analysis frameworks. While the authors argued in their rebuttal that their reformulation for incomplete information and the Price of Anarchy (PoA) analysis involved unique contributions, they failed to articulate the specific technical challenges or hurdles they overcame. Consequently, the reviewers remain unconvinced regarding the depth of the technical contribution. Conversely, the authors provided a persuasive response regarding the model's practicality by presenting concrete and realistic application scenarios.

**Reviewer Concerns:**

Addressed by Rebuttal: The authors successfully addressed concerns regarding the model's motivation and practical relevance. By providing specific application scenarios, they demonstrated that the problem setting is realistic and significant.

Outstanding Concerns: The concern regarding technical novelty remains a major outstanding issue. The rebuttal lacked a detailed explanation of the "technical delta" or the non-triviality of the analysis compared to prior work. Without a clear description of the specific technical difficulties addressed, the contribution is perceived as incremental.

**Reviewer Scores:**

If a full discussion period had occurred, I anticipate the following score movements:

- Reviewer i74L: Since this reviewer already provided a relatively high score, they would likely have maintained their assessment, remaining supportive but not increasing their score further.

- Reviewer 5a9t: This reviewer would likely have upgraded their score due to the convincing rebuttal regarding the model's motivation and practical scenarios.

- Reviewer ZZ8N: While this reviewer might have appreciated the additional results identifying the conditions under which the PoA can be bounded, it is doubtful whether they would have deemed this sufficient to raise their score, as the core novelty concerns persist.

- Reviewer Ntpc: Their score would likely have remained unchanged, as the fundamental doubts regarding technical depth were not resolved.

Conclusion: The paper sits on the borderline. However, while one reviewer (5a9t) might have turned more positive, the overall consensus lacks a champion who strongly advocates for the paper's technical significance. Given the persistent skepticism regarding the novelty of the analysis and the highly competitive standards of ICLR, I recommend a Reject.

---

### Decision · Program_Chairs · 2026-01-26

Reject